# PriorBand: Practical Hyperparameter Optimization in the Age of Deep Learning

**Neeratyoy Mallik**
University of Freiburg
mallik@cs.uni-freiburg.de

**Edward Bergman**
University of Freiburg
bergmane@cs.uni-freiburg.de

**Carl Hvarfner**
Lund University
carl.hvarfner@cs.lth.se

**Danny Stoll**
University of Freiburg
stolld@cs.uni-freiburg.de

**Maciej Janowski**
University of Freiburg
janowski@cs.uni-freiburg.de

**Marius Lindauer**
Leibniz University Hannover
m.lindauer@ai.uni-hannover.de

**Luigi Nardi**
Lund University
Stanford University
DBtune
luigi.nardi@cs.lth.se

**Frank Hutter**
University of Freiburg
fh@cs.uni-freiburg.de

## Abstract

Hyperparameters of Deep Learning (DL) pipelines are crucial for their downstream performance. While a large number of methods for Hyperparameter Optimization (HPO) have been developed, their incurred costs are often untenable for modern DL. Consequently, manual experimentation is still the most prevalent approach to optimize hyperparameters, relying on the researcher's intuition, domain knowledge, and cheap preliminary explorations. To resolve this misalignment between HPO algorithms and DL researchers, we propose `PriorBand`, an HPO algorithm tailored to DL, able to utilize both expert beliefs and cheap proxy tasks. Empirically, we demonstrate `PriorBand`'s efficiency across a range of DL benchmarks and show its gains under informative expert input and robustness against poor expert beliefs.

## 1 Introduction

The performance of Deep Learning (DL) models crucially depends on dataset-specific settings of their hyperparameters (HPs) [1, 2]. Therefore, Hyperparameter Optimization (HPO) is an integral step in the development of DL models [3–6]. HPO methods have typically been applied to traditional machine learning models (including shallow neural networks) that focus on fairly small datasets [7, 8]. Current DL practitioners, however, often utilize much larger models and datasets (e.g., a single training of GPT-3 [9] was estimated to require $3 \cdot 10^{23}$ FLOPS, i.e., months on a thousand V100 GPUs [10]). As recent HPO practices continue to apply simple techniques like grid, random, or manual search [11–14], existing HPO methods seem misaligned with DL practice.

To make the misalignment between existing HPO methods and DL practice explicit, we identify the following desiderata for an efficient, scalable HPO method suitable for current DL:

37th Conference on Neural Information Processing Systems (NeurIPS 2023).

Table 1: Comparison with respect to identified desiderata for DL pipelines. Shown is our algorithm, PriorBand, along with Grid Search, multi-fidelity random search (MF-RS), e.g. `HyperBand`, ASHA, `Bayesian Optimization` (BO), BO with Expert Priors ($\pi$BO), multi-fidelity BO (MF-BO), e.g. BOHB, Mobster. A checkmark and red cross indicates whether they satisfy the corresponding desideratum. Parenthesized checkmark indicates the desideratum is partly fulfilled, or requires additional modifications to be fulfilled. Model-based methods require customized kernels for discrete hyperparameters and asynchronous parallel approaches [7, 15] to achieve speed-up under (asynchronous) parallelism. Multi-fidelity algorithms' final performance is contingent on the budget spent on HPO.

| Criterion | Grid Search | MF-RS | BO | BO with Priors | MF-BO | PriorBand |
|---|---|---|---|---|---|---|
| Good anytime performance | ✗ | ✓ | ✗ | ✗ | ✓ | ✓ |
| Good final performance | ✗ | (✓) | ✓ | ✓ | ✓ | ✓ |
| Proxy task integration | ✗ | ✓ | ✗ | ✗ | ✓ | ✓ |
| Integrate expert beliefs | ✗ | ✗ | ✗ | ✓ | ✗ | ✓ |
| Mixed search spaces | ✓ | ✓ | (✓) | (✓) | (✓) | ✓ |
| Model free | ✓ | ✓ | ✗ | ✗ | ✗ | ✓ |
| Speedup under parallelism | ✓ | ✓ | (✓) | (✓) | (✓) | ✓ |

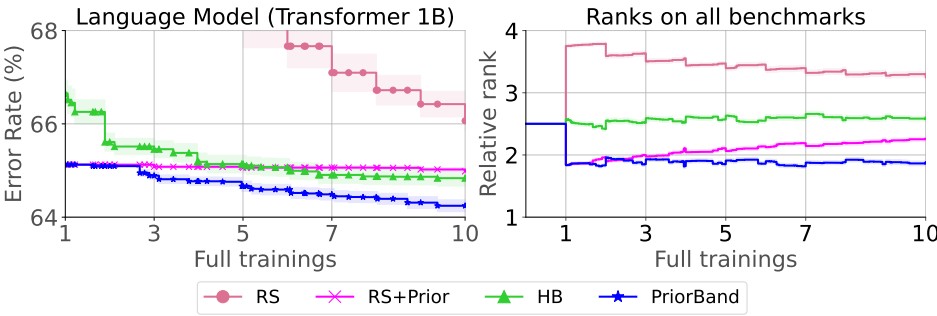

Figure 1: Both plots compare Random Search (RS), sampling from prior (RS+Prior), HyperBand (HB), and our method `PriorBand` which utilizes a *good* prior as defined by an expert. [**Left**] Tuning a large transformer on the 1B word benchmark. `PriorBand` leverages the prior, achieving strong anytime performance. [**Right**] Ranks on all our 12 benchmarks; `PriorBand` consistently ranks best.

1. **Strong performance under low compute budgets**: As DL pipelines become larger and more complex, only a few model training can be afforded to find performant configurations.
2. **Integrate cheap proxy tasks**: To save resources, cheap evaluations must be used to find promising configurations early while maintaining robustness to uninformative proxies.
3. **Integrate expert beliefs**: DL models often come with a competitive default setting or an expert may have prior beliefs on what settings to consider for HPO. Incorporating this information is essential while maintaining robustness to such beliefs being inaccurate.
4. **Handle mixed type search spaces**: Modern DL pipelines tend to be composed of many kinds of hyperparameters, e.g. categorical, numerical, and log hyperparameters.
5. **Simple to understand and implement**: The HPO method should be easy to apply to different DL pipelines and the behavior should be conceptually easy to understand.
6. **Parallelism**: Scaling to modern compute resources must be simple and effective.

Existing HPO algorithms satisfy subsets of these desiderata (see Table 1). For example, methods that utilize cheaper proxy tasks exploit training data subsets [16], fewer epochs/updates [17, 18], and proxy performance predictors [19, 20]. Previous works also consider how to combine expert knowledge into the optimization procedure in the form of the optimal value [21], meta or transfer learned knowledge [22–25] or configurations and regions that are known to the expert to have worked well previously [26–28]. Nonetheless, no HPO method exists that meets all of the desiderata above, especially, a simple model-free approach that can run cheaper proxy tasks, letting the DL expert integrate their domain knowledge for HPO.

We propose `PriorBand`, the first method to fulfill all desiderata. Our **contributions** are as follows:

1. We are the first to develop an approach to leverage cheap proxy tasks with an expert prior input (Section 2), and we show the need beyond a naive solution to do so (Section 4).

2. We contribute a general HPO algorithm with `PriorBand`, fulfilling all desiderata (Table 1) required for application to DL (Section 5; also see Figure 1)

3. We demonstrate the efficiency and robustness of `PriorBand` on a wide suite of DL tasks under practically feasible compute resources (Section 7).

4. We highlight the flexibility of the method behind `PriorBand` by improving different multi-fidelity algorithms and their model-based extensions (Section 7.2 & 7.3), which highlights the flexible, modular design and possible extensions of `PriorBand`.

We conclude, highlight limitations, and discuss further research directions in Section 10. Our code for reproducing the experiments is open-sourced at `https://github.com/automl/mf-prior-exp`.

## 2  Problem statement

We consider the problem of *minimizing* an expensive-to-evaluate objective function $f : \Lambda \to \mathbb{R}$, i.e., $\boldsymbol{\lambda}^* \in \arg\min_{\boldsymbol{\lambda} \in \Lambda} f(\boldsymbol{\lambda})$, where a configuration $\boldsymbol{\lambda}$ comes from a search space $\Lambda$ which may consist of any combination of continuous, discrete and categorical variables. In HPO particularly, $f$ refers to training and validating some model based on hyperparameters defined by $\boldsymbol{\lambda}$.

Multi-fidelity optimization of $f(\boldsymbol{\lambda})$ requires a proxy function, namely $\hat{f}(\boldsymbol{\lambda}, \boldsymbol{z})$, that provides a cheap approximation of $f$ at a given fidelity $\boldsymbol{z}$, e.g., the validation loss of a model only trained for $\boldsymbol{z}$ epochs. Our methodology considers a fidelity scale $\boldsymbol{z} \in [\boldsymbol{z}_{\min}, \boldsymbol{z}_{\max}]$ such that evaluating at $\boldsymbol{z}_{\max}$ coincides with our true objective: $f(\boldsymbol{\lambda}) = \hat{f}(\boldsymbol{\lambda}, \boldsymbol{z}_{\max})$.

To take all desiderata into account, we additionally include a user-specified belief $\pi : \Lambda \to \mathbb{R}$, where $\pi(\boldsymbol{\lambda}) = \mathbb{P}(\boldsymbol{\lambda} = \boldsymbol{\lambda}^*)$ represents a subjective probability that a given configuration $\boldsymbol{\lambda}$ is optimal. Thus, the overall objective is

$$\boldsymbol{\lambda}^* \in \arg\min_{\boldsymbol{\lambda} \in \Lambda} \hat{f}(\boldsymbol{\lambda}, \boldsymbol{z}_{\max}), \quad \text{guided by } \pi(\boldsymbol{\lambda}). \tag{1}$$

Our problem is to efficiently solve Equation 1 under a constrained budget, e.g., 10 full trainings of a DL model. Given DL training can diverge, the *incumbent* we return is the configuration-fidelity pair $(\boldsymbol{\lambda}, \boldsymbol{z})$ that achieved the lowest error while optimizing for Equation 1.

## 3  Background

While we are the first to target Equation 1 in its full form, below, we introduce the necessary background on methods that use either multi-fidelity optimization *or* expert beliefs.

**Successive Halving (SH)** optimizes Equation 1 (sans $\pi$) as a best-arm identification problem in a multi-armed bandit setting [29]. Given a lower and upper fidelity bound ($[\boldsymbol{z}_{\min}, \boldsymbol{z}_{\max}]$) and a reduction factor ($\eta$), SH discretizes the fidelity range geometrically with a log-factor of $\eta$. For example, if the inputs to SH are $\boldsymbol{z} \in [3, 81]$ and $\eta = 3$, the fidelity levels in SH are $\boldsymbol{z} = \{3, 9, 27, 81\}$ with accompanying enumeration called rungs, $\mathrm{r} = \{0, 1, 2, 3\}$. Any $\boldsymbol{z} < \boldsymbol{z}_{\max}$ is called a lower fidelity, a cheaper proxy task in DL training. Every iteration of each round of SH involves uniform random sampling of $\eta^{s_{\max}-1}$ configurations at the lowest fidelity or $\mathrm{r} = 0$, where $s_{\max} = \lfloor \log_\eta(\boldsymbol{z}_{\max}/\boldsymbol{z}_{\min}) \rfloor$. After evaluating all samples, only the top-performing $(1/\eta)$ configurations are retained, discarding the rest. Subsequently, these $\eta^{s_{\max}-1}/\eta$ configurations are evaluated at $\mathrm{r} = 1$, or the next higher discretized fidelity. This is repeated till there is only one surviving configuration at the highest $\mathrm{r}$. Sampling and evaluating many configurations for cheap, performing a tournament selection to retain strong samples, and repeating this, ensures that the high fidelity evaluations happen for relatively stronger configurations only. SH proves that such an early-stopping strategy guarantees a likely improvement over random search under the same total evaluation budget. Due to its random sampling component, SH is an example of MF-RS in Table 1. See Appendix D.2 for an illustrative example.

**HyperBand (HB)** attempts to address SH's susceptibility to poor fidelity performance correlations [30]. It has the same HPs as SH, and thus given SH's fidelity discretization, HB iteratively executes multiple rounds of SH where $\boldsymbol{z}_{\min}$ is set to the next higher discrete fidelity. Using the

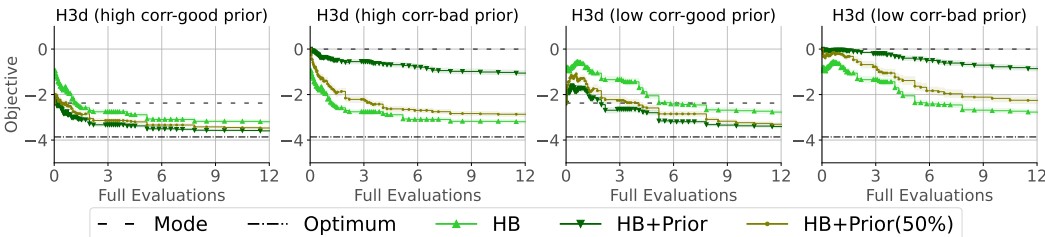

Figure 2: A comparison of naive solutions to Equation 1 on the 3-dimensional multi-fidelity Hartmann benchmarks. We compare different versions of HB, utilizing different strengths of the prior distribution $\pi(\cdot)$: HB with 0% influence of the prior (HB), HB with 50% sampling from the prior (HB+Prior(50%)) and HB with 100% sampling from prior (HB+Prior).

example from SH, given $z \in [3, 81]$ and $\eta = 3$, one full iteration of HB corresponds to running SH($z_{\min} = 3, z_{\max} = 81$) followed by SH($z_{\min} = 9, z_{\max} = 81$), SH($z_{\min} = 27, z_{\max} = 81$) and SH($z_{\min} = 81, z_{\max} = 81$). Notably, such different instantiations of SH in HB do not share any information with each other implicitly. Each execution of SH($z_{\min}, z_{\max}$) is called an SH *bracket*. The sequence of all unique SH brackets run by HB in one iteration is called an HB *bracket*.

$\pi$**BO** utilizes the unnormalized probability distribution $\pi(\boldsymbol{\lambda})$ in a BO context to accelerate the optimization using the knowledge provided by the user [28]. In $\pi$BO, the initial design is sampled entirely from $\pi$ until Bayesian Optimization (BO) begins, where the acquisition function $\alpha$ is augmented with a prior term that decays over time $t$: $\alpha_\pi^t(\boldsymbol{\lambda}) = \alpha(\boldsymbol{\lambda}) \cdot \pi(\boldsymbol{\lambda})^{\frac{\beta}{t}}$, where $\beta$ is an HP. In this work, we borrow $\pi(\cdot)$ as the expert prior interface to the HPO problem. $\pi$BO relies on the decaying prior on the acquisition $\alpha_\pi^t(\boldsymbol{\lambda})$ to gradually add more weight to the model and thus allow recovery from a poor initial design under a bad prior. We choose the model-free setting and adapt the strength of the prior based on the evidence of its usefulness.

## 4 The need to move beyond a naive solution

To solve Equation 1, an intuitive approach is to augment an existing multi-fidelity algorithm, e.g., HyperBand (HB), with sampling from the prior $\pi(\cdot)$. In this section, however, we show that this naive approach is not sufficient, motivating the introduction of our algorithm, PriorBand, in Section 5.

To study the naive combination of HyperBand with prior-based sampling, we use the parameterized 3-dimensional multi-fidelity Hartmann benchmarks [31] (Appendix D.1.1) and compare HB with uniform sampling, HB with 100% prior-based sampling, and HB with 50% uniform and prior-based sampling (Figure 2). Unsurprisingly, 100% sampling from priors works best when the prior is helpful and not sampling from the prior works best for misleading priors. 50% random sampling works quite robustly, but is never competitive with the best approach. To achieve robustness to the quality of the prior *and* rival the best approach for each case, we introduce `PriorBand` below.

## 5 PriorBand: Moving beyond a naive solution

The key idea behind `PriorBand` is to complement the sampling strategies used in the naive solution, uniform sampling, and prior-based sampling, with a third strategy: incumbent-based sampling. Thereby, we overcome the robustness issues of the naive solutions presented in Section 4, while still fulfilling the desideratum for simplicity. We first introduce and motivate sampling around the incumbent (Section 5.1), and then describe the ensemble sampling policy $\mathcal{E}_\pi$ that combines all three sampling strategies and incorporates $\mathcal{E}_\pi$ into HB (Section 5.2) for `PriorBand`.

### 5.1 Incumbent-based sampling strategy, $\hat{\lambda}(\cdot)$

`PriorBand` leverages the current incumbent to counter uninformative priors while supporting good priors, as the current incumbent can be seen as indicating a likely good region to sample from. Note that this view on the region around the incumbent is close to the definition of $\pi(\cdot)$ in $\pi$BO (Section

3), where the prior distribution encodes the expert's belief about the location of the global optima and thus a good region to sample from.

To construct the incumbent-based sampler $\hat{\lambda}(\cdot)$, we perform a local perturbation of the current best configuration. Each hyperparameter (HP) is chosen with probability $p$ for perturbation. If chosen, continuous HPs are perturbed by $\epsilon \sim \mathcal{N}(0, \sigma^2)$. For discrete HPs, we resample with a uniform probability for each categorical value except the incumbent configuration's value which has a higher probability of selection, discussed further in Appendix E.2.5. For `PriorBand`, we fix these values at $p = 0.5$ and $\sigma = 0.25$. This perturbation operation is simple, easy to implement, and has constant time complexity. We show ablations with two other possible local search designs in Appendix E.2.3.

## 5.2 `PriorBand`: The $\mathcal{E}_\pi$-augmented HyperBand

`PriorBand` exploits HB for scheduling and replaces its random sampling component with a combination of random sampling, prior-based sampling, and incumbent-based sampling. We denote the proportions of these individual sampling components as $p_\mathcal{U}$, $p_\pi$, and $p_{\hat{\lambda}}$, respectively, and their combination as the *ensemble sampling policy (ESP) $\mathcal{E}_\pi$*.

Figure 3 (left) illustrates `PriorBand` as an extension of HB that, next to the HB hyperparameters ($z_{\min}, z_{\max}, \eta$, budget) accepts the expert prior $\pi$ as an additional input and uses the ESP $\mathcal{E}_\pi$ in lieu of random sampling. This sampling from $\mathcal{E}_\pi$ is illustrated in Algorithm 1. Note that $\mathcal{E}_\pi$ has access to the optimization state ($s_t$) and can thus reactively adapt its individual sampling probabilities $p_\mathcal{U}$, $p_\pi$, and $p_{\hat{\lambda}}$ based on the optimization history. We now discuss how we decay the random sampling probability $p_\mathcal{U}$ (Section 5.2.1); and the proportion of incumbent and prior sampling (Section 5.2.2).

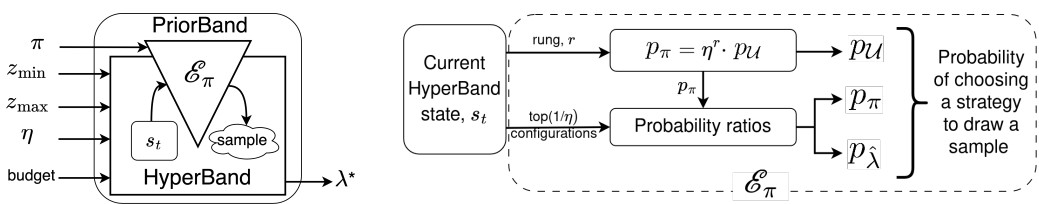

Figure 3: `PriorBand` schema; [**Left**] in the base algorithm vanilla-HB, we replace the random sampling module by $\mathcal{E}_\pi$, which can interface an expert prior $\pi$ and access the state of HB; [**Right**] $\mathcal{E}_\pi$ reads the state every iteration and determines the probabilities for selecting a sampling strategy.

---

**Algorithm 1** Sampling from $\mathcal{E}_\pi$

1: **Input:** $s, s_{\max}, \eta$, observations $\mathcal{H}$, prior $\pi$
2: $r = s_{\max} - s$     $\triangleright s_{\max}, s$ input from HB
3: $p_\mathcal{U} = 1/(1 + \eta^r), p_\pi = 1 - p_\mathcal{U}, p_{\hat{\lambda}} = 0$
4: **if** activate_incumbent($s_t$) **then**
5:     $p_\pi, p_{\hat{\lambda}} \leftarrow$ Algorithm 2($\mathcal{H}, r, p_\pi$)
6: **end if**
7: $d(\cdot) \leftarrow$ sample strategy by $\{p_\mathcal{U}, p_\pi, p_{\hat{\lambda}}\}$
8: $\boldsymbol{\lambda} \leftarrow$ sample from $d(\cdot)$
9: **return** $\boldsymbol{\lambda}$

---

**Algorithm 2** Dynamic weighting of $w_{\hat{\lambda}}$

1: **Input:** observations $\mathcal{H}$, rung $r$, $p_\pi^{\text{old}}$
2: $\Lambda'_z = \{\boldsymbol{\lambda}_i\}_{1:n} \leftarrow \text{top\_}(1/\eta)(\mathcal{H}, r)$
3: $\mathcal{S}_{\hat{\lambda}} \leftarrow \sum_{\Lambda'_z} w_i \cdot \hat{\lambda}(\boldsymbol{\lambda}_i)$     $\triangleright w_i = (n + 1) - i$
4: $\mathcal{S}_\pi \leftarrow \sum_{\Lambda'_z} w_i \cdot \pi(\boldsymbol{\lambda}_i)$
5: $p_{\hat{\lambda}} \leftarrow p_\pi^{\text{old}} \cdot \mathcal{S}_{\hat{\lambda}}/(\mathcal{S}_\pi + \mathcal{S}_{\hat{\lambda}})$
6: $p_\pi \leftarrow p_\pi^{\text{old}} \cdot \mathcal{S}_\pi/(\mathcal{S}_\pi + \mathcal{S}_{\hat{\lambda}})$
7: **return** $p_\pi, p_{\hat{\lambda}}$

---

### 5.2.1 Decaying proportion of random sampling

Given the premise that we should initially trust the expert prior, yet with the benefit of incorporating random sampling (from Section 4), we make two additional assumptions, namely (i) we trust the expert's belief most at the maximum fidelity $z_{\max}$; and (ii) we would like to use cheaper fidelities to explore more. Given HB's discretization of the fidelity range $[z_{min}, z_{max}]$ into rungs $r \in \{0, \ldots, r_{max}\}$, we geometrically increase our sampling probability from $\pi(\cdot)$ over $\mathcal{U}(\cdot)$ by

$$p_\pi = \eta^r \cdot p_\mathcal{U}, \tag{2}$$

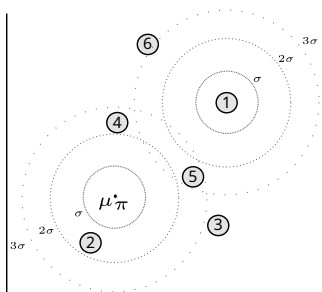

Figure 4: A visual example of how configurations contribute to the scores $\mathcal{S}_\pi$ and $\mathcal{S}_{\hat{\boldsymbol{\lambda}}}$. The prior distribution here is Gaussian $\mathcal{N}(\mu_\pi, \sigma^2)$, with a matching distribution $\hat{\boldsymbol{\lambda}}(\cdot) = \mathcal{N}(\hat{\boldsymbol{\lambda}}, \sigma^2)$ placed on the incumbent. The labels 1-6 indicate configurations and their rank in $\Lambda'_z$. Here, e.g., $\boldsymbol{\lambda}_2$ has densities such that $\pi(\boldsymbol{\lambda}_2) > \bar{\boldsymbol{\lambda}}(\boldsymbol{\lambda}_2)$, contributing more to $\mathcal{S}_\pi$ than to $\mathcal{S}_{\hat{\boldsymbol{\lambda}}}$. The configurations $\boldsymbol{\lambda}_3, \boldsymbol{\lambda}_4, \boldsymbol{\lambda}_5$ are between both distributions and $\boldsymbol{\lambda}_6$ has low density under either, whereas $\boldsymbol{\lambda}_1$ (the incumbent) will be the primary influence such that $\mathcal{S}_{\hat{\boldsymbol{\lambda}}} > \mathcal{S}_\pi$. This implies that $p_{\hat{\boldsymbol{\lambda}}} > p_\pi$ and demonstrates that all else being relatively equal, we will be more likely to sample from $\hat{\lambda}(\cdot)$.

with the constraint that $p_{\mathcal{U}} + p_\pi = 1$ (see L3, Algorithm 1). This naturally captures our assumptions, equally favouring $\pi(\cdot)$ and $\mathcal{U}(\cdot)$ initially but increasing trust in $\pi(\cdot)$ according to the rung r we sample at. This geometric decay was inspired by similar scaling used in HB's scheduling and backed by ablations over a linear decay and constant proportion (Appendix E.2.1).

### 5.2.2 Incumbent-based sampling proportion

Incumbent-based sampling intends to maintain strong anytime performance even under bad, uninformative, or adversarial priors. In `PriorBand`, initially $p_{\hat{\lambda}} = 0$ until both (i) a budget equivalent to the first SH bracket has been exhausted ($\approx \eta \cdot z_{\max}$); and (ii) at least one configuration has been evaluated at $z_{\max}$. This is captured in Line 4 of Algorithm 1 by `activate_incumbent()`.

Once incumbent-based sampling is active we need to decide how much to trust $\hat{\lambda}(\cdot)$ vs. trusting $\pi(\cdot)$. Essentially, we achieve this by rating how likely the best seen configurations would be under $\hat{\lambda}(\cdot)$ and $\pi(\cdot)$, respectively. Algorithm 2 shows the detailed steps to calculate $p_\pi$ and $p_{\hat{\lambda}}$, and Figure 4 provides an example. We first define an ordered set $\Lambda'_z = \{\boldsymbol{\lambda}_1, \ldots, \boldsymbol{\lambda}_n\}$, ordered by performance, of the top-performing $(1/\eta)$ configurations[1] for the highest rung r with at least $\eta$ evaluated configurations (Line 2 in Algorithm 2). Given $\Lambda'_z$, we compute two weighted scores $\mathcal{S}_\pi, \mathcal{S}_{\hat{\lambda}}$, capturing how likely these top configurations are under the densities of $\pi(\cdot)$ and $\hat{\lambda}(\cdot)$, respectively. We also weigh top-performing configurations more highly, which is accomplished with a simple linear mapping $w_i = (n+1) - i$ (Lines 3-4 in Algorithm 2). Finally, we obtain $p_\pi$ and $p_{\hat{\lambda}}$ by normalizing these scores as a ratio of their sum (Lines 5-6 in Algorithm 2). We note the rates are adaptive based on how the set $\Lambda'_z$ is spread out relative to the prior and the incumbent. We observe that this adaptive behavior is crucial in quickly recovering from bad prior inputs and thus maintaining strong final performance.

## 6 Experimental setup

We now describe our experiment design, benchmarks, and baselines and demonstrate the robustness of `PriorBand` in handling varying qualities of an expert prior input. Additionally, we also showcase the generality and practicality of the ESP, $\mathcal{E}_\pi$.

### 6.1 Benchmarks

We curated a set of 12 benchmarks that cover a diverse set of search spaces, including mixed-type spaces and log-scaled hyperparameters, and a wide range of downstream tasks, e.g., language modeling, image classification, tabular data, a medical application, and translation. We select 4 of the PD1 benchmarks (4 HPs) [32] that train large models such as transformers with batch sizes commonly found on modern hardware, and fit surrogates on them. Further, we select 5 benchmarks from LCBench (7 HPs) [33, 34] and consider all 3 JAHSBench [35] surrogate benchmarks that offer a 14 dimensional mixed-type search space for tuning both the architecture and training hyperparameters. All benchmarks and their selection are described in further detail in Appendix D.1.

---

[1]We want to use at least $\eta$ configurations; hence $\max(\eta, \text{\# of configurations in the rung} / \eta)$ are selected.

## 6.2 Baselines

We choose a representative from each of the families of optimizers listed in Table 1, leaving out grid search. We use the official implementation of BOHB, while all other algorithms were implemented by us (and are available as part of our repository). The prior-based baselines ($\pi$BO and RS+Prior) sample the mode of the prior distribution as the first evaluation in our experiments to ensure a fair comparison where the prior configuration is certainly evaluated, irrespective of it being good or bad. For all the HB based algorithms we use $\eta = 3$ and the fidelity bounds ($z_{\min}, z_{\max}$) as per the benchmark. Further implementation and hyperparameter details for the baselines can be found in Appendix D.2. In principle, `PriorBand` only needs an additional input of $\pi(\cdot)$ as the user belief, other than the standard HB hyperparameters. However, for a discussion on the hyperparameters of the incumbent-based local search (Section 5.1), please refer to Appendix E.2.5.

## 6.3 Design and setup

We show experiments both for single and multi-worker cases (4 workers in our experiments). For the single workers, we report the mean validation error with standard error bars for $50$ seeds; for multi-worker experiments, we use $10$ seeds. The plots show average relative ranks achieved aggregated across all 12 benchmarks when comparing the anytime incumbent configuration at $z_{\max}$. We group the runs on benchmark under different qualities of an expert prior input. We also compare the average normalized regret per benchmark under good priors. The prior construction procedure follows the design from the work by Hvarfner et al. [28], and we describe our procedure in detail in Appendix D.3. In the main paper, we only evaluate the good and bad prior settings as we believe that this reflects a practical setting of prior qualities; in Appendix F we also evaluate the near-optimal prior settings and also evaluate all 3 prior settings over high and low-performance correlation problems. In Appendix F.5, we also report additional experiments for different budget horizons. Further experiment design details can be found in Appendices D.4 and D.5. As an example of potential post-hoc analysis possible with `PriorBand`, we show how the dynamic probabilities calculated for each of the sampling strategies in `PriorBand` can be visualized (Appendix E.3), revealing the quality of the prior used.

# 7 Results

We now report and discuss the results of our experiments.

## 7.1 Robustness of `PriorBand`

Here, we demonstrate the robustness of `PriorBand` over a wide range of prior qualities by comparing them to the nearest non-prior-based algorithms: random search (RS) and `HyperBand` (HB). Figure 5(top) showcases `PriorBand` to be anytime equal or better than HB on each of our benchmarks under our good prior design. Note that the quality of a good prior varies per benchmark. The aggregation of Figure 5(top) is Figure 5(bottom-middle), which illustrates that `PriorBand` (HB+$\mathcal{E}_\pi$) can utilize good priors and gain strong anytime performance over HB. Moreover, Figure 5(bottom-right) clearly shows `PriorBand`'s ability to recover from bad prior information and match vanilla-HB's performance in this adversarial case (the bad prior was intentionally chosen to be an extremely poor configuration – the worst of $50k$ random samples). In most practical scenarios, the expert has better intuition than this setup and one can expect substantial speedups over HB, like in Figure 5(bottom-middle). Figure 5(bottom-left) demonstrates that when using an unknown quality of prior, even including the adversarial prior, `PriorBand` is still the best choice on average. We show a budget of 12 function evaluations here, which is approximately the maximal budget required for completing at least one HB iteration for the benchmarks chosen. In Appendix F.1 we show more comparisons to prior-based baselines and highlight `PriorBand`'s robustness.

## 7.2 Generality of Ensemble Sampling Policy $\mathcal{E}_\pi$

The ESP, $\mathcal{E}_\pi$, only needs access to an SH-like optimization state as input. In this section, we show that other popular multi-fidelity algorithms, such as ASHA or asynchronous-HB [36] that build on SH, can also support expert priors with the help of $\mathcal{E}_\pi$. In Figure 6, we compare the vanilla algorithms with their $\mathcal{E}_\pi$-augmented versions. Given that these algorithms were designed for parallel setups,

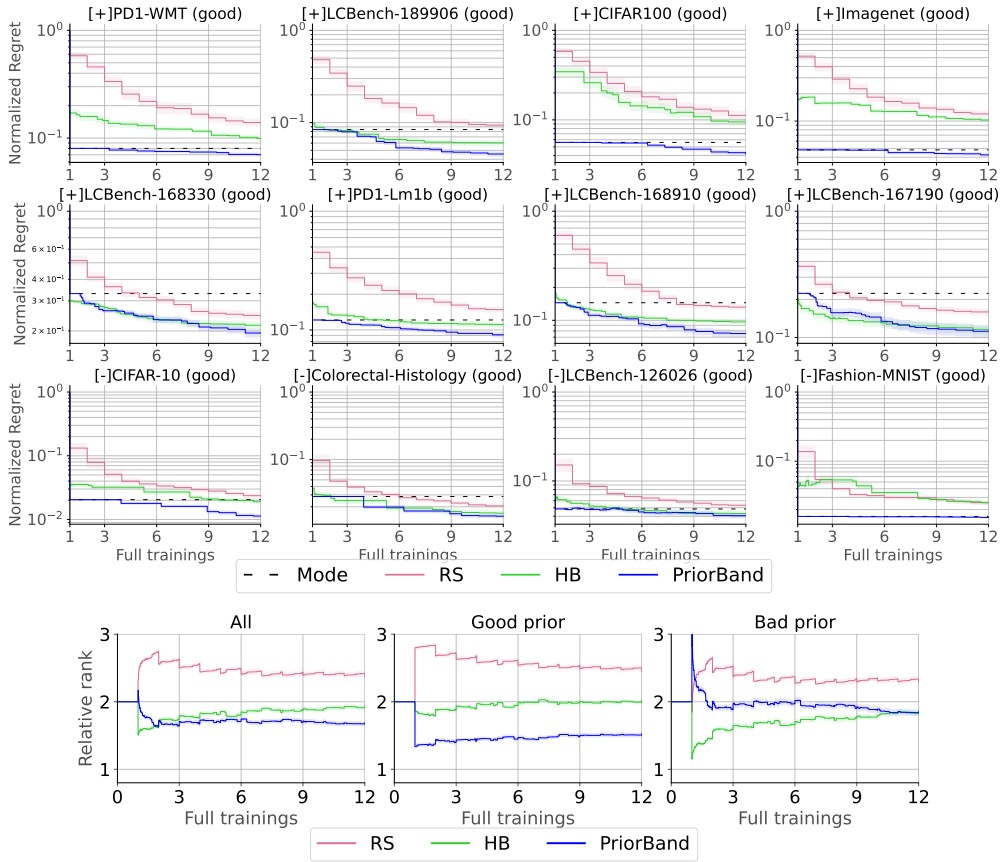

Figure 5: [Top] Comparing normalized regret in the good prior setting; The [Bottom] figure *Good* (middle) is a ranking aggregate view of the [Top] figure. [Bottom] Comparing average relative ranks of `PriorBand` to Random Search (RS) and HB, over single worker runs across benchmarks. Each benchmark-algorithm pair was run for $50$ seeds where priors per benchmark are the same across a seed. We show mean rank and standard error. The *All* (left) plot averages the benchmark across Good and Bad priors.

we compare them on runs distributed among $4$ workers, running for a total budget of $20$ function evaluations. Similar to the previous section, the ESP-augmented (+ESP) algorithms can leverage good priors and recover under bad priors. Under bad priors, asynchronous-HB (+ESP) starts worst owing to bad prior higher fidelity evaluations at the start but shows strong recovery. In Appendix F.2 we compare these variants over different correlation settings of the benchmarks.

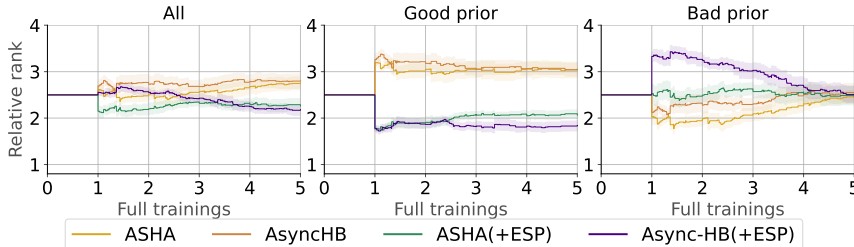

Figure 6: Comparing the average relative ranks of Asynchronous-SH (ASHA) and Asynchronous-HB (Async-HB) and their variants with the Ensemble Sampling Policy (+ESP) when distributed over $4$ workers for a cumulative budget of $20$ function evaluations.

## 7.3 Extensibility with models

Although our focus is the model-free low compute regime, we also show how `PriorBand` can be optionally extended with model-based surrogates to perform Bayesian Optimization, especially when longer training budgets are available. We compare BO, $\pi$BO, BOHB, and `PriorBand` with its model-extension, `PriorBand+BO` (Gaussian Processes as surrogates and Expected Improvement [37] for acquisition). All BO methods use an initial design of 10 function evaluations, except BOHB which sets this implicitly as $N_{\text{dim}} + 2$. Compared to other prior-based algorithms (RS+Prior, $\pi$BO), `PriorBand+BO` is consistently robust on average (*All*) under all priors. The anytime performance gain of `PriorBand+BO` under good priors is evidence of ESP $\mathcal{E}_\pi$'s utility. We note that `PriorBand+BO` recovers quickly in the bad prior setting and even outperforms all other algorithms after as little a budget as 6 full trainings. The fact that, until $10\times$ function evaluations, `PriorBand+BO` is actually just `PriorBand` and model search has not begun, highlights the effectiveness of ESP in adaptively trading off different sampling strategies and the initial strength of incumbent sampling. Appendices E.4, F.3 contain details on our modeling choices and more experimental results.

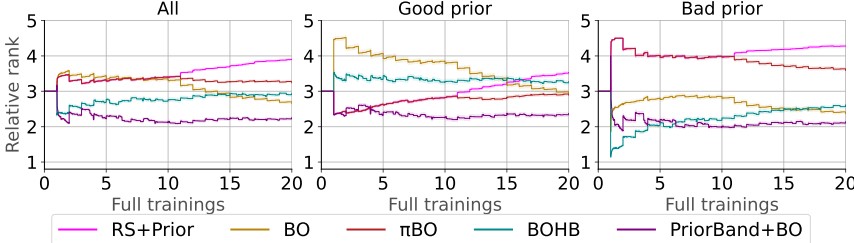

Figure 7: Comparing the average relative ranks of model-based algorithms and sampling from the prior (RS+Prior) under single-worker runs, aggregated over all benchmarks-per-prior quality.

## 7.4 Ablation studies

In Figure 8, we compare our choice of using density scores (Section 5.2.2) to trade-off prior-based and incumbent-based sampling against other similar or simpler heuristics, validating our decision. `PriorBand(constant)` employs a fixed heuristic to trade off prior and incumbent-based sampling as $p_\pi = \eta \cdot p_{\hat{\lambda}}$. In `PriorBand(decay)`, $p_\pi$ is decayed as a function of iterations completed. More precisely, $p_{\hat{\lambda}} = 2^b \cdot p_\pi$, subject to $p_\pi + p_{\hat{\lambda}} + p_{\mathcal{U}} = 1$ (from Section 5.2), where $b \in \mathbb{N}$ and indicates the index of the current SH being run in HB. Appendix E.2.2 contains more ablations.

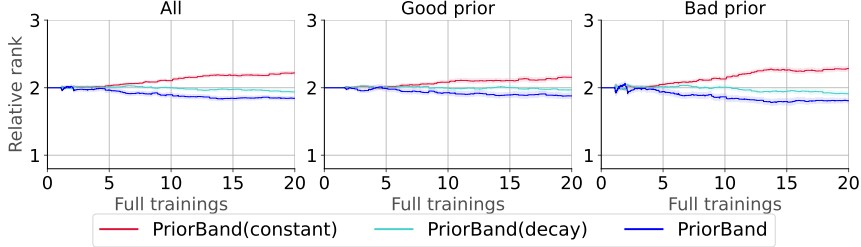

Figure 8: A comparison of `PriorBand` with 3 different strategies for trading off incumbent vs. prior distribution sampling. `PriorBand`'s default of using density scores shows a dominant performance in almost all scenarios. However, a bad prior induces a marked difference in performance. In general, the plot highlights that each variant reacts differently to bad priors. The adaptive nature of `PriorBand` clearly is robust to different scenarios comparatively.

## 8 Related work

While using expert priors and local search [38] for hyperparameter optimization has been explored previously, few works considered priors over the optimum [27, 28, 39], and they all target the

single-fidelity setting. The expert priors we consider should not be confused with the priors natively supported by BO, i.e., priors over the function structure determined by kernels [7, 40, 41].

In deep learning, training epochs and dataset subsets [17, 42, 43] are frequently used as fidelity variables to create cheap proxy tasks, with input resolution, network width, and network depth also occasionally used [35]. SH [29] and HB [30] are effective randomized policies for multi-fidelity HPO that use early stopping of configurations on a geometric spacing of the fidelity space and can also be extended to the model-based setting [44].

## 9 Limitations

We acknowledge that `PriorBand` inherits pathologies of HB such as poor fidelity correlations and can be sensitive to the choice of fidelity bounds and early stopping rate. Despite that, our results across different correlation settings suggest a relatively strong performance of `PriorBand` (Appendix F). Though `PriorBand` supports any kind of prior distribution as input, in our experiments we only considered the Gaussian distribution (with a fixed standard deviation of $0.25$), as it is a natural choice and was used previously in the literature [28]. However, the expert is free to represent the prior with other distributions. We expect `PriorBand` to show similar behaviors as we report when comparing to other prior-based algorithms under a similar distribution. We note that `PriorBand` is not entirely free of hyperparameters. In our experiments, we keep all `PriorBand` hyperparameters fixed to remove confounding factors while comparing different prior strengths and correlations. Moreover, our hyperparameter choices for the experiments, such as $\eta = 3$ and Gaussian priors are largely borrowed from existing literature [28, 36, 44]. Depending on the context of the specific deep learning expert, the compute budgets ($\sim 10 - 12$ full trainings) used in our experimental setting may not always be feasible. However, the key insight from the experiments is that `PriorBand` can interface informative priors, and provide strong anytime performance under short HPO budgets (also in less than $5$ full trainings). Longer HPO budgets for `PriorBand` ensure recovery from potential bad, uninformative priors. In practice, deep learning experts often have good prior knowledge and thus `PriorBand` is expected to retain strong anytime performance under low compute.

## 10 Conclusion

We identify that existing HPO algorithms are misaligned with deep learning (DL) practice and make this explicit with six desiderata (Section 1). To overcome this misalignment, our solution, `PriorBand`, allows a DL expert to incorporate their intuition of well-performing configurations into multi-fidelity HPO and thereby satisfies all desiderata to be practically useful for DL. The key component in `PriorBand`, the ESP, $\mathcal{E}_\pi$, is modular, flexible, and can be applied to other multi-fidelity algorithms.

## 11 Acknowledgments and disclosure of funding

N. Mallik, D. Stoll and F. Hutter acknowledge funding by the European Union (via ERC Consolidator Grant DeepLearning 2.0, grant no. 101045765). Views and opinions expressed are however those of the author(s) only and do not necessarily reflect those of the European Union or the European Research Council. Neither the European Union nor the granting authority can be held responsible for them. E. Bergman was partially supported by TAILOR, a project funded by EU Horizon 2020 research and innovation programme under GA No 952215. C. Hvarfner and L. Nardi were partially supported by the Wallenberg AI, Autonomous Sytems and Software Program (WASP) funded by the Knut and Alice Wallenberg Foundation. L. Nardi was supported in part by affiliate members and other supporters of the Stanford DAWN project — Ant Financial, Facebook, Google, Intel, Microsoft, NEC, SAP, Teradata, and VMware. Luigi Nardi was partially supported by the Wallenberg Launch Pad (WALP) grant Dnr 2021.0348. M. Lindauer was funded by the European Union (ERC, "ixAutoML", grant no.101041029). The research of Maciej Janowski was supported by the Deutsche Forschungsgemeinschaft (DFG, German Research Foundation) under grant number 417962828 and grant INST 39/963-1 FUGG (bwForCluster NEMO). In addition, Maciej Janowski acknowledges the support of the BrainLinks- BrainTools Center of Excellence.

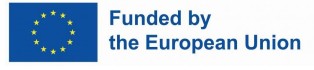

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
