## A    Resources used

All experiments in the paper were performed on cheap-to-evaluate surrogate benchmarks. We used several Intel(R) Xeon(R) Gold 6242 CPUs @ 2.80GHz to perform our experiments. Running one seed of one algorithm on one benchmark requires on average $\sim 0.2$ core hours. For single-worker experiments, we run 21 algorithms with 50 seeds over 16 benchmarks with 3 strengths of priors per benchmark, totaling $50,400$ single worker runs which equates to $10,080$ core hours. In the parallel set of experiments, we use 4 cores per run, limiting ourselves to only 12 algorithms, 10 seeds, 7 benchmarks[2] with all 3 prior strengths. This totals $10,080$ workers run in total which equates to $2,016$ total core hours.

We additionally trained surrogate models for 2 metrics on 3 datasets for 4 hours with 8 cores, totaling another 192 core hours. During the development of our final algorithm, including failed experiments, re-runs, and preliminary testing, we estimate roughly another $\sim 2,000$ core hours, a fifth of our total final cost. We estimate our total usage to have totaled $\sim 14,288$ core hours.

## B    Societal and environmental impact

Here, we discuss the potential societal and environmental impacts our work can have.

**Environmental**    The contributed algorithm `PriorBand` and its re-usable component $\mathcal{E}_\pi$ is designed to help reduce compute requirements for finding performant DL pipelines, thus reducing carbon emissions spent for HPO in DL. However, with the surplus compute available to many larger organizations, enabling robust methods for HPO could encourage further utilization of otherwise unused compute.

**Societal**    Our paper and the contributed algorithm `PriorBand` are designed to assist a wide range of DL practitioners in finding performant hyperparameters. The ability of `PriorBand` to tune DL models under affordable compute enables practitioners to find strong hyperparameters otherwise only tenable for larger organizations. The societal impact depends on which task and DL pipeline `PriorBand` is applied to.

## C    Licenses

- Our implementations - **MIT License**
- JAHS-Bench-201 benchmark [35] - **MIT License**
- YAHPO-Gym benchmark [34] - **Apache License 2.0**
- Learning curve benchmark [47] - **Apache License 2.0**
- PD1 [32] - **Apache License 2.0**
- BOHB [48] from HpBandSter - **BSD 3-Clause License**

## D    Experiment details

### D.1    Benchmarks

Following Equation 1, we frame all the benchmark tasks as a minimizing problem. The benchmarks we use are provided by our curated suite of multi-fidelity benchmarks (`mf-prior-bench` ) that treats priors as first-class citizen. We include our own synthetic Hartmann functions (D.1.1) extended to the multi-fidelity setting. We wrap JAHS-Bench-201  [35] (D.1.2) and Yahpo-Gym  [34] (D.1.4) and provide new surrogate benchmarks for large models for image and language tasks, trained from optimization data obtained from the PD1 benchmark from HyperBO  [32] (D.1.3).

---

[2]Please see D.1.6 as for why we do not include all benchmarks for parallel runs.

### D.1.1 Multi-fidelity synthetic Hartmann (MFH)

The multi-fidelity Hartmann functions follow the design of Kandasamy et al. [31], where, for $[0, 1]$-scaled $z$, the fidelity is parameterized as

$$g(\boldsymbol{x}, z) = \sum_{i=1}^{4}(\alpha_i - \alpha_i^{'}(z; b)) \exp\left(-\sum_{j=1}^{D} A_{ij}(x_j - P_{ij})^2\right) \quad (3)$$

where for Hartmann-3,

$$A = \begin{bmatrix} 3 & 10 & 30 \\ 0.1 & 10 & 35 \\ 3 & 10 & 30 \\ 0.1 & 10 & 35 \end{bmatrix}, \quad P = 10^{-4} \times \begin{bmatrix} 3689 & 1170 & 2673 \\ 4699 & 4387 & 7470 \\ 1091 & 8732 & 5547 \\ 381 & 5743 & 8828 \end{bmatrix},$$

and for Hartmann-6,

$$A = \begin{bmatrix} 10 & 3 & 17 & 3.5 & 1.7 & 8 \\ 0.05 & 10 & 17 & 0.1 & 8 & 14 \\ 3 & 3.5 & 1.7 & 10 & 17 & 8 \\ 17 & 8 & 0.05 & 10 & 0.1 & 14 \end{bmatrix}, \quad P = 10^{-4} \times \begin{bmatrix} 1312 & 1696 & 5569 & 124 & 8283 & 5886 \\ 2329 & 4135 & 8307 & 3736 & 1004 & 9991 \\ 2348 & 1451 & 3522 & 2883 & 3047 & 6650 \\ 4047 & 8828 & 8732 & 5743 & 1091 & 381 \end{bmatrix}$$

where $\alpha_i^{'}(z; b)) = b(1 - z_i)$. In the original paper, the variable which accounts for the bias between fidelities, $b$, is set to $0.1$. To account for the fact that we only consider a single fidelity variable, we set $z_i = z, \forall i$, and increase the bias terms significantly to create realistic task correlations.

For the **good correlation** version used in our evaluations, we set $b = 2.5$ with half-normally distributed noise of $\sigma = 2(1 - z)$.
The **bad correlation** version uses $b = 4$ with a noise of $\sigma = 5(1 - z)$.

Tables 2, 3 show the search space for this synthetic benchmark.

| name | type | values | info |
|------|------|--------|------|
| X_0 | continuous | $[0.0, 1.0]$ | |
| X_1 | continuous | $[0.0, 1.0]$ | |
| X_2 | continuous | $[0.0, 1.0]$ | |
| z | log integer | $[3, 100]$ | fidelity |

Table 2: Synthetic Multi-Fidelity Hartmann search space in 3 dimensions.

| name | type | values | info |
|------|------|--------|------|
| X_0 | continuous | $[0.0, 1.0]$ | |
| X_1 | continuous | $[0.0, 1.0]$ | |
| X_2 | continuous | $[0.0, 1.0]$ | |
| X_3 | continuous | $[0.0, 1.0]$ | |
| X_4 | continuous | $[0.0, 1.0]$ | |
| X_5 | continuous | $[0.0, 1.0]$ | |
| z | log integer | $[3, 100]$ | fidelity |

Table 3: Synthetic Multi-Fidelity Hartmann search space in 6 dimensions.

The global minimums of these functions are known:

- Hartmann-3d: $f(x^*) = -3.86278$ at $x^* = (0.114614, 0.555649, 0.852547)$
- Hartmann-6d: $f(x^*) = -3.32237$ at $x^* = (0.20169, 0.150011, 0.476874, 0.275332, 0.311652, 0.6573)$

### D.1.2 JAHS-Bench-201

JAHS-Bench-201 [35] is a benchmark consisting of surrogates trained on 140 million data points of Neural Networks trained on 3 datasets, namely CIFAR10, Colorectal-Histology, and Fashion-MNIST. They extend the search space beyond the original tabular search space of NAS-Bench-201 [49] consisting of purely discrete architectural choices, introducing both hyperparameters and multiple fidelities to create the first multi-multi-fidelity benchmark for deep learning hyperparameter optimization (Table 4).

Each of the three datasets shares equal search spaces while we fix the fidelity parameters, depth `N` and width `W`, to their maximum. We further limit `Resolution` to a fixed value of $1.0$ out of the three

original choices $\{0.25, 0.5, 1.0\}$. The surrogates provided by JAHS-Bench-201 do not explicitly model the monotonic constraint that as `epoch` increase, so should the training cost. In practice, this was found to be insignificant but we state so for completeness. For these benchmarks, optimizers minimize `1 - valid_acc`.

| name | type | values | info |
|---|---|---|---|
| Activation | categorical | {ReLU,Hardswish,Mish} | |
| LearningRate | continuous | $[0.001, 1.0]$ | log |
| N | constant | 5 | |
| Op1 | categorical | {0,1,2,3,4} | |
| Op2 | categorical | {0,1,2,3,4} | |
| Op3 | categorical | {0,1,2,3,4} | |
| Op4 | categorical | {0,1,2,3,4} | |
| Op5 | categorical | {0,1,2,3,4} | |
| Op6 | categorical | {0,1,2,3,4} | |
| Optimizer | constant | SGD | |
| Resolution | constant | 1.0 | |
| TrivialAugment | categorical | {True,False} | |
| W | constant | 16 | |
| WeightDecay | continuous | $[1e\text{-}05, 0.01]$ | log |
| epoch | integer | $[3, 200]$ | fidelity |

Table 4: The JAHS-Bench-201 search space for all 3 datasets, CIFAR10, Colorectal-Histology and Fashion-MNIST.

### D.1.3 PD1 (HyperBO )

The PD1 benchmarks consist of surrogates trained on the learning curves of large architectures, spanning both natural language and computer vision tasks. The original tabular data is obtainable from the output generated by HyperBO [32] using the dataset and training setup of [50], enabling us to test our methods and baselines for low-budget settings, where multi-fidelity methods are most applicable. The hyperparameters considered for the optimization runs were for Nesterov Momentum [51] which constitutes our search space (Table 5-8). All other hyperparameters were fixed according to their training setup and provided data.

This tabular data consists of 4 collections of optimization records, a grid-like spread of configurations and also those chosen by their optimizer, in both an initial testing phase and a later full experiment phase. To maximize the data available to the surrogate, we utilize all of this data but take care to drop duplicated runs from their test runs. We select these 4 benchmarks out of the available 24, opting to have a variety of tasks, favoring larger models where possible, or tasks that use big batch sizes such as 2048.

For these benchmarks, each optimizer aims to minimize the `valid_error_rate`. The hyperparameters listed are based on the minimum and maximum values found within the original tabular data, rather than the reported ranges by the authors of HyperBO [32]. This was to prevent surrogates from being required to extrapolate outside of their training domain.

- **lm1b_transformer_2048** derives from the optimization trace of a transformer model [52] with batch size of 2048 on the `l1mb` statistical language modelling dataset [53].

- **translatewmt_xformer_64** derives from the optimization trace of an `xformer` [54] with batch size of 64 on the WMT15 German-English [55].

- **cifar100_wideresnet_2048** dervies from the optimization trace of a wideresnet model [56] with batch size 2048 on the cifar100 dataset [57].

- **imagenet_resnet_512** dervies from the optimization trace of a resnet model [58] with batch size of 512 on the imagenet dataset [59].

| name | type | values | info |
|---|---|---|---|
| lr_decay_factor | continuous | $[0.010543, 0.9885653]$ | |
| lr_initial | continuous | $[1e\text{-}05, 9.986256]$ | log |
| lr_power | continuous | $[0.100811, 1.999659]$ | |
| opt_momentum | continuous | $[5.9e\text{-}05, 0.9989986]$ | log |
| epoch | integer | $[1, 74]$ | fidelity |

Table 5: The `lm1b_transformer_2048` search space.

| name | type | values | info |
|---|---|---|---|
| lr_decay_factor | continuous | $[0.0100221257, 0.988565263]$ | |
| lr_initial | continuous | $[1.00276e\text{-}05, 9.8422475735]$ | log |
| lr_power | continuous | $[0.1004250993, 1.9985927056]$ | |
| opt_momentum | continuous | $[5.86114e\text{-}05, 0.9989999746]$ | log |
| epoch | integer | $[1, 19]$ | fidelity |

Table 6: The `translatewmt_xformer_64` search space.

**Training surrogates on the PD1 tabular data** The original data is a mix of several datasets, models, and their parameters for which we do some preprocessing. All data-preprocessing is available as part of `mf-prior-bench` and consists of:

1. Splitting the raw data by all available $\{datasetname, model, batchsize\}$ subsets.

2. Identify which columns are hyperparameters by those being marked as such and consist of more than one unique value.

3. Drop all columns which are not hyperparameters or metrics.

4. Drop all `NaN` values for which no metrics are recorded.

5. Drop configurations that recorded *divergent* training costs.

6. Drop duplicated entries, preferring to keep those from their full experimental runs.

To decide if a configuration diverged was to find outliers that reported extreme outlier costs, with a cutoff applied heuristically to each individual dataset, the details of which can be found within `mf-prior-bench` . This was done to ease the learning process of the surrogate model and to remove emphasis on these outliers. The resulting surrogate is no longer aware of these divergences and offers smooth interpolation for the training cost for these configurations.

Once the datasets are prepared, we then train a single surrogate XGBoost model [60] per metric recorded. This training was performed using DEHB [61], optimizing for the mean R2 loss of 5-fold cross-validation for a total of 4 hours, 8 CPU cores, and the seed set to 1. All surrogate models were found to converge in their R2 loss. These surrogates are available as part of `mf-prior-bench` for further inspection. While certainly improvements can be made in this modeling phase, for the purpose of our experiments, they offer a good approximation of the entire optimization landscape.

| name | type | values | info |
|---|---|---|---|
| lr_decay_factor | continuous | $[0.010093, 0.989012]$ | |
| lr_initial | continuous | $[1e\text{-}05, 9.779176]$ | log |
| lr_power | continuous | $[0.100708, 1.999376]$ | |
| opt_momentum | continuous | $[5.9e\text{-}05, 0.998993]$ | log |

Table 7: The `cifar100_wideresenet_2048` search space.

| name | type | values | info |
|------|------|--------|------|
| lr_decay_factor | continuous | $[0.010294, 0.989753]$ | |
| lr_initial | continuous | $[1e\text{-}05, 9.774312]$ | log |
| lr_power | continuous | $[0.100225, 1.999326]$ | |
| opt_momentum | continuous | $[5.9e\text{-}05, 0.998993]$ | log |

Table 8: The `imagenet_resnet_512` search space.

### D.1.4  Yahpo-Gym

The Yahpo-Gym [34] collection is a large collection of multi-fidelity surrogates across a wide range of tasks, including traditional machine learning models with dataset size as a fidelity, as well as Neural Network, benchmarks such as LCBench and NAS-Bench-301 [62].

Yahpo-Gym provides surrogates trained on a shared search space between all of these tasks. We ignore the rest of the available benchmarks from Yahpo-Gym as the others consist of non-DL tasks or contain conditional search spaces which are not suitable for most of our baselines. For these benchmarks, the optimization objective was to minimize `1 - val_balanced_accuracy`. Table 9 shows the search space for these 5 benchmarks.

| name | type | values | info |
|------|------|--------|------|
| batch_size | integer | $[16, 512]$ | log |
| learning_rate | continuous | $[0.0001, 0.1]$ | log |
| max_dropout | continuous | $[0.0, 1.0]$ | |
| max_units | integer | $[64, 1024]$ | log |
| momentum | continuous | $[0.1, 0.99]$ | |
| num_layers | integer | $[1, 5]$ | |
| weight_decay | continuous | $[1e\text{-}05, 0.1]$ | |
| epoch | integer | $[1, 52]$ | fidelity |

Table 9: The `lcbench` search space.

For our experiments, we choose 5 LCBench tasks from 34 from OpenML [63] according to the Spearman rank correlation of configurations at the $10\%$ epoch and the final epoch $z_{\max}$. The chosen tasks, 126026, 167190, 168330, 168910, and 189906 are equally spaced according to this correlation and include the task featuring the least (126026) and most (189906) correlation. The correlation for all LCBench tasks can be seen in Figure 9.

### D.1.5  Classifying benchmarks into high-low correlation

Our definition of a *high* or a *low* correlation benchmark is that the benchmark must have $0.8$ spearman correlation of rankings at $10\%$ of the maximum fidelity to be classified as *high*, otherwise it is classified as *low*. We depict this in Figure 10, showing the spearman correlation between each fidelity available and the final full fidelity. While other cutoffs and classifications are possible, given our suite of benchmarks, we find this to be a reasonable separation given the data.

To further motivate this choice, Figure 11 shows the performance of random search and HyperBand across these 12 benchmarks. We choose HyperBand as under the hood, it runs different instantiations of SH. Under a single worker-run, HB becomes a sequential run of SH with increasing $z_{\min}$. Strong performance of SH requires a high correlation of performance across fidelities. Thus in $5\times$ in Figure 11, wherever HB the performance gap between random search and hyperband is not pronounced in the short budget regime shown, it can be inferred that the lower fidelities do not provide reliable prediction when early stopping configurations. Based on the classification strategy derived from Figure 10, we denote high correlation benchmarks with a $(+)$ and low correlation benchmarks with a $(-)$ in Figure 11. The comparison of random search and HB in this figure reflects the reasonable correctness of our classification strategy.

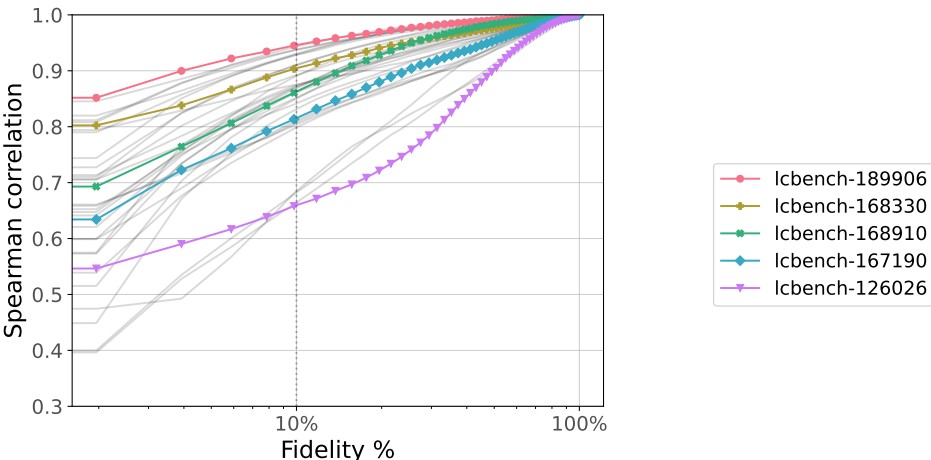

Figure 9: Each curve shows the spearman-correlation of 25 configurations from the given fidelity along the x-axis to the last fidelity, where the standard deviation is estimated with repeated samples until the mean curve converges to within a $0.001$ Euclidean distance update to the previous mean. The highlighted lines are the LCBench tasks selected by taking the $(0, 0.25, 0.5, 0.75, 1)$ quantiles of the correlations at $10\%$ of the full fidelity range. The faded gray lines with no markers represent LCBench tasks not selected.

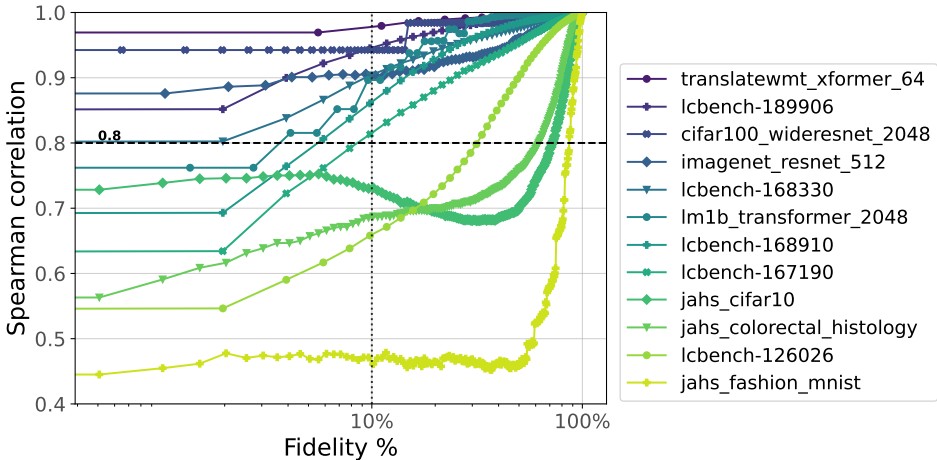

Figure 10: Each curve shows the spearman-correlation of 25 configurations from the given fidelity along the x-axis to the last fidelity, where the standard deviation is estimated with repeated samples until the mean curve converges to within a $0.001$ Euclidean distance update to the previous mean. Our good/bad correlation definition corresponds to a spearman-correlation cutoff of $0.8$ at $10\%$ of the maximum budget. The legend is sorted by their correlation at $10\%$ fidelity and corresponds to the order of benchmarks in Figure 11.

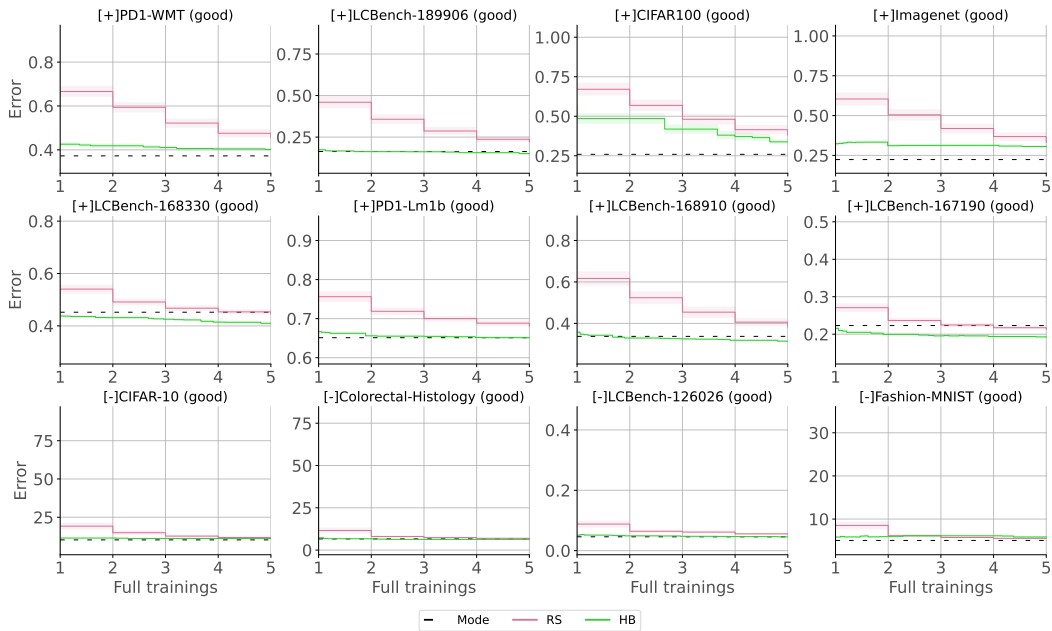

Figure 11: Comparing Random Search and HyperBand over $5\times$ budget to gauge quality of performance correlation across fidelities. We classify benchmarks marked with $[+]$ as *high correlation* benchmarks (Row 1 and 2). The benchmarks marked as $[-]$ are classified as *low correlation* benchmarks (Row 3).

### D.1.6 Issues with Benchmarks in parallel setting

During our parallel worker runs, we noticed that workers running on LCBench (see D.1.4) would silently drop out. This is an artifact of Yahpo-Gym , where the shared loading of system resources does not play nicely with workers being started in parallel. The optimization runs would still progress without issue but only utilize 1-3 processes instead of the deployed 4. The degree to which the problem occurred was minimal but likely to impact aggregated results. As a safety precaution, we remove LCBench from our parallel algorithm evaluations to prevent undue biases from leaking into our evaluations.

### D.2 Baselines

For fair comparison, customizability and certain technical constraints, we reimplement all the baselines listed below, other than BOHB.

**HyperBand** We implement our own version of HB[30] to allow for the input of priors. We verified our implementation with the popular HB implementation provided in BOHB [44]. We use $\eta = 3$ for all experiments with the minimum and maximum budget coming as an input from the problem to solve, in this case, benchmarks. As described in Section 3, HB iterates over different instantiations of SuccessiveHalving (SH). In Figure 12 we show an example[3] for an SH run under $\eta = 2$.

**Bayesian optimization** is a prominent framework for Hyperparameter Optimization [7, 64], hence we choose its Gaussian Processes (GP) implementation as a model-based competitor. We incorporate expert priors to the optimization following the $\pi$BO algorithm [28]. In a low-budget setting, model-based search proves challenging for high-dimensional search spaces (e.g. JAHS-Bench-201 ) as common practices require the number of initial random observations equal to the search space dimensionality. To allow model-based search in BO and $\pi$BO we set their initial design size to $10$, to allow model fitting under our experiment budgets. Expected Improvement is used as the acquisition function across all BO algorithms.

---

[3]image and caption sourced under CC-BY-4.0 from `https://www.automl.org/wp-content/uploads/2019/05/AutoML_Book.pdf`

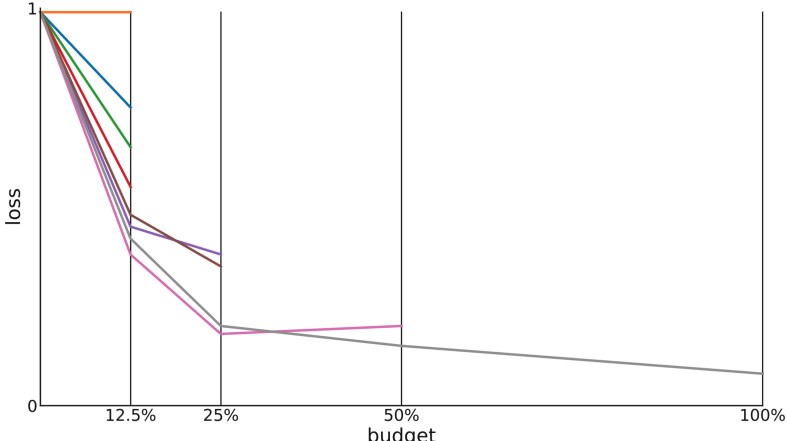

Figure 12: Illustration of `SuccessiveHalving` for eight algorithms/configurations. After evaluating all algorithms on $1/8$ of the total budget, half of them are dropped and the budget given to the remaining algorithms are doubled.

**BOHB** [44][4] incorporates multi-fidelity HB into the Bayesian optimization framework by building KDE models on each fidelity level to efficiently guide the search. The official implementation has no direct way of accepting a prior distribution over optimal configurations and incorporating it into search. We keep the other default settings intact.

**ASHA** [36] was developed as an extension to `SuccessiveHalving` that can run on massively parallel systems, designed to minimize idle workers. ASHA modified SH and HB to allow for a configuration to be promoted as soon as $\eta$ configurations have been seen at a fidelity, calling such a step an *asynchronous promotion*. Thereby, a free worker need not remain idle till an entire predefined number of configurations have finished evaluation at a fidelity, like vanilla-SH or HB. Since HB effectively runs multiple SH brackets, asynchronous-HB can be designed as HB running multiple ASHA brackets. However, for this work, we chose the sampling distribution for the brackets as used in Klein et al. [65] for asynchronous-HB.

**Mobster** [65] extended asynchronous-HB with a surrogate model to create an asynchronous version of BOHB. The asynchronous design aside, Mobster is different from BOHB as it uses Gaussian Processes as the surrogate model, unlike BOHB which uses Tree Parzen Estimators (TPE).

**ESP based baselines.**    We construct multiple baselines that do not explicitly exist in the literature but serve as important baselines for a fair comparison and a deeper understanding of the problem and the method.

- *[X]+Prior* methods: for such a method, the uniform random search component in X is replaced with sampling from the prior. If accompanied by a Y%, that indicates the percentage of prior-based sampling, with (1 - Y)% for random sampling. For example, HB+Prior is HB but with only sampling from prior and HB+Prior(30%) would indicate that sampling from prior happens with a probability of $0.3$ and random sampling with a probability of $0.7$.

- *[X]+ESP* methods: for such a method, the uniform random search component in X is replaced with sampling from the ESP, $\mathcal{E}_\pi$.

- `PriorBand+BO`: this runs `PriorBand` as described in Section 5 for a budget equivalent to an initial design of $10$, and then switching to sampling a configuration from the acquisition in a BO loop, disabling the ESP.

**Parallel runs.**    We run each benchmark-optimizer-seed combination over $4$ workers. For HB based algorithms, if a worker is free, an evaluation from the next SH bracket is already started, in order to maximize worker efficiency. However, a pending evaluation from the earliest active SH bracket

---

[4]Implementation from HpBandSter: `https://github.com/automl/HpBandSter`

has the highest priority. For BO algorithms, under our parallel setup for multi-fidelity optimization, we did not require batch acquisitions. However, we needed to account for incomplete evaluations from configurations that are still training when fitting the surrogate. We do this simply by fitting the surrogate on the finished evaluations and predicting the mean for the pending evaluations, before retraining on the required set of configurations[5].

### D.2.1 GP kernels

**Numerical kernel.** In the numerical continuous and discrete domain, we use the standard ([7, 66]) *Matérn*-5/2 kernel [67]. For $D$-dimensional numerical inputs $x$ and $x'$,

$$k_{M5/2}(x, x') = \theta_0 \left( 1 + \sqrt{5r^2(x, x')} + \frac{5}{3}r^2(x, x') \right) \exp\left( -\sqrt{5r^2(x, x')} \right), \quad (4)$$

where $r^2(x, x') = \sum_{d=1}^{D} (x_d - x'_d)^2 / \theta_d^2$ denotes a scaled Euclidean distance between points, and $\theta_d$ is a dimension-specific lengthscale.

**Categorical kernel.** A straightforward extension of a Matérn kernel for a categorical domain is to use *1-in-K encoding*. However, this solution increases the dimensionality of the input, which might result in poor regression performance. Instead, we follow [68] who propose direct handling of categorical inputs by computing a weighted Hamming distance:

$$k_{HM}(x, x') = \exp\left( \sum_{d=1}^{D} \left( -\theta_d \cdot \mathbb{1}_{x_d \neq x'_d} \right) \right) \quad (5)$$

### D.3 Generating priors

We borrow the prior generation procedure in Hvarfner et al. [28] to generate the *near optimum* and *bad* priors. Additionally, we construct another class of *good* priors to reflect a different strategy that may be closer to practice, since for DL, the optimum is generally not known. We thus generate three kinds of priors, *near optimum*, *bad*, and *good*, the first two to simulate boundary conditions of priors that may be received and *good* to simulate a practitioner with some prior knowledge of a good configuration to choose. In each of these cases, the prior distribution is a normal $\mathcal{N}(\boldsymbol{\lambda}, \sigma^2)$, with $\sigma = 0.25$, where the configuration $\boldsymbol{\lambda}$ is generated by the following processes:

**Near optimum:** Given we don't know the optimum configuration for a benchmark, we approximate this by taking the best of $50,000$ configurations according to their observed loss value. During each seeded run, we perturb this configuration by Gaussian noise with a $\sigma = 0.25$ for numerical values, while using this same $0.25$ for uniformly selecting a different categorical value. This is done to ensure there is still some room for improvement possible over the initial prior configuration. All algorithms receive the same prior configuration as defined by the seed. In the case of the Multi Fidelity Hartmann benchmarks, we can access the optimum of the function analytically and so we take this optimum as the configuration which is to be perturbed.

**Good:** We define a *good* prior as one that is suggested by limited prior evaluations done by a practitioner. For this, we take the best of 25 configurations and apply no further noise modifications per seed. This means each run will see the same prior irrespective of the prior. We choose a budget of 25 random samples since in our experimental setup our focus is strictly on HPO budgets under $20\times$. As we show later, the relative quality of this class of priors varies per benchmark, compared to the near-optimum priors.

**Bad:** As we don't know the worst configuration for each benchmark, we simulate this process by taking the worst configuration of $50,000$ samples. No additional noise is added per seed, to ensure we always treat the worst known configuration as the prior input.

It is worth noting that while *near optimum* and *bad* are the best and worst of $50,000$ samples, respectively, it need not be that these configurations are also the best and worst for search. For benchmarks with a high categorical count, such as JAHS Bench as described in D.1.2 (middle row

---

[5]unlike typical BO, multi-fidelity BO may have different data subsets to fit the surrogate over

in Figure 13), simply switching a categorical value can drastically alter the performance of the configuration. More generally, a single point of the search space does not immediately provide information about the performance in its locality. As such, bad performing configurations could be reached with the noise perturbations of the *near optimum* prior.

To investigate the performance impacts of each prior, in Figure 13 we plot the distribution of performances of 25 configurations sampled from the prior distribution used. This is additionally done for 50 seeds, giving a total of $1,250$ configurations plotted per violin.

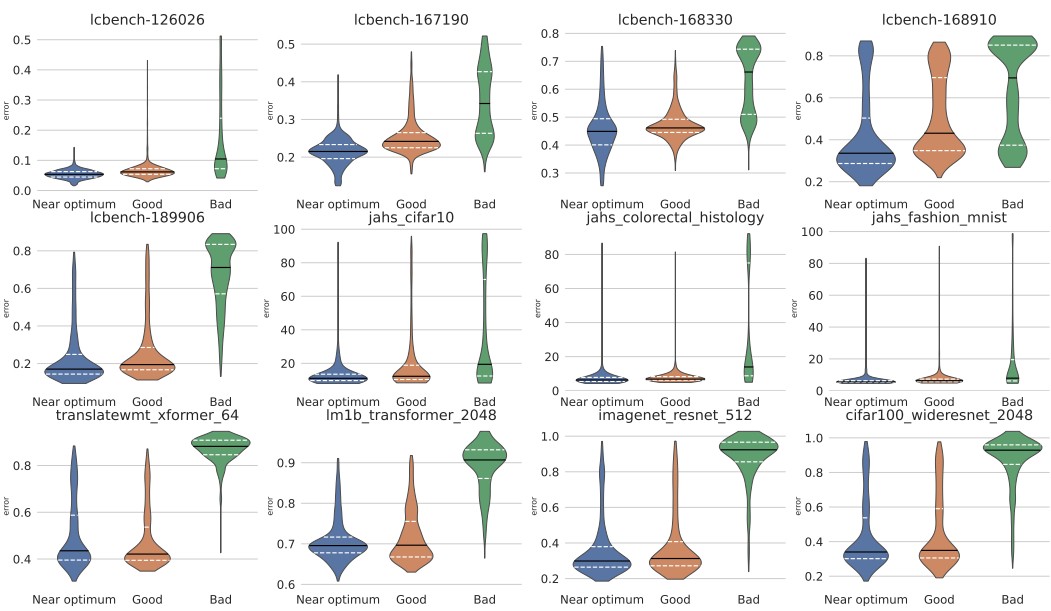

Figure 13: Each violin shows the density of performances achieved by random search on a distribution using the given prior. The black line in the violin is the mean with the dashed white lines indicating the 25% and 75% quartiles. In all cases, we see *near optimum* and *good* outperform the *bad* prior, as expected. In the case of the PD1 datasets (bottom row), some *good* priors achieve a better mean error than *near optimum* but their best configurations found are not as strong as they are when using *near optimum* priors. This could be a result of the near optimum prior being perturbed each run. Some iterations may put the prior at a slightly worse region than the fixed good prior. However, the near-optimum priors generally achieve the best-seen score.

### D.4 Experimental setup

Generally, the HPO landscape for a new task is unknown and the quality of the expert prior input cannot be gauged without previous run data available. In order to simulate how an expert may use HPO in their task, we design and present the experiments to represent such scenarios. We show the aggregated results (*All*) over different qualities of prior input to highlight robustness across prior qualities, which may be unknown at the beginning of the problem. Subsequently, we break the results down into performances under the *near optimum* priors, *good* prior, and *bad* prior. This setup intends to highlight that `PriorBand`'s performance benefits are more substantial if the quality of the prior is better. Our experiment design substantiates the hypothesis that `PriorBand` and algorithms extended with $\mathcal{E}_\pi$ are robust across any given prior input, a property not held by our baseline algorithms.

The metrics reported for single-worker runs are for 50 seeds across 12 benchmarks, 5 from LCBench (D.1.4), 3 from JAHS-Bench-201 (D.1.2) and 4 from PD1 (D.1.3). For the parallel case, we run 4 workers for each algorithm, 10 seeds, for only 7 [6] of these benchmarks. We chose 10 seeds due to the increased cost of running many workers[7].

---

[6]We exclude 5 LCBench benchmarks due to parallelism issues as described in D.1.6

[7]asynchrony is simulated for the queries to the benchmarks by sleeping for *epoch* seconds

## D.5 Evaluation protocol

For the reduction factor $\eta$, in HB and thereby `PriorBand`, we chose $\eta = 3$. For the various $[z_{\min}, z_{\max}]$ coming from the 12 benchmarks chosen, there are 2, 3, or 4 levels of fidelity created (rungs) in HB for our set of experiments. Therefore, for the single worker case, we show a budget of 12 full function evaluations, which is approximately equivalent to the budget of one complete HB iteration with 4 fidelity levels ($12 \times z_{\max}$). For other cases, we show a budget of $20\times$ and report under $5\times$ for multi-worker runs.

The evaluation setup is motivated by how an HPO algorithm might be used in practice. The HPO algorithm stopped anytime, will return the best-seen configuration, i.e., will return the current incumbent. To compare algorithms, we plot the incumbent over the budget spent in epochs. The incumbent is simply chosen as the configuration with the lowest error across all fidelities. The average relative rank plots are computed over the validation error of the incumbent at $z_{\max}$, as obtained from the benchmark. For each seed, we obtain the ranks of algorithms for each benchmark, averaging them to get an estimate of each algorithm's robustness across tasks. By computing the mean and standard error across seeds, we obtain an expectation of the algorithm's performance with respect to its stochastic components and other variations derived from seeding. We show all relative ranks only after the $1\times$ budget has been exhausted.

We compute ranks over the error or loss on the validation sets, where the choice of benchmark determines the exact metric to be minimized. Given we aggregate over the average ranks on a benchmark, the choice of per benchmark metric is irrelevant. For all prior-based baselines, we evaluate the mode of the prior distribution at $z_{\max}$ as the first evaluation.

# E Algorithm details

## E.1 Pseudocode

Algorithm 3 is the main HB loop and the optimizer interface for `PriorBand`. Depending on the actual framework where this is implemented, Lines 8-9 may be arranged differently. L8 is an asynchronous call that allows HB and by extensions `PriorBand` to be parallel since different SH brackets can be scheduled simultaneously. That is, it can begin a new SH bracket even if L8 has not returned the sampled and evaluated base rung of the current SH bracket. L4 in Algorithm 4 can also be an asynchronous call, allowing multiple configurations from L3 to be evaluated in parallel. The pseudo-code for HB (Algorithm 3) thus presents itself in a way where the ensemble sampling policy $\mathcal{E}_\pi$ replaces vanilla random search, for the lowest fidelity (base rung) in the current SH bracket in HB (L3 in Algorithm 4). Once the samples are collected and evaluated, vanilla-SH's promotion can be run normally to survive and continue training chosen configurations.

---

**Algorithm 3** HB base for `PriorBand`

---

1: **Input:** Distribution over optimum $\pi(\cdot)$, halving parameter $\eta$, resource bounds $[z_{\min}, z_{\max}]$.
2: $s_{\max} \leftarrow \lfloor \log_\eta(z_{\max}/z_{\min}) \rfloor$
3: $\mathcal{H} \leftarrow \emptyset$
4: **while** $budget\ remains$ **do**
5:     **for** $s \in \{s_{\max}, \ldots, 0\}$ **do**
6:         $n \leftarrow \left\lceil \frac{(s_{\max}+1)}{s+1} \cdot \eta^s \right\rceil, \quad \boldsymbol{z} \leftarrow z_{\max} \cdot \eta^{-s}$
7:         $r = s_{\max} - s$
8:         $\mathcal{H} \leftarrow$ Alg. 4$(s, s_{\max}, \eta, n, \boldsymbol{z}, \pi, \mathcal{H})$         ▷ sample and evaluate $n$ times
9:         $\mathcal{H} \leftarrow$ Alg. 5$(s, s_{\max}, \eta, \boldsymbol{z}, \mathcal{H})$         ▷ run SH-based promotions
10:     **end for**
11: **end while**

---

---

**Algorithm 4** Sample and evaluate $n$ configurations

---

1: **Input:** $s, s_{\max}, \eta$, number of samples $n$, evaluation resource $z$, prior $\pi(\cdot)$, observations $\mathcal{H}$
2: **for** $i \in \{1, 2, \ldots, n\}$ **do**
3:     $\boldsymbol{\lambda} \leftarrow$ Alg. $1(s, s_{\max}, \eta, \mathcal{H}, \pi(\cdot))$          $\triangleright$ sampling from $\mathcal{U}(\cdot)$ here runs vanilla-HB
4:     $y \leftarrow evaluate(\boldsymbol{\lambda}, z)$
5:     $\mathcal{S} \leftarrow \mathcal{S} \cup \{(\boldsymbol{\lambda}, y, z)\}$
6: **end for**
7: **return** $\mathcal{S}$

---

---

**Algorithm 5** Perform a SH iteration given the base rung

---

1: **Input:** $s, s_{\max}, \eta$, minimum evaluation resource $z, \mathcal{H}$
2: $rung = s_{\max} - s$
3: $(\hat{X}, \hat{Y}) \leftarrow$ retrieve all observations from $r$ in $\mathcal{H}$
4: **for** $\hat{s} \in \{s - 1, \ldots, 0\}$ **do**
5:     $z = \eta \cdot z$                                       $\triangleright$ the next higher fidelity
6:     $\hat{X} \leftarrow top\_1/\eta(\hat{X}, \hat{Y})$          $\triangleright$ the top-performing $(1/\eta)$ configurations in $\hat{X}$
7:     $\hat{y} \leftarrow evaluate(\boldsymbol{\lambda}, z), \forall \boldsymbol{\lambda} \in \hat{X}$
8:     $\mathcal{S} \leftarrow \mathcal{S} \cup \{(\hat{X}, \hat{Y}, z)\}$
9: **end for**
10: **return** $\mathcal{S}$

---

## E.2 Ablations

This section contains ablations over the various design choices in `PriorBand` (Section 5.2). All ablations are compared over different qualities of prior strength. Additionally, the set of benchmarks is also grouped into combinations of high-low correlation of lower fidelity to the $z_{\max}$ and good-bad priors as discussed in Appendix D.1.5 and D.3. In Sections E.2.1-E.2.3 the next 3 sections we show ablations over different choices for the main design components in ESP for `PriorBand`. In Appendix E.2.4 we show the benefits of incumbent sampling in `PriorBand`.

### E.2.1 Random sampling proportions

Section 5.2.1 describes how the proportion of random samples is traded-off with the proportion of prior sampling across fidelities. In Figure 14 we compare 2 other designs for how the proportion of random and prior samples can be determined. `PriorBand`($50\%$) uses a fixed heuristic where $50\%$ of the samples at each rung is going to be a random sample. `PriorBand`(linear) is similar to `PriorBand`, where instead of a geometric decay (Section 5.2.1) of random sample proportions like the latter, a linear decay is applied. That is, at the lowest rung, $p_\pi = p_\mathcal{U}$, and at the highest rung, $p_\pi = \eta \cdot p_\mathcal{U}$. For all intermediate rungs, the relationship is derived from this interpolated line.

### E.2.2 Prior and incumbent-based sampling trade-off

The choice of incumbent-prior trade-off strategy (Section 5.2.2) is crucial to the robustness of `PriorBand`. As the simplest design, we kept the proportion of incumbent sampling to prior sampling a fixed constant in `PriorBand`(constant), $p_\pi = \eta \cdot p_{\hat{\lambda}}$. `PriorBand`(decay)[8] in Figure 15 was designed such that the prior is decayed over time, similar to $\pi$BO. After every SH bracket, the proportion of prior samples with respect to the incumbent proportion was halved. As Figure 15 highlights, the data-driven likelihood score-based method described in Section 5.2.2 is essential for robust performance across all cases. This further supports our justification for the need for a non-naive method to achieve robustness, in Section 4.

### E.2.3 Choice of incumbent-based sampling

In this section, we look at different methods of defining an incumbent-based sampler. The role of the incumbent sampler is to allow exploitation by performing a local search around the best-seen

---

[8]this `PriorBand`(decay) is different from the decay version in Figure 14 in Appendix E.2.1

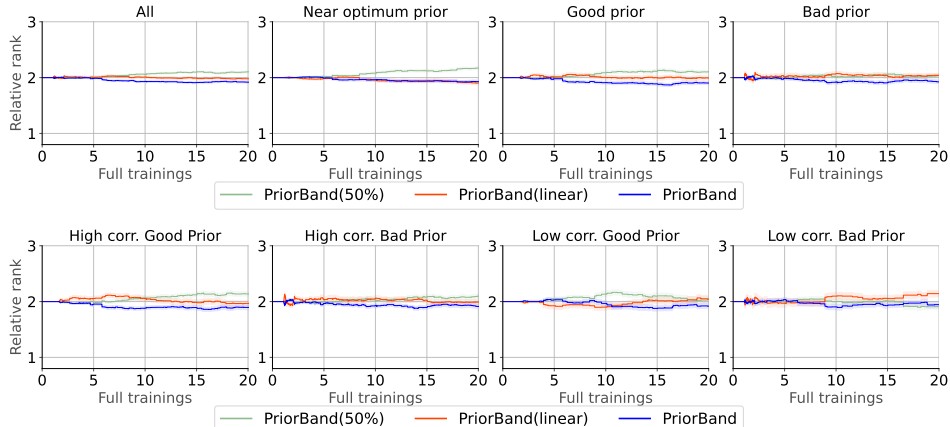

Figure 14: A comparison of `PriorBand` with 3 different strategies for setting the proportion of random samples in ESP as a fixed function of the rung. PriorBand(50%) fails to meaningfully utilize the *near optimum* and *good* prior when compared to both linear and geometric decay, with no aggregated benefit seen at any budget. PriorBand(linear) is an equivalent heuristic to PriorBand(linear) in intuition and simplicity, but overall `PriorBand`'s geometric decay comparatively performs robustly across all scenarios.

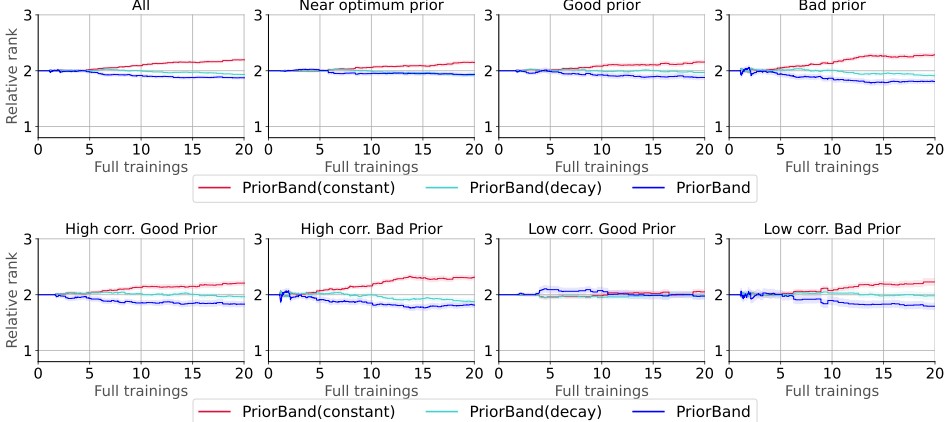

Figure 15: A comparison of `PriorBand` with 3 different strategies for trading off between incumbent sampling and prior distribution sampling. `PriorBand`'s default of using likelihood scores for weighting shows a dominant performance in almost all scenarios. The one exception is in low correlation settings with a good prior where all versions seem to suffer under low correlations. However, under the same set of tasks, a bad prior induces a marked difference in performance. In general, the plot highlights that each variant reacts differently to bad priors. The adaptive nature of `PriorBand` clearly is robust to different scenarios comparatively.

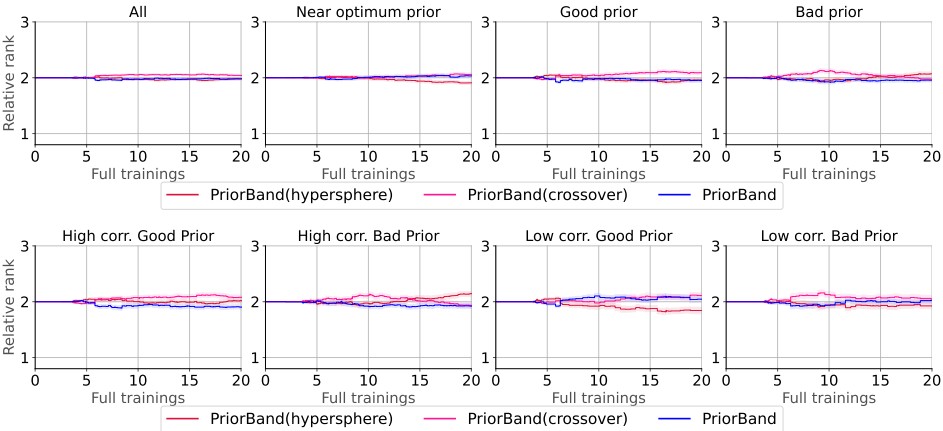

Figure 16: A comparison of 3 different methods in `PriorBand` to sample based on the incumbent configuration. No single method dominates in terms of performance. The hypersphere sampling method shows the most variation across scenarios which is undesirable for robustness while uniform crossover requires a less intuitive and complex operation. `PriorBand` uses local perturbation which is more intuitive, a much simpler operation, and shows similar performance to the crossover variant.

configuration. `PriorBand`(hypersphere) does so by sampling uniformly from a hypersphere around the incumbent, of a radius equivalent to the distance of the incumbent to its nearest neighbor. This method is intuitive, however, in practice, numerical issues may arise if the incumbent and its nearest neighbor are close to each other. This can make the sampling procedure extremely slow as the radius will keep shrinking, the more local search is performed. Moreover, the distance measure for different hyperparameter types can be an extra design choice. `PriorBand`(crossover) leverages the likelihood scores computed in Section 5.2.2 to choose if a random sample or a sample from the prior should participate in a simple uniform crossover with the incumbent. Though this alleviates the computation issue of the *hypersphere* method, it adds more complexity, is more difficult to interpret, and may not be as exploitative as desired of local search. Our final choice described in Section 5.1 is not only simple, fast, and easy to implement but also shows competitive performance with the other methods, as shown in Figure 16.

We additionally ablate over the chosen local mutation operation hyperparameters in Figure 17 and Figure 18.

### E.2.4 Importance of incumbent-based sampling

In order to demonstrate the role of incumbent-based sampling in `PriorBand`'s design, Figure 19 shows a version of `PriorBand` where $p_{\hat{\lambda}}$ is set to 0 with $p_\pi$ inheriting all its probability (`PriorBand`(w/o inc)). Thereby, switching off the incumbent sampling, while keeping all other designs intact. We see that for the chosen geometric trade-off between prior and random sampling (Section 5.2.1, Appendix E.2.1), the inclusion of incumbent-based local search is essential.

### E.2.5 Hyperparameters of incumbent-based sampling

For the incumbent-based sampling procedure described in Section 5.1, we employ a local perturbation around the incumbent using sampling from a Gaussian, $\mathcal{N}(\hat{\lambda}, \sigma^2)$. Here, we assume a fixed standard deviation (with $\sigma = 0.25$) for the distribution. However, to make the perturbation more local, especially in higher dimensional spaces, we randomly choose if an HP will be perturbed or not, with a fixed probability of $p = 0.5$, an unbiased coin toss.

For the discrete hyperparameters, given $k$ categorical choices $C = \{c_1, c_2, \ldots, c_k\}$, we use a discrete distribution to perturb the category in the incumbent configuration. We give a weight of $k$ to the current incumbent's categorical choice $c_j$ and weight of 1 to each other choice, giving us sampling probabilities of $p(c_j) = \frac{k}{2k-1}$ and $p(c_i) = \frac{1}{2k-1}, \forall c_i \in C, c_i \neq c_j$. This sampling procedure is hyperparameter free.

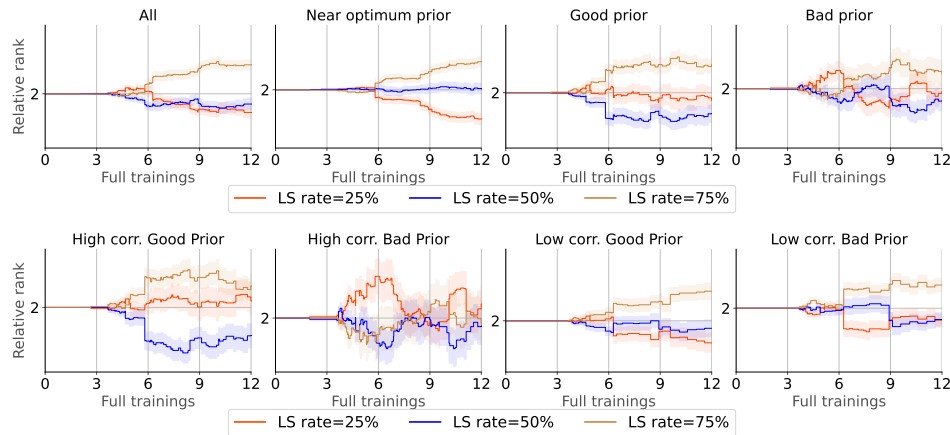

Figure 17: An ablation study over the hyperparameter $p$ in the incumbent-based sampling in `PriorBand` which chooses the probability of selection of an HP to be perturbed for search. `PriorBand` chose $p = 0.5$ which corresponds to $50\%$ in the figure. This setting appears to be never the worst method. A local search (LS) rate of $25\%$ is exploitative and can yield better performance under certain scenarios. However, our choice of $50\%$ was chosen to be generally balanced. A high rate of $75\%$ is too explorative and does not thus utilize the gains that the incumbent-based search can provide.

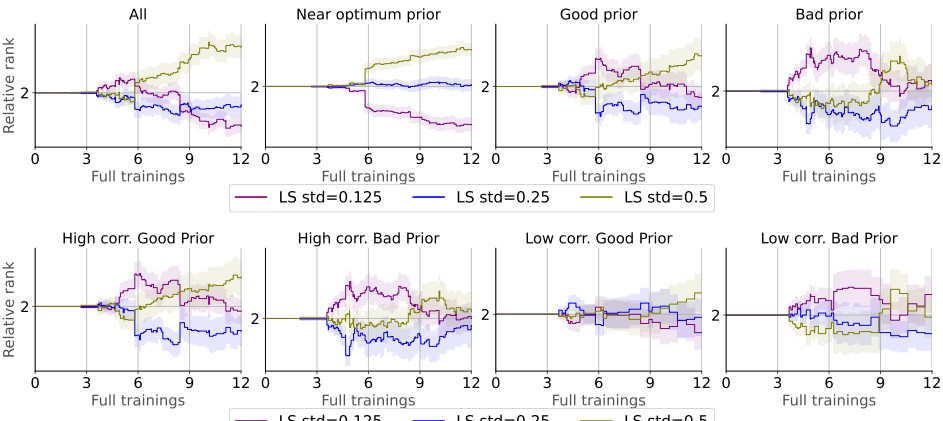

Figure 18: An ablation study over the hyperparameter $\sigma$ in the incumbent-based sampling in `PriorBand` which chooses the standard deviation of the Gaussian that will be centered around the HP to be perturbed for local search (LS). `PriorBand` chose $\sigma = 0.25$. This setting appears to be never the worst method. As expected, a peaker distribution under $\sigma = 0.125$ leads to more exploitation and thus strong performance under near-optimum priors. However, the quality of an incumbent improves over time. Being too exploitative, too early, could hurt optimization which is seen under the varying quality of good priors. The broad distribution under $\sigma = 0.5$ is much more explorative and is thus relatively worst under good priors and competitive under bad priors.

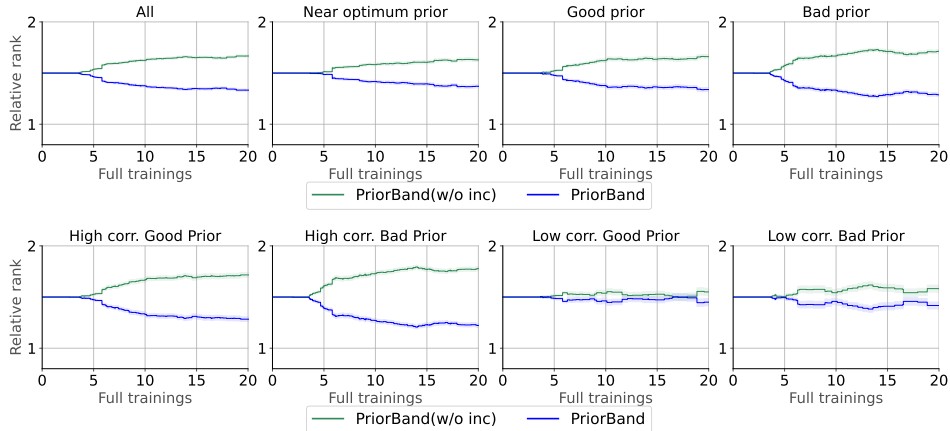

Figure 19: The top row shows different qualities of priors with the leftmost subplot being their aggregation. The bottom row shows different combinations of low and high correlation across fidelities in both the good and bad prior setting. The plot evidently shows the strength of incumbent-based sampling in all presented scenarios. The top row demonstrates how incumbent-based sampling helps escape the prior, which has less effect in the near optimum setting but a much stronger effect in the bad prior setting, where incumbent-based sampling is essential. In the bottom row, we see that incumbent-based sampling is more important in high correlation settings, where incumbents in low fidelities are likely to carry to later fidelities. The low correlation setting contrasts this effect, where spending the budget on configurations near the incumbent at low fidelities is not likely to translate to high-performing configurations at the highest fidelity.

The goal of `PriorBand` is to adapt to use good priors but recover from bad priors. To establish this through different experiments on different task settings and scenarios, we reduce confounding factors, keeping the aforementioned design for local search fixed.

Instead, hyperparameters could dynamically be adjusted to control `PriorBand`'s behavior, for example:

- A schedule to reduce the standard deviation of the Gaussian to sample the perturbation noise.
- Dynamically adjusting the probability $p$ for HP perturbation as optimization proceeds, potentially yielding better exploitation.
- Expert prior insight into HPO landscapes can allow more custom setting of the standard deviation for the Gaussian distribution.
- Expert knowledge of interaction effect or the importance of the hyperparameters in the search space could also allow for a tuned setting of $p$ for the selection of hyperparameters to perturb.

There are many possibilities for design, however, as our goal was to design the simplest approach that fulfills our desiderata, we choose a reasonable fixed setting and show it to be a robust choice in our experiments.

### E.2.6  Role of sampling the prior mode

Comparing `PriorBand`, where the prior mode is sampled at the maximum fidelity, with a version of `PriorBand` where the prior mode is sampled at the minimum fidelity (Mode@min), and not sampling the mode at all (No-Mode), in Figure 20. It appears that sampling the mode, in the beginning, provides huge gains especially if the prior is of good quality. In the case when the prior is not informative, `PriorBand` shows it can recover well even then. Given that most practical settings have multiple workers available, the fact that `PriorBand` can recover rapidly from misleading priors, and ultimately the utility of a good expert prior, it is reasonable to simply evaluate the expert default as the first evaluation. Under a multi-worker setup, the cost of this evaluation is amortized with the cheaper evaluations proceeding with the other workers. Optionally, the choice of whether the prior mode

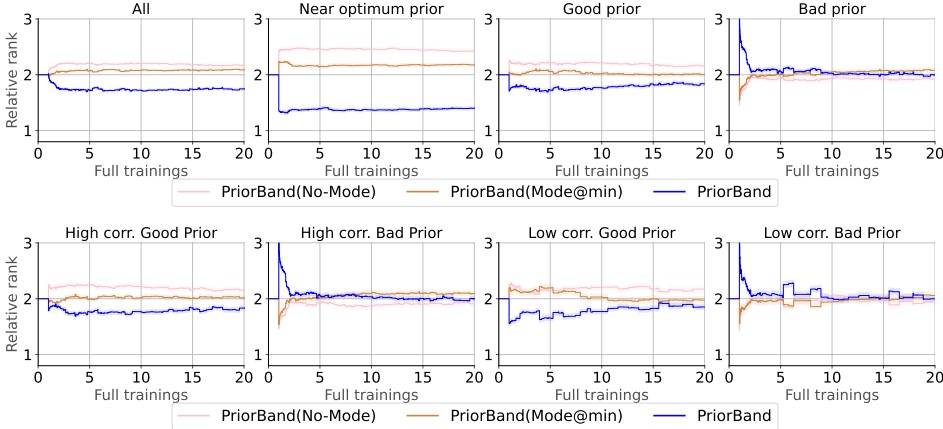

Figure 20: We show 3 possible methods of how to take the first evaluation. `PriorBand` by default choosing to sample the prior's mode at the maximum fidelity, with (Mode@min) doing so at the minimum fidelity and (No-Mode) simply beginning with a random sample at the lowest fidelity. The most prominent failure case of `PriorBand` happens in the bad prior settings, as no exploration occurs until after 1 full training worth of budget is exhausted. However, the algorithm still recovers rapidly.

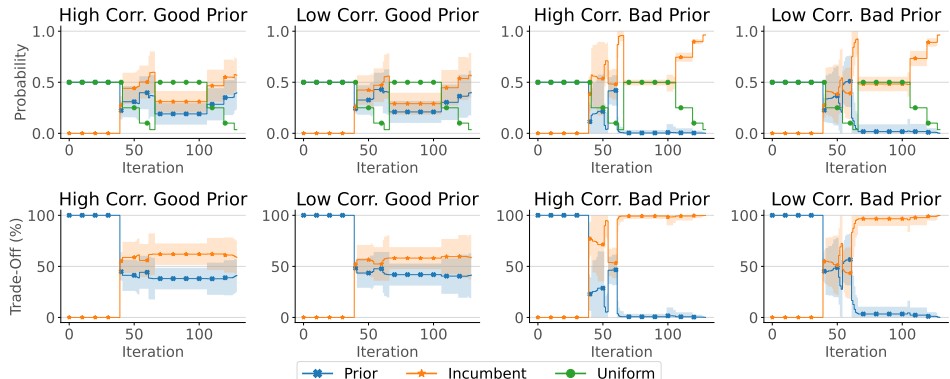

Figure 21: Evolution of the Ensemble Sampling policy $\mathcal{E}_\pi$ and its probabilities $p_\mathcal{U}$, $p_\pi$ and $p_{\hat{\lambda}}$ for runs on the multi-fidelity Hartmann-3 benchmark, for all combinations of high/low fidelity correlation and good/bad prior strength. The x-axis counts the number of multi-fidelity evaluations made by HB. [**Top**] The y-axis shows the exact probability assigned to each sampling strategy, $p_\mathcal{U}$, $p_\pi$, and $p_{\hat{\lambda}}$. Uniform sampling follows a repeating pattern as a function of fidelity. Incumbent sampling remains inactive at the beginning, but once activated, it dynamically adjusts depending on the quality of the prior given. [**Bottom**] We visualize the dynamic trade-off between prior and incumbent-based sampling, showing $p_{\hat{\lambda}}$ and $p_\pi$ as a percentage of $p_{\hat{\lambda}} + p_\pi$ on the y-axis. Under good priors, the probability of sampling from around the incumbent and sampling from priors is almost similar. Whereas under bad priors, the prior is discarded almost completely after one complete round of HB. Differences across high-low correlation setups exist, in the period between activation of the incumbent and one complete iteration of HB, but values stabilize after this iteration for all cases.

should be evaluated as the first evaluation or not can be toggled. For example, when the expert is confident of a good prior configuration and has knowledge of its performance from previous runs.

### E.3 Post-hoc view of `PriorBand` for interpretability

A DL expert can often provide a prior $\pi(\cdot)$ but is often unsure as to its merits. By tracking the evolution of `PriorBand`'s sampling probabilities during the optimization process, we can get an indicator of the strength of the $\pi(\cdot)$. The more useful the prior is for performance, the higher probability of sampling from $\pi(\cdot)$ and thus a higher $p_\pi$. We illustrate this in Figure 21, where we plot

the sampling probabilities $p_{\mathcal{U}}$, $p_{\pi}$ and $p_{\hat{\lambda}}$ of `PriorBand` during an HPO run with 50 seeds on the synthetic multi-fidelity Hartmann-3 function (Appendix D.1.1). We clearly see that our motivation for the design choice of `PriorBand` is well supported as the probability of sampling from the incumbent is not suppressing sampling from prior under good priors, but are aggressively affecting the chance of sampling from bad priors. Such a post-hoc analysis offers a DL expert insights into their own prior knowledge, allowing them to update or re-enforce their beliefs about what a good prior is for the problem at hand.

### E.4 Model extensions

In this section, we elucidate the model-based components when extending algorithms with $\mathcal{E}_{\pi}$ as shown in Figure 7. Firstly, we briefly explain BOHB and its modeling choice. BOHB subsumes the hyperparameter of the initial design size by setting it to $N_{dim} + 2$ where $N_{dim}$ is the dimensionality of the search space of a task. To activate model-based search, BOHB uses the following criteria: find the highest fidelity level with at least $N_{dim} + 2$ evaluated observations. If no such fidelity level exists, continue with uniform random sampling. If such fidelity exists, use all the observations at that fidelity to build a TPE as the surrogate. Since a model is built at a fidelity level, the fidelity variable is not part of the feature set for the surrogate. During acquisition, EI is used to obtain a configuration that approximately maximizes the TPE surrogate.

$$\text{EI}_z(\boldsymbol{x}|\mathcal{D}) = \mathbb{E}[\max\{f_z(\boldsymbol{x}) - y_z^{min}, 0\}], \tag{6}$$

where $y_z^{min}$ is the best score seen at the fidelity $z$. Given $f$ is a model built at a fidelity level $z$, the EI acquisition estimates the improvement of the suggested configuration at fidelity $z$. When the number of observations at $z_{\max}$ is $N_{dim} + 2$, the EI acquisition performs similarly to vanilla-BO. Mobster or asynchronous-BOHB follows the exact BO loop as BOHB, except that it uses a GP instead of a TPE and uses asynchronous HB for scheduling and not vanilla-HB.

**Model extension to `PriorBand`.** In our experiments to extend `PriorBand` with a model in `PriorBand+BO`, the above procedure of automatically switching to model-search lead to $\mathcal{E}_{\pi}$ not taking action and affecting optimization as we desire.

Similar to the initial design size in BO, incumbent-based sampling has an 'activation criteria' of one full SH bracket being evaluated, and at least one configuration evaluated at $z_{\max}$, after which incumbent-based search begins. We could default to BOHB's 'activation criteria' but for certain problems, the number of observations would satisfy the $N_{dim} + 2$ criteria even before the 1st SH bracket is over. This implies that incumbent-based sampling, a crucial component of $\mathcal{E}_{\pi}$, is never activated. Hence, we follow an approach that is more in line with BO and $\pi$BO in which 10 function evaluations are used as the initial design budget before switching to model search. This is most evident in Figure 7, where $\pi$BO behaves identically to only performing Random Search on the prior distribution, diverging at 10 full function evaluations.

We treat $10\times$ as $(10 \cdot z_{\max})$ the total budget (in epochs) exhausted during multi-fidelity optimization. After which, a GP model is activated for search that models *all* the observations made during the optimization, across any fidelity available. That is, the fidelity is an extra dimension modeled along with the search space. During the acquisition, since it is known from the optimization state $s_t$ which fidelity $z$ the new sample will be evaluated at, a 2-step optimization is performed when maximizing EI. In the first step, a set of configurations (10 in our experiments) is extracted for fidelity $z$ through Monte Carlo estimates of Equation 6 returning configurations likely to improve over the best configuration found so far at $z$. Following this, the EI score is calculated for this selected set of configurations using Equation 6 but with $z = z_{\max}$. At this stage, $y^{min}$ is chosen to be the best score obtained across all observations. The idea is to choose a configuration that is likely to maximize performance at the fidelity level where it is being queried and is also likely to improve at the target fidelity $z_{\max}$.

**Contextualizing model-search under $\mathcal{E}_{\pi}$.** During the typical initial design phase of `PriorBand+BO`, it is `PriorBand` that runs with the defined $\mathcal{E}_{\pi}$ comprising of $\mathcal{U}(\cdot)$, $\pi(\cdot)$, $\hat{\lambda}(\cdot)$ as actions. At the end of the initial design phase of `PriorBand+BO`, the action set $\mathcal{A}$ updates from { $\pi(\cdot), \mathcal{U}(\cdot), \hat{\lambda}(\cdot)$ } to { $\pi(\cdot), \mathcal{U}(\cdot), \hat{\lambda}(\cdot), \mathcal{M}(\cdot)$ } with new weights as $p_{\pi} = p_{\mathcal{U}} = p_{\hat{\lambda}} = 0$ and $p_{\mathcal{M}} = 1$, where $\mathcal{M}$ denotes model-based search.

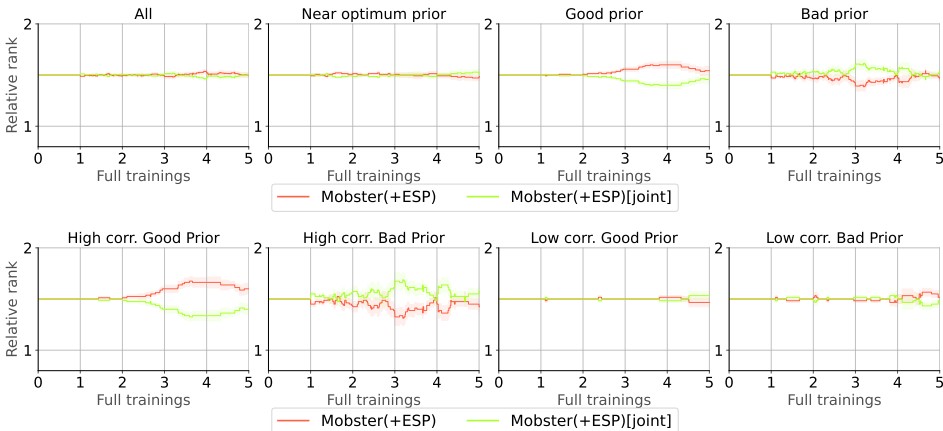

Figure 22: Comparing multi-fidelity modelling on asynchronous-HB per *rung* and *jointly* over the search space and fidelity ([j]). (Top) Compares the performances over different qualities of priors. (Bottom) Compares the good (at25) and bad prior cases, under benchmarks grouped into good-bad correlation. All algorithms were run for a total of $20\times$ over $4$ workers.

**Comparing multi-fidelity modeling.** In the previous section we motivate criterion and acquisition that allows $\mathcal{E}_\pi$ to influence search and the initial design space. For `PriorBand`, we note that the incumbent activation criteria are not fulfilled if modeling per fidelity with dimensionality as a criterion, as is done in BOHB and Mobster. In contrast, asynchronous-HB samples at random fidelities instead of the lowest fidelity first. When used with the ESP this allows the possibility of incumbent-based samples being activated before the model search begins. Hence, for asynchronous HB we can apply per-fidelity modeling. In Figure 22, we thus compare the 2 types of modeling discussed above: per-fidelity (Mobster+E) and joint modeling like `PriorBand`+BO (Mobster+E[joint]). We conclude that there is no significant difference in the performance of the two modeling choices. Though the joint model seems to perform slightly better under good priors.

# F More on experiments

This section gives a detailed experiment analysis that expounds on the results from Section 7.1-7.3. We compare similar setups but include our constructed prior-based baselines, over a different grouping of benchmarks with high and low correlations (Appendix F.1, F.2, F.3). In Appendix F.4 we compare the prior-based algorithms with each other. We further show the final validation performance tables of algorithms for the different experiments for every benchmark-prior combination, over 2 budget horizons in Appendix F.5.

## F.1 Robustness of `PriorBand`

More supporting results for Figure 5 in Section 7.1. we show the same results but with the addition of *near optimum* priors. Figure 23 shows the relative rank comparison over near optimum, good, and bad priors, as well as the aggregate (*All*). Most notably, we see the relation between `PriorBand` and HB with respect to prior strength, showcasing substantial benefits of strong priors while recovering in the bad prior setting.

To further illustrate how different tasks affect optimizer behavior, we cluster the set of $12$ benchmarks we've chosen into $8$ good correlation benchmarks and $4$ bad correlation benchmarks as defined in Appendix D.1.5. Figure 23 shows the relative ranks for the same set of algorithms when grouped and aggregated along these criteria. `PriorBand` is the most robust algorithm shown here.

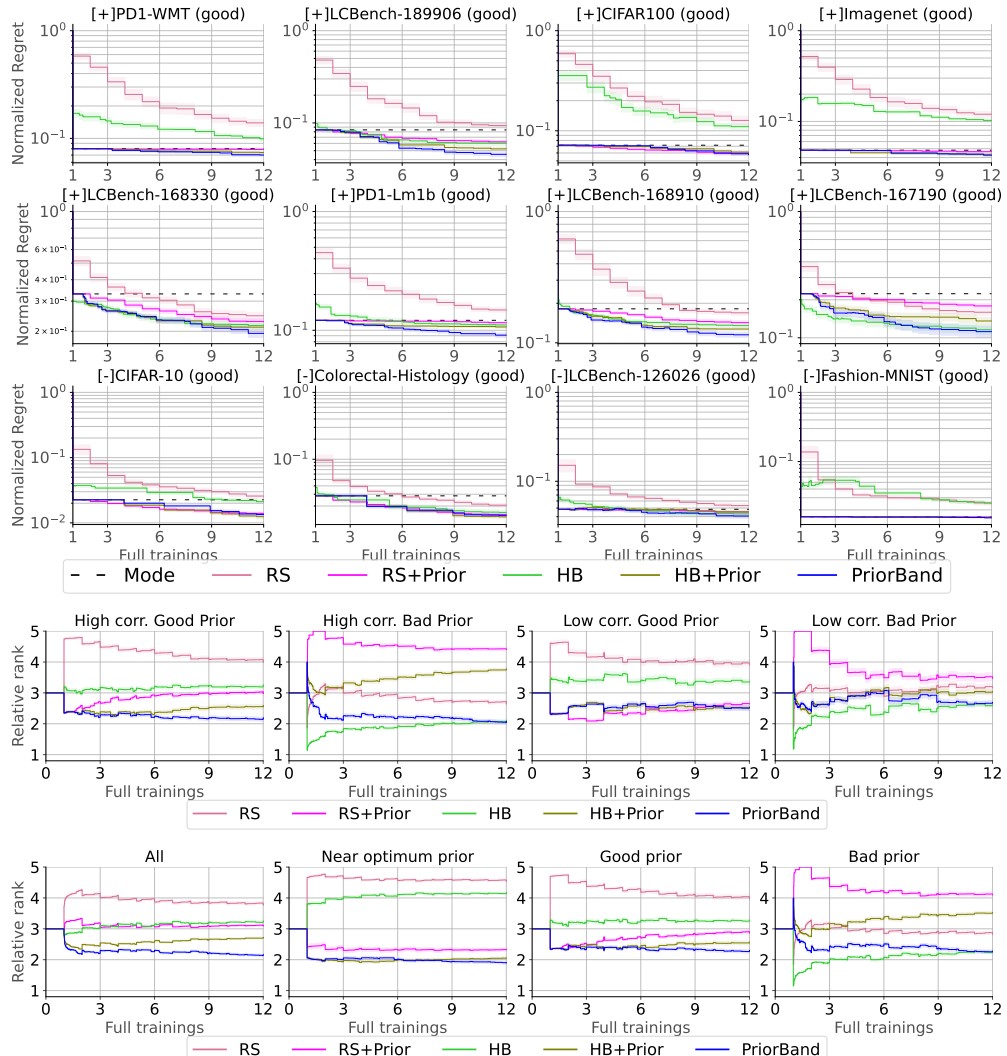

Figure 23: [Top] Each plot compares the algorithms based on the average normalized regret across 50 seeds, under the good prior setting. The optima for a benchmark was assumed to be the best scores achieved by all algorithms across all seeds. `PriorBand` is anytime best across in all cases. [+] denotes the benchmark tends to have a strong correlation across fidelities, [-] denotes a weak or poor correlation across fidelities; [Bottom] Comparing PriorBand with other baselines on single-worker runs for 50 seeds. The *top* row compares the average relative ranks across all benchmarks under different prior qualities, where the *All* averages over the 3 prior strengths too. The *bottom* row groups the benchmarks into high-low correlation sets based on fidelity correlations and creates 4 scenarios when combined with good-bad priors. `PriorBand` emerges as the most consistent performer across all 7 scenarios. Every other baseline has at least one setting where it is one of the 2 worst algorithms. Prior-based methods show improved performance with a good prior but suffer with a bad prior, as expected. Likewise, multi-fidelity methods benefit from high correlation. Under low-correlation settings, the ranking gap between RS and HB expectedly is lower.

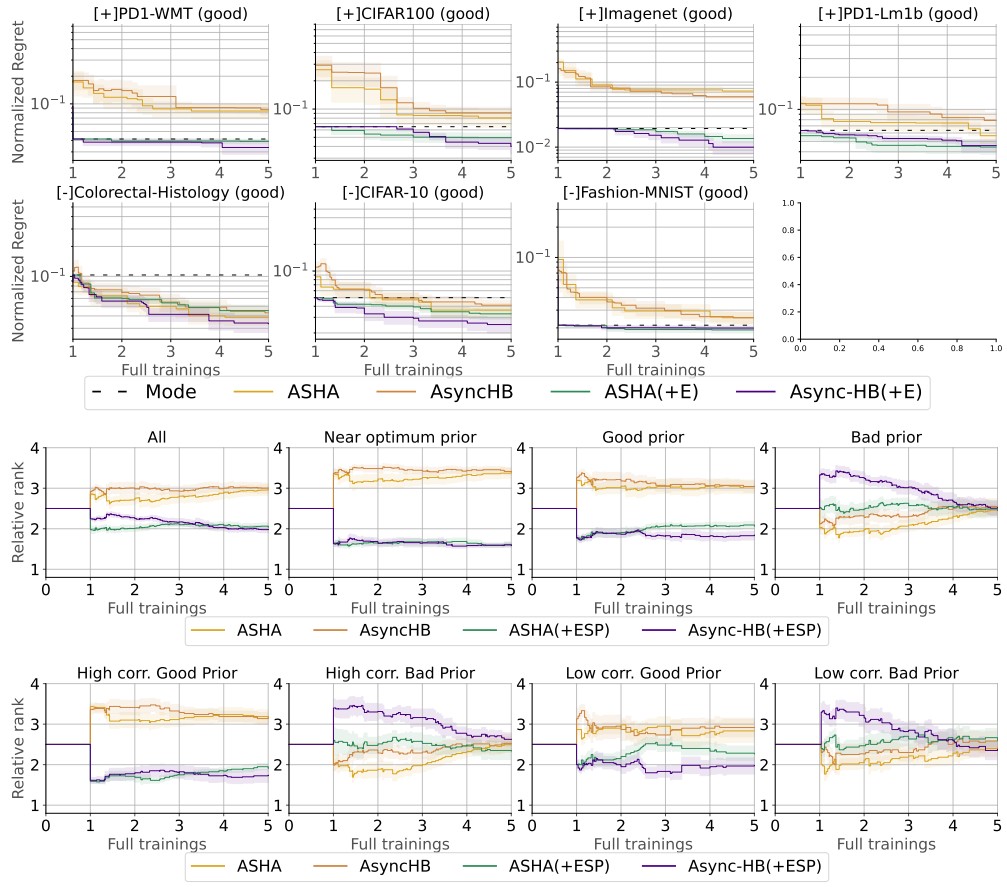

Figure 24: [Top] Each plot compares the algorithms based on the average normalized regret across 50 seeds, under the good prior setting, run over 4 workers each. The optima for a benchmark was assumed to be the best scores achieved by all algorithms across all seeds. [+] denotes the benchmark tends to have a strong correlation across fidelities, [-] denotes a weak or poor correlation across fidelities. Refer to Appendix D.1.6 for missing benchmark; [Bottom] Comparing ASHA, AsynchronousHB with their ESP augmented versions (+E) and `PriorBand` when distributed over 4 workers for a total budget of $20\times$. The *top* row compares the average relative ranks across all benchmarks under different prior qualities, where the *All* averages over the 3 prior strengths. The *bottom* row groups the benchmarks into high-low correlation sets based on fidelity correlations and creates 4 scenarios when combined with good-bad priors. All 3 algorithms with the ESP $\mathcal{E}_\pi$ show superior performance under good priors and strong recovery under bad priors. Under a bad prior for good correlation benchmarks, the recovery of ESP-based asynchronous methods is slower since the vanilla algorithms perform better under good correlation. Unlike the low correlation set under bad priors, where all $\mathcal{E}_\pi$-based algorithms show faster recovery. Overall, `PriorBand` remains a strong performer even in the parallel setup.

## F.2 Generality of $\mathcal{E}_\pi$

Figure 24 (top) compares 3 different prior qualities for more supporting results for Figure 6 in Section 7.2. Figure 24 (bottom) splits the set of 12 benchmarks into high-low correlations and plots their interaction with good-bad priors.

## F.3 Extensibility of $\mathcal{E}_\pi$ with models

More supporting results for Figure 7 in Section 7.3. In Figure 25, which compares 3 different prior qualities, we assess how the quality of the prior effects model-based methods. We also compare the

Table 10: Table comparing Random Search (RS), HyperBand (HB), and `PriorBand`'s final validation errors of the current incumbent at the highest fidelity at 2 budget horizons of $5\times$ and $12\times$. Runs are averaged over 50 seeds on 1 worker each. `PriorBand` shows superior anytime performance under informative priors. Under extremely bad priors, for shorter compute budgets ($5\times$) `PriorBand` has a poor start, however, given an adequate budget ($12\times$) `PriorBand` can match HB's performance on average.

| Benchmark | 5x | | | 12x | | |
|---|---|---|---|---|---|---|
| | | | **Near Optimum Prior** | | | |
| | RS | HB | PriorBand | RS | HB | PriorBand |
| **JAHS-C10** | $11.454 \pm 1.935$ | $11.129 \pm 1.256$ | $\mathbf{8.252 \pm 0.215}$ | $10.451 \pm 1.055$ | $10.077 \pm 0.769$ | $\mathbf{8.236 \pm 0.197}$ |
| **JAHS-CH** | $6.752 \pm 0.943$ | $6.332 \pm 1.12$ | $\mathbf{4.449 \pm 0.124}$ | $5.977 \pm 0.646$ | $5.704 \pm 0.659$ | $\mathbf{4.445 \pm 0.127}$ |
| **JAHS-FM** | $5.497 \pm 0.396$ | $5.846 \pm 0.823$ | $\mathbf{4.724 \pm 0.06}$ | $5.282 \pm 0.299$ | $5.306 \pm 0.267$ | $\mathbf{4.719 \pm 0.055}$ |
| **LC-126026** | $0.053 \pm 0.011$ | $0.046 \pm 0.01$ | $\mathbf{0.024 \pm 0.008}$ | $0.048 \pm 0.008$ | $0.044 \pm 0.008$ | $\mathbf{0.023 \pm 0.006}$ |
| **LC-167190** | $0.214 \pm 0.021$ | $0.193 \pm 0.025$ | $\mathbf{0.136 \pm 0.017}$ | $0.2 \pm 0.02$ | $0.187 \pm 0.023$ | $\mathbf{0.133 \pm 0.013}$ |
| **LC-168330** | $0.444 \pm 0.046$ | $0.41 \pm 0.035$ | $\mathbf{0.29 \pm 0.041}$ | $0.406 \pm 0.038$ | $0.397 \pm 0.032$ | $\mathbf{0.275 \pm 0.028}$ |
| **LC-168910** | $0.38 \pm 0.093$ | $0.314 \pm 0.021$ | $\mathbf{0.2 \pm 0.021}$ | $0.324 \pm 0.032$ | $0.308 \pm 0.018$ | $\mathbf{0.194 \pm 0.019}$ |
| **LC-189906** | $0.222 \pm 0.09$ | $0.151 \pm 0.021$ | $\mathbf{0.113 \pm 0.015}$ | $0.166 \pm 0.033$ | $0.145 \pm 0.019$ | $\mathbf{0.107 \pm 0.011}$ |
| **PD1-Cifar100** | $0.378 \pm 0.156$ | $0.338 \pm 0.091$ | $\mathbf{0.257 \pm 0.051}$ | $0.302 \pm 0.074$ | $0.29 \pm 0.039$ | $\mathbf{0.238 \pm 0.04}$ |
| **PD1-ImageNet** | $0.333 \pm 0.108$ | $0.306 \pm 0.041$ | $\mathbf{0.239 \pm 0.038}$ | $0.282 \pm 0.049$ | $0.268 \pm 0.022$ | $\mathbf{0.228 \pm 0.026}$ |
| **PD1-LM1B** | $0.681 \pm 0.034$ | $0.651 \pm 0.013$ | $\mathbf{0.637 \pm 0.015}$ | $0.658 \pm 0.013$ | $0.648 \pm 0.012$ | $\mathbf{0.632 \pm 0.015}$ |
| **PD1-WMT** | $0.454 \pm 0.097$ | $0.402 \pm 0.032$ | $\mathbf{0.357 \pm 0.029}$ | $0.403 \pm 0.038$ | $0.384 \pm 0.02$ | $\mathbf{0.347 \pm 0.025}$ |
| | | | **Good Prior** | | | |
| | RS | HB | PriorBand | RS | HB | PriorBand |
| **JAHS-C10** | $11.454 \pm 1.935$ | $11.129 \pm 1.256$ | $\mathbf{9.986 \pm 0.369}$ | $10.451 \pm 1.055$ | $10.077 \pm 0.769$ | $\mathbf{9.457 \pm 0.415}$ |
| **JAHS-CH** | $6.752 \pm 0.943$ | $6.332 \pm 1.12$ | $\mathbf{5.974 \pm 0.535}$ | $5.977 \pm 0.646$ | $5.704 \pm 0.659$ | $\mathbf{5.589 \pm 0.413}$ |
| **JAHS-FM** | $5.497 \pm 0.396$ | $5.846 \pm 0.823$ | $\mathbf{5.037 \pm 0.021}$ | $5.282 \pm 0.299$ | $5.306 \pm 0.267$ | $\mathbf{5.026 \pm 0.044}$ |
| **LC-126026** | $0.053 \pm 0.011$ | $0.046 \pm 0.01$ | $\mathbf{0.045 \pm 0.006}$ | $0.048 \pm 0.008$ | $0.044 \pm 0.008$ | $\mathbf{0.043 \pm 0.007}$ |
| **LC-167190** | $0.214 \pm 0.021$ | $\mathbf{0.193 \pm 0.025}$ | $0.198 \pm 0.026$ | $0.2 \pm 0.02$ | $0.187 \pm 0.023$ | $\mathbf{0.186 \pm 0.026}$ |
| **LC-168330** | $0.444 \pm 0.046$ | $\mathbf{0.41 \pm 0.035}$ | $0.411 \pm 0.024$ | $0.406 \pm 0.038$ | $0.397 \pm 0.032$ | $\mathbf{0.387 \pm 0.035}$ |
| **LC-168910** | $0.38 \pm 0.093$ | $0.314 \pm 0.021$ | $\mathbf{0.313 \pm 0.019}$ | $0.324 \pm 0.032$ | $0.308 \pm 0.018$ | $\mathbf{0.295 \pm 0.023}$ |
| **LC-189906** | $0.222 \pm 0.09$ | $0.151 \pm 0.021$ | $\mathbf{0.147 \pm 0.013}$ | $0.166 \pm 0.033$ | $0.145 \pm 0.019$ | $\mathbf{0.134 \pm 0.013}$ |
| **PD1-Cifar100** | $0.378 \pm 0.156$ | $0.338 \pm 0.091$ | $\mathbf{0.258 \pm 0.002}$ | $0.302 \pm 0.074$ | $0.29 \pm 0.039$ | $\mathbf{0.248 \pm 0.013}$ |
| **PD1-ImageNet** | $0.333 \pm 0.108$ | $0.306 \pm 0.041$ | $\mathbf{0.224 \pm 0.001}$ | $0.282 \pm 0.049$ | $0.268 \pm 0.022$ | $\mathbf{0.219 \pm 0.008}$ |
| **PD1-LM1B** | $0.681 \pm 0.034$ | $0.651 \pm 0.013$ | $\mathbf{0.647 \pm 0.008}$ | $0.658 \pm 0.013$ | $0.648 \pm 0.012$ | $\mathbf{0.642 \pm 0.009}$ |
| **PD1-WMT** | $0.454 \pm 0.097$ | $0.402 \pm 0.032$ | $\mathbf{0.37 \pm 0.008}$ | $0.403 \pm 0.038$ | $0.384 \pm 0.02$ | $\mathbf{0.367 \pm 0.009}$ |
| | | | **Bad Prior** | | | |
| | RS | HB | PriorBand | RS | HB | PriorBand |
| **JAHS-C10** | $11.454 \pm 1.935$ | $\mathbf{11.129 \pm 1.256}$ | $11.729 \pm 1.795$ | $10.451 \pm 1.055$ | $10.077 \pm 0.769$ | $\mathbf{10.075 \pm 0.777}$ |
| **JAHS-CH** | $6.752 \pm 0.943$ | $\mathbf{6.332 \pm 1.12}$ | $6.553 \pm 1.284$ | $5.977 \pm 0.646$ | $5.704 \pm 0.659$ | $\mathbf{5.551 \pm 0.516}$ |
| **JAHS-FM** | $\mathbf{5.497 \pm 0.396}$ | $5.846 \pm 0.823$ | $5.962 \pm 0.973$ | $5.282 \pm 0.299$ | $5.306 \pm 0.267$ | $\mathbf{5.277 \pm 0.248}$ |
| **LC-126026** | $0.053 \pm 0.011$ | $\mathbf{0.046 \pm 0.01}$ | $0.051 \pm 0.011$ | $0.048 \pm 0.008$ | $\mathbf{0.044 \pm 0.008}$ | $0.045 \pm 0.01$ |
| **LC-167190** | $0.214 \pm 0.021$ | $\mathbf{0.193 \pm 0.025}$ | $0.196 \pm 0.024$ | $0.2 \pm 0.02$ | $0.187 \pm 0.023$ | $\mathbf{0.183 \pm 0.026}$ |
| **LC-168330** | $0.444 \pm 0.046$ | $\mathbf{0.41 \pm 0.035}$ | $0.424 \pm 0.035$ | $0.406 \pm 0.038$ | $\mathbf{0.397 \pm 0.032}$ | $0.401 \pm 0.042$ |
| **LC-168910** | $0.38 \pm 0.093$ | $0.314 \pm 0.021$ | $\mathbf{0.309 \pm 0.019}$ | $0.324 \pm 0.032$ | $0.308 \pm 0.018$ | $\mathbf{0.297 \pm 0.016}$ |
| **LC-189906** | $0.222 \pm 0.09$ | $\mathbf{0.151 \pm 0.021}$ | $0.167 \pm 0.03$ | $0.166 \pm 0.033$ | $\mathbf{0.145 \pm 0.019}$ | $0.15 \pm 0.027$ |
| **PD1-Cifar100** | $0.378 \pm 0.156$ | $\mathbf{0.338 \pm 0.091}$ | $0.473 \pm 0.2$ | $0.302 \pm 0.074$ | $\mathbf{0.29 \pm 0.039}$ | $0.297 \pm 0.058$ |
| **PD1-ImageNet** | $0.333 \pm 0.108$ | $\mathbf{0.306 \pm 0.041}$ | $0.315 \pm 0.04$ | $0.282 \pm 0.049$ | $\mathbf{0.268 \pm 0.022}$ | $0.28 \pm 0.03$ |
| **PD1-LM1B** | $0.681 \pm 0.034$ | $\mathbf{0.651 \pm 0.013}$ | $0.656 \pm 0.014$ | $0.658 \pm 0.013$ | $\mathbf{0.648 \pm 0.012}$ | $0.648 \pm 0.013$ |
| **PD1-WMT** | $0.454 \pm 0.097$ | $\mathbf{0.402 \pm 0.032}$ | $0.425 \pm 0.057$ | $0.403 \pm 0.038$ | $0.384 \pm 0.02$ | $\mathbf{0.383 \pm 0.021}$ |

model-based extension of `PriorBand` (`PriorBand+BO`) against Mobster (asynchronous BOHB) in the distributed setting, in Figure 26.

## F.4 Comparing prior-based baselines

We would like to determine if `PriorBand` is sufficient to outperform existing prior-based baselines as well as our own extension of `PriorBand` which is model-based, namely PriorBand+BO. Figure 27 show the results of this comparison.

## F.5 Final performance tables

In this section, Tables 10, 11, 12, 13, 14, compare the performance of `PriorBand`, its BO extension and other applications of $\mathcal{E}_\pi$, to commonly used algorithms from the literature for the different classes of optimizers. The tables show the final performance at 2 budget horizons, highlighting the robustness of the ESP-based algorithms.

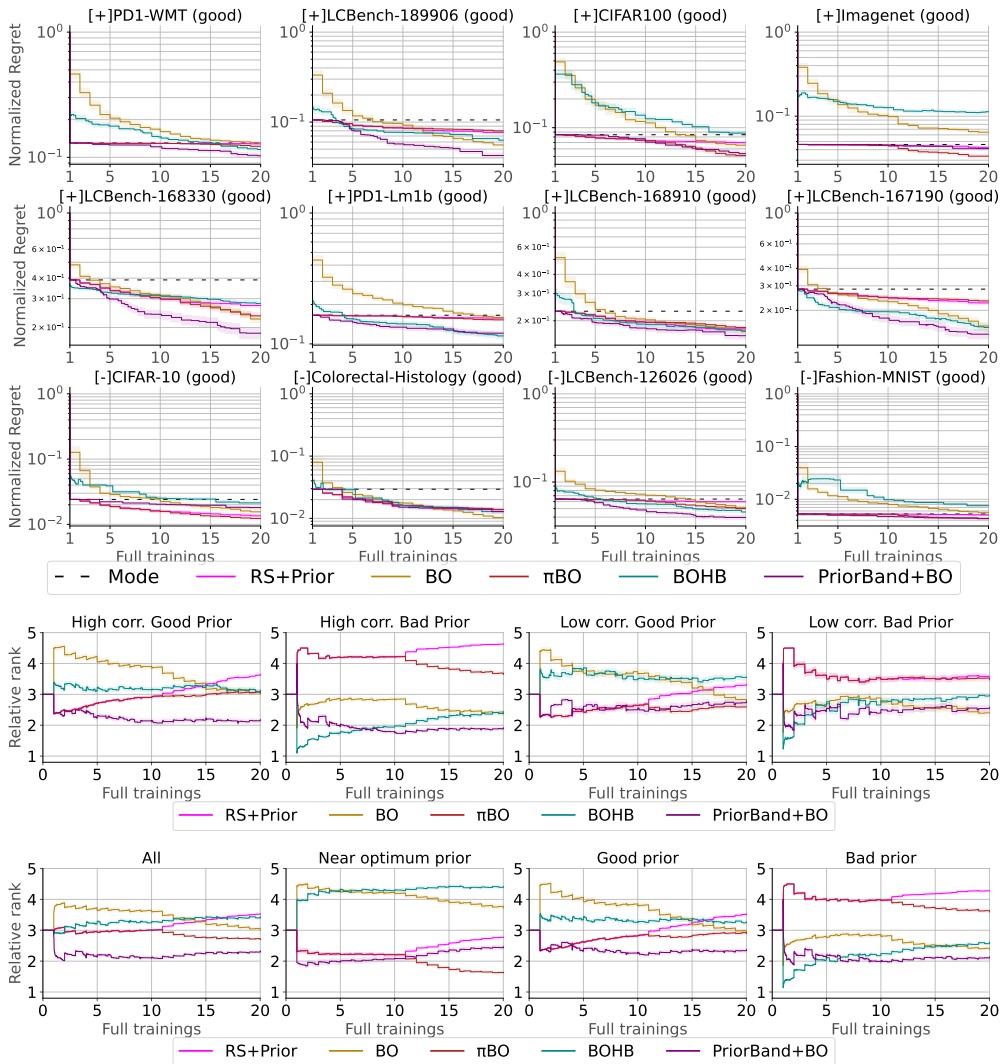

Figure 25: [Top] Each plot compares the algorithms based on the average normalized regret across 50 seeds, under the good prior setting. The optima for a benchmark was assumed to be the best scores achieved by all algorithms across all seeds. `PriorBand+BO` is anytime best across in all cases. [+] denotes the benchmark tends to have a strong correlation across fidelities, [-] denotes a weak or poor correlation across fidelities; [Bottom] Comparing model extensions over 3 sets of prior qualities. Using PriorBand with BO shows a dominant performance in almost all cases. In the first row, we show that $\pi$BO does outperform all other methods in the near optimum setting, once model sampling activates. This is likely due to $\pi$BO's emphasis on the prior which is the same cause for its poor performance in the bad prior setting. Both BO and BOHB suffer from having no access to the prior in the near optimum/good prior setting while only BOHB really has an advantage over PriorBand+BO early on in the bad prior setting. In the second row, we see that extending Mobster with the ensembling policy $\mathcal{E}_\pi$ imbues Mobster with an effective mechanism for exploiting priors. However, this exploitation has a more consistent negative impact when the prior is bad, at least when compared to vanilla Mobster and our model-based PriorBand+BO.

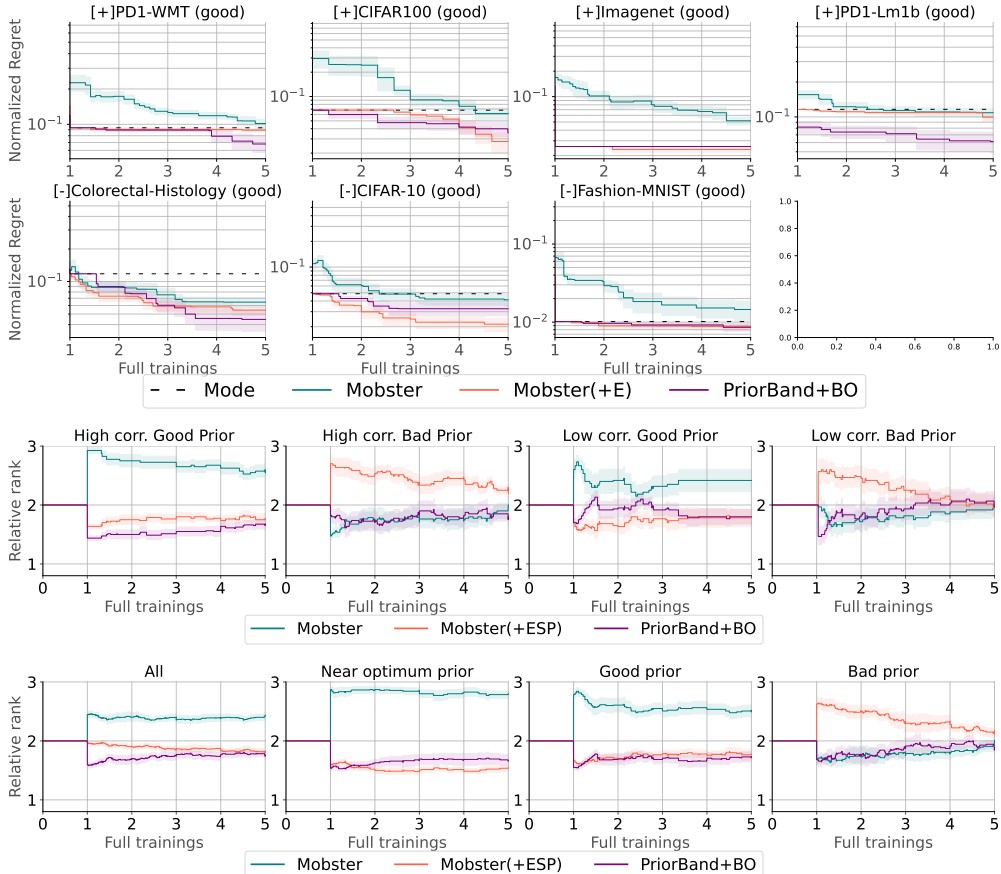

Figure 26: [Top] Each plot compares the algorithms based on the average normalized regret across 50 seeds, under the good prior setting, run over 4 workers each. The optima for a benchmark was assumed to be the best scores achieved by all algorithms across all seeds. [+] denotes the benchmark tends to have a strong correlation across fidelities, [-] denotes a weak or poor correlation across fidelities. Refer to Appendix D.1.6 for missing benchmark; [Bottom] Comparing model extensions over 3 sets of prior qualities. Using PriorBand with BO shows a dominant performance in almost all cases. In the first row, we show that $\pi$BO does outperform all other methods in the near optimum setting, once model sampling activates. This is likely due to $\pi$BO's emphasis on the prior which is the same cause for its poor performance in the bad prior setting. Both BO and BOHB suffer from having no access to the prior in the near optimum/good prior setting while only BOHB really has an advantage over PriorBand+BO early on in the bad prior setting. In the second row, we see that extending Mobster with the ensembling policy $\mathcal{E}_\pi$ imbues Mobster with an effective mechanism for exploiting priors. However, this exploitation has a more consistent negative impact when the prior is bad, at least when compared to vanilla Mobster and our model-based PriorBand+BO.

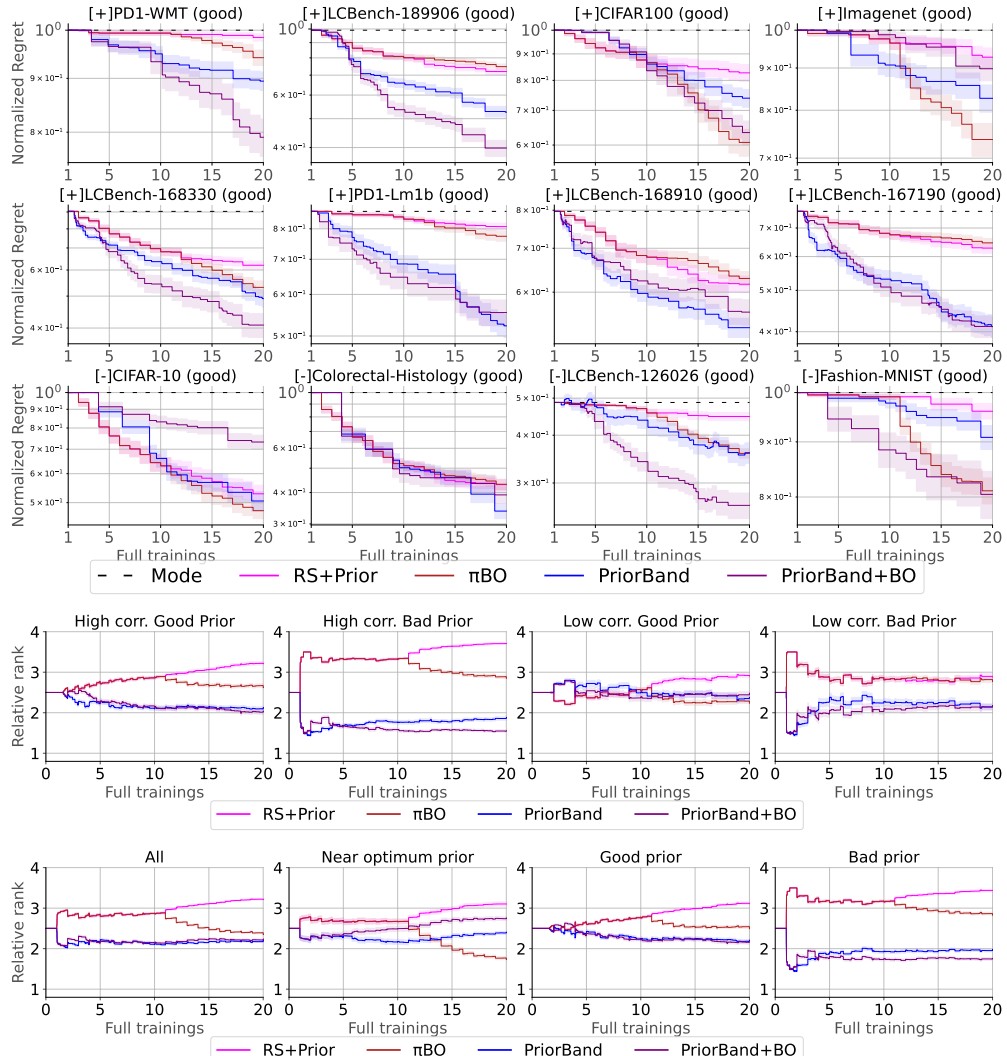

Figure 27: [Top] Each plot compares the algorithms based on the average normalized regret across 50 seeds, under the good prior setting. The optima for a benchmark was assumed to be the best scores achieved by all algorithms across all seeds. `PriorBand+BO` offers its model benefits over `PriorBand` under longer budgets but has mixed performance under poor correlation of fidelity performance. [+] denotes the benchmark tends to have a strong correlation across fidelities, [-] denotes a weak or poor correlation across fidelities; [Bottom] Comparing model-based methods which incorporate the prior. In the near optimum case, πBO's strong performance only occurs once its model-based sampling occurs. This strong emphasis on searching around the prior is also what causes πBO's weaker performance in the bad prior case. When comparing PriorBand against its model counterpart, PriorBand+BO, we see that PriorBand utilizes the near optimum prior better while PriorBand+BO with its model-based sampling leads to better performance in the long run if the prior is bad. This trade-off is averaged off in the case of a good prior which is neither overly optimistic as the near optimum prior is, nor neither as pessimistic as the bad prior is.

Table 11: Table comparing Asynchronous Successive Halving (ASHA), Asynchronous HyperBand (AsyncHB) and `PriorBand` final validation errors of the current incumbent at the highest fidelity at 2 budget horizons of $1\times$ and $5\times$. Runs are averaged over 10 seeds where each run is with 4 workers. Unlike the other 2 algorithms, `PriorBand` does not performance ASHA-like asynchronous promotions to minimize idle workers. `PriorBand` simply starts the next SH bracket if a worker is free. The table shows that `PriorBand` can maintain its robust performance when run in a parallel setting while being competitive to related asynchronous algorithms.

| Benchmark | 1x | | | 5x | | |
|---|---|---|---|---|---|---|
| | ASHA | AsyncHB | PriorBand | ASHA | AsyncHB | PriorBand |
| **Near Optimum Prior** | | | | | | |
| **JAHS-C10** | $11.382 \pm 2.523$ | $12.192 \pm 1.558$ | $\mathbf{8.281 \pm 0.221}$ | $9.739 \pm 0.716$ | $9.901 \pm 0.598$ | $\mathbf{8.281 \pm 0.221}$ |
| **JAHS-CH** | $6.325 \pm 1.116$ | $6.771 \pm 1.296$ | $\mathbf{4.466 \pm 0.188}$ | $5.492 \pm 0.362$ | $5.565 \pm 0.342$ | $\mathbf{4.399 \pm 0.143}$ |
| **JAHS-FM** | $6.627 \pm 3.384$ | $6.245 \pm 0.999$ | $\mathbf{4.713 \pm 0.052}$ | $5.127 \pm 0.235$ | $5.134 \pm 0.274$ | $\mathbf{4.709 \pm 0.049}$ |
| **PD1-Cifar100** | $0.411 \pm 0.152$ | $0.434 \pm 0.172$ | $\mathbf{0.243 \pm 0.056}$ | $0.271 \pm 0.04$ | $0.279 \pm 0.032$ | $\mathbf{0.221 \pm 0.043}$ |
| **PD1-ImageNet** | $0.366 \pm 0.049$ | $0.333 \pm 0.056$ | $\mathbf{0.259 \pm 0.059}$ | $0.265 \pm 0.019$ | $0.255 \pm 0.011$ | $\mathbf{0.218 \pm 0.021}$ |
| **PD1-LM1B** | $0.665 \pm 0.016$ | $0.663 \pm 0.015$ | $\mathbf{0.643 \pm 0.027}$ | $0.649 \pm 0.008$ | $0.655 \pm 0.016$ | $\mathbf{0.628 \pm 0.008}$ |
| **PD1-WMT** | $0.446 \pm 0.09$ | $0.443 \pm 0.06$ | $\mathbf{0.362 \pm 0.033}$ | $0.396 \pm 0.026$ | $0.402 \pm 0.027$ | $\mathbf{0.349 \pm 0.025}$ |
| **Good Prior** | | | | | | |
| **JAHS-C10** | $11.382 \pm 2.523$ | $12.192 \pm 1.558$ | $\mathbf{10.194 \pm 0.0}$ | $9.739 \pm 0.716$ | $9.901 \pm 0.598$ | $\mathbf{9.413 \pm 0.42}$ |
| **JAHS-CH** | $\mathbf{6.325 \pm 1.116}$ | $6.771 \pm 1.296$ | $6.603 \pm 0.0$ | $5.492 \pm 0.362$ | $5.565 \pm 0.342$ | $\mathbf{5.238 \pm 0.325}$ |
| **JAHS-FM** | $6.627 \pm 3.384$ | $6.184 \pm 1.009$ | $\mathbf{5.042 \pm 0.0}$ | $5.127 \pm 0.235$ | $5.126 \pm 0.211$ | $\mathbf{5.024 \pm 0.04}$ |
| **PD1-Cifar100** | $0.411 \pm 0.152$ | $0.434 \pm 0.172$ | $\mathbf{0.259 \pm 0.0}$ | $0.271 \pm 0.04$ | $0.279 \pm 0.032$ | $\mathbf{0.237 \pm 0.016}$ |
| **PD1-ImageNet** | $0.366 \pm 0.049$ | $0.333 \pm 0.056$ | $\mathbf{0.224 \pm 0.0}$ | $0.265 \pm 0.019$ | $0.254 \pm 0.011$ | $\mathbf{0.217 \pm 0.008}$ |
| **PD1-LM1B** | $0.665 \pm 0.016$ | $0.665 \pm 0.013$ | $\mathbf{0.65 \pm 0.002}$ | $0.649 \pm 0.008$ | $0.655 \pm 0.013$ | $\mathbf{0.638 \pm 0.013}$ |
| **PD1-WMT** | $0.446 \pm 0.09$ | $0.45 \pm 0.065$ | $\mathbf{0.372 \pm 0.0}$ | $0.396 \pm 0.026$ | $0.398 \pm 0.019$ | $\mathbf{0.356 \pm 0.014}$ |
| **Bad Prior** | | | | | | |
| **JAHS-C10** | $11.382 \pm 2.523$ | $12.135 \pm 1.62$ | $\mathbf{11.055 \pm 1.768}$ | $\mathbf{9.739 \pm 0.716}$ | $9.869 \pm 0.586$ | $9.806 \pm 0.632$ |
| **JAHS-CH** | $\mathbf{6.325 \pm 1.116}$ | $6.771 \pm 1.296$ | $6.622 \pm 0.877$ | $5.492 \pm 0.362$ | $5.565 \pm 0.342$ | $\mathbf{5.316 \pm 0.667}$ |
| **JAHS-FM** | $6.627 \pm 3.384$ | $6.184 \pm 1.009$ | $\mathbf{5.715 \pm 0.485}$ | $\mathbf{5.127 \pm 0.235}$ | $5.175 \pm 0.274$ | $5.333 \pm 0.332$ |
| **PD1-Cifar100** | $\mathbf{0.411 \pm 0.152}$ | $0.434 \pm 0.172$ | $0.657 \pm 0.266$ | $0.271 \pm 0.04$ | $0.279 \pm 0.032$ | $\mathbf{0.256 \pm 0.036}$ |
| **PD1-ImageNet** | $0.366 \pm 0.049$ | $0.335 \pm 0.058$ | $\mathbf{0.324 \pm 0.062}$ | $0.265 \pm 0.019$ | $0.258 \pm 0.015$ | $\mathbf{0.257 \pm 0.027}$ |
| **PD1-LM1B** | $0.665 \pm 0.016$ | $\mathbf{0.665 \pm 0.013}$ | $0.668 \pm 0.016$ | $0.649 \pm 0.008$ | $0.654 \pm 0.013$ | $\mathbf{0.646 \pm 0.012}$ |
| **PD1-WMT** | $\mathbf{0.446 \pm 0.09}$ | $0.45 \pm 0.065$ | $0.485 \pm 0.141$ | $0.396 \pm 0.026$ | $0.398 \pm 0.019$ | $\mathbf{0.375 \pm 0.014}$ |

Table 12: Table comparing Asynchronous Successive Halving (ASHA), Asynchronous HyperBand (AsyncHB), ASHA+$\mathcal{E}_\pi$ and Asynchronous-HyperBand+$\mathcal{E}_\pi$ final validation errors of the current incumbent at the highest fidelity at 2 budget horizons of $1\times$ and $5\times$. Runs are averaged over 10 seeds where each run is with 4 workers. This table showcases the flexibility of the ESP, as it can be applied off-the-shelf to other multi-fidelity algorithms too. Under informative priors, ASHA(+ESP) performs marginally better than Asynchronous-HB(+ESP) but wanes for longer budgets. Interestingly, ASHA(+ESP) retains a better performance under the bad priors. Since ASHA(+ESP) samples only at the $r = 0$, the ESP fixes $p_{\mathcal{U}}$ to $50\%$. This could explain the increased exploration under bad priors and reduced exploitation under the good priors for ASHA(+ESP).

| Benchmark | 1x | | | | 5x | | | |
|---|---|---|---|---|---|---|---|---|
| | ASHA | AsyncHB | ASHA(+ESP) | Async-HB(+ESP) | ASHA | AsyncHB | ASHA(+ESP) | Async-HB(+ESP) |
| **Near Optimum Prior** | | | | | | | | |
| **JAHS-C10** | $11.382 \pm 2.523$ | $12.192 \pm 1.558$ | $\mathbf{8.281 \pm 0.221}$ | $8.281 \pm 0.221$ | $9.739 \pm 0.716$ | $9.901 \pm 0.598$ | $8.223 \pm 0.183$ | $\mathbf{8.18 \pm 0.177}$ |
| **JAHS-CH** | $6.325 \pm 1.116$ | $6.771 \pm 1.296$ | $\mathbf{4.466 \pm 0.188}$ | $4.466 \pm 0.188$ | $5.492 \pm 0.362$ | $5.565 \pm 0.342$ | $\mathbf{4.466 \pm 0.188}$ | $4.466 \pm 0.188$ |
| **JAHS-FM** | $6.627 \pm 3.384$ | $6.245 \pm 0.999$ | $\mathbf{4.713 \pm 0.052}$ | $4.713 \pm 0.052$ | $5.127 \pm 0.235$ | $5.134 \pm 0.274$ | $4.709 \pm 0.049$ | $\mathbf{4.707 \pm 0.05}$ |
| **PD1-Cifar100** | $0.411 \pm 0.152$ | $0.434 \pm 0.172$ | $\mathbf{0.243 \pm 0.056}$ | $0.259 \pm 0.083$ | $0.271 \pm 0.04$ | $0.279 \pm 0.032$ | $\mathbf{0.221 \pm 0.037}$ | $0.227 \pm 0.04$ |
| **PD1-ImageNet** | $0.366 \pm 0.049$ | $0.333 \pm 0.056$ | $0.269 \pm 0.07$ | $\mathbf{0.261 \pm 0.06}$ | $0.265 \pm 0.019$ | $0.255 \pm 0.011$ | $0.22 \pm 0.021$ | $\mathbf{0.214 \pm 0.017}$ |
| **PD1-LM1B** | $0.665 \pm 0.016$ | $0.663 \pm 0.015$ | $0.644 \pm 0.025$ | $\mathbf{0.639 \pm 0.018}$ | $0.649 \pm 0.008$ | $0.655 \pm 0.016$ | $0.629 \pm 0.01$ | $\mathbf{0.628 \pm 0.009}$ |
| **PD1-WMT** | $0.446 \pm 0.09$ | $0.443 \pm 0.06$ | $\mathbf{0.364 \pm 0.036}$ | $0.373 \pm 0.053$ | $0.396 \pm 0.026$ | $0.402 \pm 0.027$ | $\mathbf{0.344 \pm 0.035}$ | $0.353 \pm 0.026$ |
| **Good Prior** | | | | | | | | |
| **JAHS-C10** | $11.382 \pm 2.523$ | $12.192 \pm 1.558$ | $\mathbf{10.194 \pm 0.0}$ | $10.194 \pm 0.0$ | $9.739 \pm 0.716$ | $9.901 \pm 0.598$ | $9.603 \pm 0.412$ | $\mathbf{9.365 \pm 0.51}$ |
| **JAHS-CH** | $\mathbf{6.325 \pm 1.116}$ | $6.771 \pm 1.296$ | $6.603 \pm 0.0$ | $6.603 \pm 0.0$ | $5.492 \pm 0.362$ | $5.565 \pm 0.342$ | $5.592 \pm 0.31$ | $\mathbf{5.386 \pm 0.327}$ |
| **JAHS-FM** | $6.627 \pm 3.384$ | $6.184 \pm 1.009$ | $\mathbf{5.042 \pm 0.0}$ | $5.042 \pm 0.0$ | $5.127 \pm 0.235$ | $5.126 \pm 0.211$ | $\mathbf{4.995 \pm 0.088}$ | $5.013 \pm 0.06$ |
| **PD1-Cifar100** | $0.411 \pm 0.152$ | $0.434 \pm 0.172$ | $\mathbf{0.259 \pm 0.0}$ | $0.259 \pm 0.0$ | $0.271 \pm 0.04$ | $0.279 \pm 0.032$ | $0.241 \pm 0.014$ | $\mathbf{0.24 \pm 0.017}$ |
| **PD1-ImageNet** | $0.366 \pm 0.049$ | $0.333 \pm 0.056$ | $\mathbf{0.224 \pm 0.0}$ | $0.224 \pm 0.0$ | $0.265 \pm 0.019$ | $0.254 \pm 0.011$ | $0.22 \pm 0.005$ | $\mathbf{0.217 \pm 0.005}$ |
| **PD1-LM1B** | $0.665 \pm 0.016$ | $0.665 \pm 0.013$ | $\mathbf{0.649 \pm 0.004}$ | $0.651 \pm 0.0$ | $0.649 \pm 0.008$ | $0.655 \pm 0.013$ | $\mathbf{0.646 \pm 0.006}$ | $0.646 \pm 0.005$ |
| **PD1-WMT** | $0.446 \pm 0.09$ | $0.45 \pm 0.065$ | $\mathbf{0.372 \pm 0.0}$ | $0.372 \pm 0.0$ | $0.396 \pm 0.026$ | $0.398 \pm 0.019$ | $0.371 \pm 0.004$ | $\mathbf{0.368 \pm 0.009}$ |
| **Bad Prior** | | | | | | | | |
| **JAHS-C10** | $\mathbf{11.382 \pm 2.523}$ | $12.135 \pm 1.62$ | $12.697 \pm 2.049$ | $13.857 \pm 3.841$ | $9.739 \pm 0.716$ | $9.869 \pm 0.586$ | $\mathbf{9.669 \pm 0.502}$ | $9.809 \pm 0.774$ |
| **JAHS-CH** | $\mathbf{6.325 \pm 1.116}$ | $6.771 \pm 1.296$ | $6.778 \pm 0.904$ | $9.416 \pm 2.559$ | $\mathbf{5.492 \pm 0.362}$ | $5.565 \pm 0.342$ | $5.856 \pm 0.401$ | $5.506 \pm 0.703$ |
| **JAHS-FM** | $6.627 \pm 3.384$ | $\mathbf{6.184 \pm 1.009}$ | $7.091 \pm 3.287$ | $7.244 \pm 3.419$ | $5.127 \pm 0.235$ | $5.175 \pm 0.274$ | $\mathbf{5.113 \pm 0.299}$ | $5.134 \pm 0.264$ |
| **PD1-Cifar100** | $\mathbf{0.411 \pm 0.152}$ | $0.434 \pm 0.172$ | $0.479 \pm 0.226$ | $0.665 \pm 0.186$ | $\mathbf{0.271 \pm 0.04}$ | $0.279 \pm 0.032$ | $0.29 \pm 0.041$ | $0.293 \pm 0.02$ |
| **PD1-ImageNet** | $0.366 \pm 0.049$ | $\mathbf{0.335 \pm 0.058}$ | $0.401 \pm 0.177$ | $0.558 \pm 0.155$ | $0.265 \pm 0.019$ | $\mathbf{0.258 \pm 0.015}$ | $0.291 \pm 0.098$ | $0.288 \pm 0.052$ |
| **PD1-LM1B** | $0.665 \pm 0.016$ | $\mathbf{0.665 \pm 0.013}$ | $0.687 \pm 0.046$ | $0.705 \pm 0.042$ | $0.649 \pm 0.008$ | $0.654 \pm 0.013$ | $\mathbf{0.647 \pm 0.018}$ | $0.652 \pm 0.012$ |
| **PD1-WMT** | $\mathbf{0.446 \pm 0.09}$ | $0.45 \pm 0.065$ | $0.54 \pm 0.172$ | $0.621 \pm 0.152$ | $0.396 \pm 0.026$ | $0.398 \pm 0.019$ | $\mathbf{0.383 \pm 0.022}$ | $0.395 \pm 0.029$ |

Table 13: Table comparing Bayesian Optimization (BO), $\pi$BO, BOHB and `PriorBand+BO` final validation errors of the current incumbent at the highest fidelity at 2 budget horizons of $10\times$ and $20\times$. Runs are averaged over $50$ seeds on 1 worker each. BO, $\pi$BO and `PriorBand+BO` use an initial design size of 10. This table highlights the model extensibility of `PriorBand`. Under near-optimum priors, $\pi$BO emerges as the best, especially after $10\times$. Knowledge of a near-optimum region is not available in practice and $\pi$BO's rate of recovery from bad priors is costlier for DL. `PriorBand+BO` importantly is better than BOHB in almost all settings and better than vanilla-BO.

| Benchmark | 10x | | | | 20x | | | |
|---|---|---|---|---|---|---|---|---|
| | **Near Optimum Prior** | | | | | | | |
| | BO | $\pi$BO | BOHB | PriorBand+BO | BO | $\pi$BO | BOHB | PriorBand+BO |
| **JAHS-C10** | $9.723 \pm 0.949$ | $\mathbf{8.23 \pm 0.229}$ | $10.281 \pm 0.88$ | $8.252 \pm 0.215$ | $9.115 \pm 0.725$ | $\mathbf{8.07 \pm 0.125}$ | $9.969 \pm 0.688$ | $8.248 \pm 0.21$ |
| **JAHS-CH** | $5.445 \pm 0.493$ | $4.481 \pm 0.112$ | $5.628 \pm 0.613$ | $\mathbf{4.452 \pm 0.127}$ | $4.964 \pm 0.426$ | $\mathbf{4.402 \pm 0.089}$ | $5.393 \pm 0.442$ | $4.452 \pm 0.127$ |
| **JAHS-FM** | $5.097 \pm 0.223$ | $\mathbf{4.71 \pm 0.048}$ | $5.406 \pm 0.323$ | $4.724 \pm 0.06$ | $4.917 \pm 0.172$ | $\mathbf{4.687 \pm 0.042}$ | $5.227 \pm 0.255$ | $4.724 \pm 0.06$ |
| **LC-126026** | $0.042 \pm 0.007$ | $0.026 \pm 0.006$ | $0.043 \pm 0.009$ | $\mathbf{0.024 \pm 0.007}$ | $0.031 \pm 0.008$ | $\mathbf{0.017 \pm 0.006}$ | $0.038 \pm 0.008$ | $0.023 \pm 0.005$ |
| **LC-167190** | $0.188 \pm 0.021$ | $0.138 \pm 0.014$ | $0.191 \pm 0.022$ | $\mathbf{0.136 \pm 0.016}$ | $0.164 \pm 0.023$ | $\mathbf{0.119 \pm 0.012}$ | $0.175 \pm 0.023$ | $0.134 \pm 0.014$ |
| **LC-168330** | $0.38 \pm 0.043$ | $\mathbf{0.289 \pm 0.03}$ | $0.412 \pm 0.032$ | $0.295 \pm 0.044$ | $0.322 \pm 0.044$ | $\mathbf{0.258 \pm 0.024}$ | $0.395 \pm 0.033$ | $0.281 \pm 0.035$ |
| **LC-168910** | $0.294 \pm 0.033$ | $\mathbf{0.198 \pm 0.021}$ | $0.31 \pm 0.033$ | $0.2 \pm 0.021$ | $0.266 \pm 0.042$ | $\mathbf{0.171 \pm 0.022}$ | $0.297 \pm 0.025$ | $0.2 \pm 0.02$ |
| **LC-189906** | $0.14 \pm 0.021$ | $0.112 \pm 0.011$ | $0.141 \pm 0.014$ | $\mathbf{0.111 \pm 0.012}$ | $0.117 \pm 0.015$ | $\mathbf{0.101 \pm 0.008}$ | $0.13 \pm 0.011$ | $0.106 \pm 0.011$ |
| **PD1-Cifar100** | $0.284 \pm 0.041$ | $0.239 \pm 0.035$ | $0.291 \pm 0.039$ | $\mathbf{0.238 \pm 0.035}$ | $0.238 \pm 0.022$ | $\mathbf{0.224 \pm 0.03}$ | $0.262 \pm 0.03$ | $0.227 \pm 0.029$ |
| **PD1-ImageNet** | $0.261 \pm 0.033$ | $\mathbf{0.222 \pm 0.02}$ | $0.289 \pm 0.035$ | $0.238 \pm 0.035$ | $0.234 \pm 0.02$ | $\mathbf{0.215 \pm 0.017}$ | $0.278 \pm 0.029$ | $0.228 \pm 0.028$ |
| **PD1-LM1B** | $0.66 \pm 0.015$ | $0.644 \pm 0.02$ | $0.643 \pm 0.014$ | $\mathbf{0.637 \pm 0.014}$ | $0.646 \pm 0.014$ | $\mathbf{0.631 \pm 0.018}$ | $0.634 \pm 0.014$ | $0.633 \pm 0.012$ |
| **PD1-WMT** | $0.387 \pm 0.026$ | $\mathbf{0.349 \pm 0.025}$ | $0.382 \pm 0.022$ | $0.353 \pm 0.027$ | $0.362 \pm 0.021$ | $\mathbf{0.335 \pm 0.022}$ | $0.363 \pm 0.017$ | $0.346 \pm 0.021$ |
| | **Good Prior** | | | | | | | |
| | BO | $\pi$BO | BOHB | PriorBand+BO | BO | $\pi$BO | BOHB | PriorBand+BO |
| **JAHS-C10** | $10.049 \pm 0.983$ | $\mathbf{9.52 \pm 0.427}$ | $10.281 \pm 0.88$ | $9.887 \pm 0.39$ | $9.451 \pm 0.806$ | $\mathbf{9.222 \pm 0.287}$ | $9.969 \pm 0.688$ | $9.706 \pm 0.416$ |
| **JAHS-CH** | $5.743 \pm 0.482$ | $5.632 \pm 0.323$ | $5.628 \pm 0.613$ | $\mathbf{5.56 \pm 0.51}$ | $\mathbf{5.158 \pm 0.387}$ | $5.439 \pm 0.256$ | $5.393 \pm 0.442$ | $5.393 \pm 0.417$ |
| **JAHS-FM** | $5.257 \pm 0.245$ | $5.038 \pm 0.015$ | $5.406 \pm 0.323$ | $\mathbf{4.996 \pm 0.091}$ | $5.047 \pm 0.205$ | $4.965 \pm 0.063$ | $5.227 \pm 0.255$ | $\mathbf{4.964 \pm 0.114}$ |
| **LC-126026** | $0.049 \pm 0.007$ | $0.044 \pm 0.003$ | $0.043 \pm 0.009$ | $\mathbf{0.039 \pm 0.006}$ | $0.04 \pm 0.008$ | $0.04 \pm 0.003$ | $0.038 \pm 0.008$ | $\mathbf{0.036 \pm 0.006}$ |
| **LC-167190** | $0.204 \pm 0.019$ | $0.209 \pm 0.017$ | $0.191 \pm 0.022$ | $\mathbf{0.184 \pm 0.027}$ | $0.175 \pm 0.023$ | $0.203 \pm 0.018$ | $0.175 \pm 0.023$ | $\mathbf{0.169 \pm 0.026}$ |
| **LC-168330** | $0.412 \pm 0.032$ | $0.404 \pm 0.029$ | $0.412 \pm 0.032$ | $\mathbf{0.374 \pm 0.051}$ | $0.361 \pm 0.04$ | $0.366 \pm 0.039$ | $0.395 \pm 0.033$ | $\mathbf{0.344 \pm 0.052}$ |
| **LC-168910** | $0.318 \pm 0.028$ | $0.313 \pm 0.024$ | $0.31 \pm 0.033$ | $\mathbf{0.302 \pm 0.032}$ | $0.295 \pm 0.03$ | $0.301 \pm 0.022$ | $0.297 \pm 0.025$ | $\mathbf{0.289 \pm 0.035}$ |
| **LC-189906** | $0.153 \pm 0.02$ | $0.148 \pm 0.011$ | $0.141 \pm 0.014$ | $\mathbf{0.126 \pm 0.016}$ | $0.124 \pm 0.013$ | $0.141 \pm 0.009$ | $0.13 \pm 0.011$ | $\mathbf{0.115 \pm 0.014}$ |
| **PD1-Cifar100** | $0.282 \pm 0.039$ | $0.25 \pm 0.014$ | $0.291 \pm 0.039$ | $\mathbf{0.248 \pm 0.015}$ | $0.242 \pm 0.028$ | $\mathbf{0.23 \pm 0.017}$ | $0.262 \pm 0.03$ | $0.234 \pm 0.016$ |
| **PD1-ImageNet** | $0.266 \pm 0.029$ | $\mathbf{0.223 \pm 0.005}$ | $0.289 \pm 0.035$ | $0.223 \pm 0.003$ | $0.237 \pm 0.017$ | $\mathbf{0.213 \pm 0.009}$ | $0.278 \pm 0.029$ | $0.22 \pm 0.008$ |
| **PD1-LM1B** | $0.663 \pm 0.016$ | $0.65 \pm 0.003$ | $0.643 \pm 0.014$ | $\mathbf{0.641 \pm 0.011}$ | $0.648 \pm 0.013$ | $0.646 \pm 0.006$ | $\mathbf{0.634 \pm 0.014}$ | $0.636 \pm 0.011$ |
| **PD1-WMT** | $0.391 \pm 0.018$ | $0.372 \pm 0.003$ | $0.382 \pm 0.022$ | $\mathbf{0.367 \pm 0.014}$ | $0.372 \pm 0.014$ | $0.368 \pm 0.009$ | $0.363 \pm 0.017$ | $\mathbf{0.356 \pm 0.018}$ |
| | **Bad Prior** | | | | | | | |
| | BO | $\pi$BO | BOHB | PriorBand+BO | BO | $\pi$BO | BOHB | PriorBand+BO |
| **JAHS-C10** | $\mathbf{10.132 \pm 1.143}$ | $10.24 \pm 1.242$ | $10.281 \pm 0.88$ | $10.213 \pm 0.846$ | $\mathbf{9.354 \pm 0.722}$ | $9.534 \pm 0.857$ | $9.969 \pm 0.688$ | $9.716 \pm 0.597$ |
| **JAHS-CH** | $5.874 \pm 0.803$ | $6.572 \pm 1.379$ | $5.628 \pm 0.613$ | $\mathbf{5.608 \pm 0.582}$ | $\mathbf{5.316 \pm 0.537}$ | $6.204 \pm 1.057$ | $5.393 \pm 0.442$ | $5.429 \pm 0.531$ |
| **JAHS-FM** | $5.323 \pm 0.38$ | $5.425 \pm 0.343$ | $5.406 \pm 0.323$ | $\mathbf{5.261 \pm 0.291}$ | $\mathbf{5.088 \pm 0.227}$ | $5.212 \pm 0.276$ | $5.227 \pm 0.255$ | $5.138 \pm 0.199$ |
| **LC-126026** | $0.051 \pm 0.007$ | $0.059 \pm 0.009$ | $0.043 \pm 0.009$ | $\mathbf{0.039 \pm 0.01}$ | $0.042 \pm 0.008$ | $0.051 \pm 0.005$ | $0.038 \pm 0.008$ | $\mathbf{0.034 \pm 0.01}$ |
| **LC-167190** | $0.21 \pm 0.024$ | $0.228 \pm 0.024$ | $0.191 \pm 0.022$ | $\mathbf{0.185 \pm 0.028}$ | $0.184 \pm 0.026$ | $0.216 \pm 0.019$ | $0.175 \pm 0.023$ | $\mathbf{0.163 \pm 0.024}$ |
| **LC-168330** | $0.427 \pm 0.043$ | $0.463 \pm 0.042$ | $0.412 \pm 0.032$ | $\mathbf{0.395 \pm 0.04}$ | $\mathbf{0.35 \pm 0.052}$ | $0.429 \pm 0.04$ | $0.395 \pm 0.033$ | $0.366 \pm 0.053$ |
| **LC-168910** | $0.323 \pm 0.028$ | $0.321 \pm 0.045$ | $0.31 \pm 0.033$ | $\mathbf{0.304 \pm 0.032}$ | $0.291 \pm 0.028$ | $0.29 \pm 0.012$ | $0.297 \pm 0.025$ | $\mathbf{0.288 \pm 0.029}$ |
| **LC-189906** | $0.217 \pm 0.076$ | $0.365 \pm 0.139$ | $0.141 \pm 0.014$ | $\mathbf{0.126 \pm 0.019}$ | $0.129 \pm 0.016$ | $0.185 \pm 0.057$ | $0.13 \pm 0.011$ | $\mathbf{0.115 \pm 0.017}$ |
| **PD1-Cifar100** | $0.408 \pm 0.155$ | $0.586 \pm 0.195$ | $0.291 \pm 0.039$ | $\mathbf{0.282 \pm 0.054}$ | $0.251 \pm 0.031$ | $0.336 \pm 0.164$ | $0.262 \pm 0.03$ | $\mathbf{0.241 \pm 0.021}$ |
| **PD1-ImageNet** | $0.355 \pm 0.135$ | $0.619 \pm 0.188$ | $0.289 \pm 0.035$ | $\mathbf{0.285 \pm 0.032}$ | $0.26 \pm 0.109$ | $0.41 \pm 0.233$ | $0.278 \pm 0.029$ | $\mathbf{0.249 \pm 0.025}$ |
| **PD1-LM1B** | $0.688 \pm 0.034$ | $0.782 \pm 0.06$ | $\mathbf{0.643 \pm 0.014}$ | $0.644 \pm 0.013$ | $0.653 \pm 0.015$ | $0.667 \pm 0.03$ | $\mathbf{0.634 \pm 0.014}$ | $0.636 \pm 0.012$ |
| **PD1-WMT** | $0.523 \pm 0.124$ | $0.751 \pm 0.099$ | $0.382 \pm 0.022$ | $\mathbf{0.38 \pm 0.025}$ | $0.375 \pm 0.024$ | $0.497 \pm 0.133$ | $0.363 \pm 0.017$ | $\mathbf{0.361 \pm 0.018}$ |

Table 14: Table comparing Asynchronous-HyperBand+BO (Mobster), Mobster+ESP and `PriorBand`+BO final validation errors of the current incumbent at the highest fidelity at 2 budget horizons of $1\times$ and $5\times$. Runs are averaged over $10$ seeds where each run is with $4$ workers. This table shows the extensibility of ESP to asynchronous model-based HB and also verifies the effectiveness of running `PriorBand` in a parallel setting. The key differences between Mobster+ESP and `PriorBand`+BO are their choice of initial design and the nature of multi-fidelity scheduling.

| Benchmark | 1x | | | 5x | | |
|---|---|---|---|---|---|---|
| **Good Prior** | | | | | | |
| | Mobster | Mobster(+ESP) | PriorBand+BO | Mobster | Mobster(+ESP) | PriorBand+BO |
| **JAHS-C10** | $12.192 \pm 1.558$ | $\mathbf{8.281 \pm 0.221}$ | $8.281 \pm 0.221$ | $9.872 \pm 0.579$ | $\mathbf{8.18 \pm 0.177}$ | $8.281 \pm 0.221$ |
| **JAHS-CH** | $6.771 \pm 1.296$ | $\mathbf{4.466 \pm 0.188}$ | $4.466 \pm 0.188$ | $5.565 \pm 0.342$ | $4.466 \pm 0.188$ | $\mathbf{4.424 \pm 0.169}$ |
| **JAHS-FM** | $6.172 \pm 1.015$ | $\mathbf{4.713 \pm 0.052}$ | $4.713 \pm 0.052$ | $5.177 \pm 0.279$ | $\mathbf{4.713 \pm 0.052}$ | $4.713 \pm 0.052$ |
| **PD1-Cifar100** | $0.434 \pm 0.172$ | $0.259 \pm 0.083$ | $\mathbf{0.243 \pm 0.056}$ | $0.252 \pm 0.029$ | $\mathbf{0.217 \pm 0.033}$ | $0.217 \pm 0.033$ |
| **PD1-ImageNet** | $0.338 \pm 0.056$ | $0.261 \pm 0.06$ | $\mathbf{0.258 \pm 0.053}$ | $0.249 \pm 0.023$ | $\mathbf{0.214 \pm 0.017}$ | $0.234 \pm 0.029$ |
| **PD1-LM1B** | $0.663 \pm 0.012$ | $\mathbf{0.639 \pm 0.018}$ | $0.642 \pm 0.022$ | $0.647 \pm 0.01$ | $0.634 \pm 0.012$ | $\mathbf{0.63 \pm 0.018}$ |
| **PD1-WMT** | $0.45 \pm 0.065$ | $0.368 \pm 0.04$ | $\mathbf{0.358 \pm 0.032}$ | $0.374 \pm 0.015$ | $\mathbf{0.348 \pm 0.023}$ | $0.353 \pm 0.027$ |
| **Good Prior** | | | | | | |
| | Mobster | Mobster(+ESP) | PriorBand+BO | Mobster | Mobster(+ESP) | PriorBand+BO |
| **JAHS-C10** | $12.192 \pm 1.558$ | $\mathbf{10.194 \pm 0.0}$ | $10.194 \pm 0.0$ | $9.952 \pm 0.637$ | $\mathbf{9.297 \pm 0.39}$ | $9.609 \pm 0.576$ |
| **JAHS-CH** | $6.768 \pm 1.295$ | $\mathbf{6.603 \pm 0.0}$ | $6.603 \pm 0.0$ | $5.668 \pm 0.371$ | $5.492 \pm 0.312$ | $\mathbf{5.321 \pm 0.553}$ |
| **JAHS-FM** | $6.245 \pm 0.999$ | $\mathbf{5.042 \pm 0.0}$ | $5.042 \pm 0.0$ | $5.134 \pm 0.274$ | $5.013 \pm 0.06$ | $\mathbf{5.008 \pm 0.063}$ |
| **PD1-Cifar100** | $0.434 \pm 0.172$ | $\mathbf{0.259 \pm 0.0}$ | $0.259 \pm 0.0$ | $0.252 \pm 0.029$ | $\mathbf{0.226 \pm 0.017}$ | $0.234 \pm 0.02$ |
| **PD1-ImageNet** | $0.333 \pm 0.056$ | $\mathbf{0.224 \pm 0.0}$ | $0.224 \pm 0.0$ | $0.244 \pm 0.013$ | $\mathbf{0.222 \pm 0.005}$ | $0.224 \pm 0.0$ |
| **PD1-LM1B** | $0.663 \pm 0.012$ | $0.651 \pm 0.0$ | $\mathbf{0.641 \pm 0.009}$ | $0.649 \pm 0.009$ | $0.646 \pm 0.008$ | $\mathbf{0.635 \pm 0.011}$ |
| **PD1-WMT** | $0.45 \pm 0.065$ | $\mathbf{0.372 \pm 0.0}$ | $0.372 \pm 0.0$ | $0.377 \pm 0.017$ | $0.37 \pm 0.006$ | $\mathbf{0.358 \pm 0.02}$ |
| **Bad Prior** | | | | | | |
| | Mobster | Mobster(+ESP) | PriorBand+BO | Mobster | Mobster(+ESP) | PriorBand+BO |
| **JAHS-C10** | $12.375 \pm 1.474$ | $13.857 \pm 3.841$ | $\mathbf{10.252 \pm 1.069}$ | $9.792 \pm 0.624$ | $9.744 \pm 0.643$ | $\mathbf{9.614 \pm 0.622}$ |
| **JAHS-CH** | $\mathbf{6.768 \pm 1.295}$ | $9.218 \pm 2.769$ | $7.291 \pm 1.25$ | $5.668 \pm 0.371$ | $5.56 \pm 0.707$ | $\mathbf{5.407 \pm 0.483}$ |
| **JAHS-FM** | $\mathbf{6.245 \pm 0.999}$ | $7.244 \pm 3.419$ | $6.845 \pm 3.545$ | $\mathbf{5.134 \pm 0.274}$ | $5.181 \pm 0.266$ | $5.17 \pm 0.227$ |
| **PD1-Cifar100** | $\mathbf{0.434 \pm 0.172}$ | $0.678 \pm 0.195$ | $0.657 \pm 0.266$ | $0.252 \pm 0.029$ | $0.258 \pm 0.025$ | $\mathbf{0.244 \pm 0.03}$ |
| **PD1-ImageNet** | $0.333 \pm 0.056$ | $0.558 \pm 0.155$ | $\mathbf{0.328 \pm 0.035}$ | $\mathbf{0.244 \pm 0.013}$ | $0.248 \pm 0.028$ | $0.266 \pm 0.026$ |
| **PD1-LM1B** | $0.663 \pm 0.012$ | $0.699 \pm 0.034$ | $\mathbf{0.661 \pm 0.022}$ | $0.649 \pm 0.009$ | $0.657 \pm 0.006$ | $\mathbf{0.639 \pm 0.01}$ |
| **PD1-WMT** | $\mathbf{0.45 \pm 0.065}$ | $0.609 \pm 0.16$ | $0.47 \pm 0.088$ | $0.375 \pm 0.013$ | $0.372 \pm 0.017$ | $\mathbf{0.371 \pm 0.016}$ |