# OpenReview forum: "PriorBand: Practical Hyperparameter Optimization in the Age of Deep Learning"
_NeurIPS.cc/2023/Conference — NeurIPS 2023 poster_

### Official Review · Reviewer_meuK · 2023-06-30

**Soundness:** 2 fair
**Presentation:** 2 fair
**Contribution:** 3 good
**Rating:** 5
**Confidence:** 4

**Summary:**

This paper presents a method to enhance the random sampling component of hyperband. The authors propose replacing it with a combination of random sampling, prior-based sampling, and incumbent-based sampling. They also suggest adjusting the proportion of these samplers based on the current state in the hyperparameter optimization process. The authors perform experiments on a series of benchmarks and compare the proposed method with multiple classical HBO baselines.

**Strengths:**

This work might be useful in some particular situations.

**Weaknesses:**

- The motivation for this work may appear artificial, aiming to reduce the cost of hyperparameter optimization in the age of deep learning. However, the paper lacks a detailed explanation of how the proposed adjustments to the sampling component can make hyperparameter optimization more practical and cost-effective in the era of deep learning. Can the authors provide a more comprehensive reasoning process to support this claim?

- Technically, this method is essentially a combination of prior work, including multi-fidelity optimization [1], expert priors [2], and local search [3]. The authors' contributions primarily build upon and benefit from these existing approaches, rather than introducing original ideas. The authors declare that their approach fulfills all the desired requirements for application to deep learning, but this claim is essentially derived from the benefits provided by multi-fidelity optimization.

[1] Li, L., Jamieson, K., DeSalvo, G., Rostamizadeh, A., & Talwalkar, A. (2017). Hyperband: A novel bandit-based approach to hyperparameter optimization. The Journal of Machine Learning Research, 18(1), 6765-6816.

[2]Hvarfner, C., Stoll, D., Souza, A., Lindauer, M., Hutter, F., & Nardi, L. (2022). $\pi $ BO: Augmenting acquisition functions with user beliefs for bayesian optimization. arXiv preprint arXiv:2204.11051.

[3]Wu, Q., Wang, C., & Huang, S. (2021, May). Frugal optimization for cost-related hyperparameters. In Proceedings of the AAAI Conference on Artificial Intelligence (Vol. 35, No. 12, pp. 10347-10354).

**Questions:**

Why modifying the random sampling component of HyperBand can help decrease the expenses associated with tuning deep learning models?

**Limitations:**

As a hyperparameter optimization algorithm, the proposed method encompasses several hyperparameters that may significantly influence its final outcomes. These include the hyperparameters associated with the perturbation operation and those control the proportion of the three sampling methods. However, there is currently a dearth of corresponding ablation studies that specifically investigate the individual contributions of these hyperparameters.

---

> ### Author Rebuttal · Authors · 2023-08-08
>
> We thank you for your comments.
>
> We would request you elaborate on your thoughts on the situations in which our work could be useful. Understanding this could give us perspective on how to address your concerns.
>
> ---
> > The motivation for this work may appear artificial...the paper lacks a detailed explanation of how the proposed adjustments to the sampling component can make HPO more practical and cost-effective in the era of deep learning. Can the authors provide a more comprehensive reasoning process to support this claim?
>
> > Why modifying the random sampling component of HyperBand (HB) can help decrease the expenses associated with tuning deep learning models?
> ---
>
> 1.
>   **a)** We respectfully disagree with the statement of artificial motivation. In many cases, human experts have strong knowledge about hyperparameter settings that are likely to perform well. E.g., if you’re optimizing GPT-4, you do not run HyperBand (HB), but you rather reuse previously known good hyperparameters, maybe with small local adjustments. PriorBand exploits the same prior sampling approach and marries it with multi-fidelity optimization.
>
>   **b)** In the original Hyperband paper, the following is stated in Future Work:
>     "_Finally, **Hyperband can benefit from different sampling schemes aside from simple random search** ... meta-learning can be used to **define intelligent priors informed by previous experimentation**.  Finally, as mentioned in Section 2, exploring ways to combine Hyperband with **adaptive configuration selection strategies** is a very promising future direction._"
>
>   As such, the original authors share the opinion that this is a good direction for increased performance; the results confirm this hypothesis.
>
>   **c)** _Fig. 2_ shows that augmenting HB’s random sampling with prior sampling improves anytime performance. To account for failure modes, a local search around the incumbent is introduced as the third sampler, and a dynamic weighting algorithm is designed to trade off the 3 samplers. _Fig. 1_ illustrates the improved anytime performance of this method. _Section 7_ and _Appendix F_ show strong anytime performance for PriorBand under good priors. _Tables 10-14_ show per-dataset performance gains even under budgets equivalent to 4-5 model training.
>
>   **d)** Improving anytime performance directly translates to lower compute requirements for HPO, allowing practitioners to find suitable model configurations at lower costs than previously existing methods. Our work thus contributes to making HPO practical for DL.
>
>   **e)** Our experiment design illustrates results for a maximum budget of 5 model training when 4 parallel model training are possible. Whether such a budget is feasible for HPO is heavily subjective. However, we believe it captures tractable hardware and compute resources in many more cases than previously possible.
>
>
> ---
> > Technically, this method is essentially a combination of prior work, including multi-fidelity optimization [1], expert priors [2], and local search [3].
> > The authors declare that their approach fulfills all the desired requirements for application to deep learning, but this claim is essentially derived from the benefits provided by multi-fidelity optimization.
> ---
>
> 2.
>   **a)** In the realm of HPO, Multi-fidelity Optimization (MFO) and expert priors represent two key paradigms, with HB [1] and $\pi$BO [2] serving as their respective representatives in our work. Neither one of these in isolation is sufficient to fulfill all desired requirements. To our knowledge, there is no existing work which efficiently merges these two paradigms. Our contribution, the Ensemble Sampling Policy (ESP), is the first to propose a general yet systematic approach to doing so robustly, allowing for multi-fidelity algorithms to interface with expert priors and fulfill the required desiderata. Moreover, the ESP enables an explicit expert prior interface to an entire family of multi-fidelity algorithms.
>
>   **b)** The referenced paper [3] can be integrated into our related literature, thank you. Our proposed method’s contribution over [3] is that we are agnostic to the exact form of local search used. Sampling from the unit sphere [3] aligns with one of our local search ablations ("hypersphere" in _Fig. 16_), but we found it to exhibit higher variance and to be less robust than the local mutation in PriorBand. Sampling from a neighborhood sphere offered more hyperparameters and issues mentioned in _L811-816 (Appendix E.2.3)_.
>
>
> ---
> > As a HPO algorithm, the proposed method encompasses several hyperparameters … a dearth of corresponding ablation studies that specifically investigate the individual contributions of these hyperparameters.
> ---
>
> 3. In Fig. 3 of the rebuttal PDF, we have added ablation studies on the local search hyperparameters (Appendix E.2.5) as pointed out. We ablate over the standard deviation of the Gaussian around the incumbent configuration and the mutation rate, that is, the probability of selection of a hyperparameter for perturbation. Our chosen default setting for these hyperparameters, which we keep fixed across all our experiments, is amongst the better choices, but performance is very robust to these hyperparameters.
>
>
> ### ___
>
> We hope that our explanations were satisfactory and brought about more clarity. If so, we would appreciate it if you consider increasing our score. If you have any further questions and comments we would be very glad to discuss them.
>
> ### References
>
> [1] Li et al. (2017). Hyperband: A novel bandit-based approach to hyperparameter optimization. The Journal of Machine Learning Research, 18(1), 6765-6816.
>
> [2] Hvarfner et al. (2022). $\pi$BO: Augmenting acquisition functions with user beliefs for Bayesian Optimization. arXiv preprint arXiv:2204.11051.
>
> [3] Wu et al. (2021, May). Frugal optimization for cost-related hyperparameters. In Proceedings of the AAAI Conference on Artificial Intelligence (Vol. 35, No. 12, pp. 10347-10354).

---

> > ### Comment · Reviewer_meuK · 2023-08-18
> >
> > I think most of my concerns are addressed. I will raise my score. Thanks for your comments.

---

> > > ### Author Response · Authors · 2023-08-18
> > > **Re: Official Comment by Reviewer meuK**
> > >
> > > We thank you for your revision; it is much appreciated.
> > >
> > > We are eager to address any remaining concerns you may have, in order to enhance your confidence in our work.
> > >
> > > We also welcome feedback to improve the final draft.

---

### Official Review · Reviewer_zHth · 2023-07-04

**Soundness:** 3 good
**Presentation:** 3 good
**Contribution:** 2 fair
**Rating:** 7
**Confidence:** 4

**Summary:**

This paper proposes PriorBand, an extension of HyperBand that adds expert priors and a novel sampling technique to replace random sampling in HB, called the Ensemble Sampling Policy (ESP). The ESP allows the algorithm to lean on the expert prior, but also use the current incumbent in case the prior is non-optimal. Under good and bad priors, the authors demonstrate the efficacy of this approach, showing additionally that the ESP also improves the performance of other Successive Halving (SH)-based methods.

**Rebuttal:** The authors have clearly addressed my comments in their rebuttal. I have updated my score from 5 to 7.

**Strengths:**

- My favorite part of this paper is Section 7.2, which shows that the ESP proposed in this paper extends to other SH-like HPO methods as well. This contribution should perhaps be highlighted more.
- The paper's verbiage is very clear, and examples presented in the introductory sections are useful aids to readers who may not be familiar with HyperBand's workings.

**Weaknesses:**

- My major concern is in the use of average relative rank to demonstrate efficacy. While a lower rank indicates better performance, it does not indicate whether that performance delta is statistically significant. As opposed to confidence intervals over *ranks*, I would much rater see the actual improvement in per-dataset metrics. For example, the original HyperBand paper [1] showed the test error over wall time for multiple HPO algorithms.
- minor: In L130, you're missing a period.
- L182: typo: modelling --> modeling

[1] Li, L., Jamieson, K., DeSalvo, G., Rostamizadeh, A., & Talwalkar, A. (2017). Hyperband: A novel bandit-based approach to hyperparameter optimization. The journal of machine learning research, 18(1), 6765-6816.

**Questions:**

- In Alg. 2, L5, can you explain why $p_\pi$ is used instead of $p_{\hat{\lambda}}$?
- In Fig. 14, it seems the lines have confidence intervals. How are these computed, and how many repeats of the experiment were used?
- In Appendix D.3, the authors describe the way a "good" prior is generated, using the best of 25 configurations. In practice, this would add computational overhead to the overall process of training a model with good hyper-parameters. In that sense, perhaps it should be emphasized that the Bad prior results are more useful to a practitioner, who, following their intuition/expertise/random choice, might possibly use a non-optimal prior (i.e., without running anything). Can you comment on this?

**Limitations:**

The authors have addressed the limitations of their work appropriately in the main text and the supplementary materials.

---

> ### Author Rebuttal · Authors · 2023-08-08
>
> We are glad to hear that the reviewer appreciated our presentation and found a favorite part too! Thank you for your review and comments.
>
> ---
> > My major concern is in the use of average relative rank to demonstrate efficacy. While a lower rank indicates better performance, it does not indicate whether that performance delta is statistically significant. As opposed to confidence intervals over ranks, I would much rather see the actual improvement in per-dataset metrics. For example, the original HyperBand paper showed the test error over wall time for multiple HPO algorithms.
> ---
>
> 1. Thank you for raising this important point. We agree that relative ranks only show part of the picture, and we used it throughout for ease of aggregation across far more benchmarks than, e.g., the Hyperband paper, for additional robustness of results. Aggregating raw performance across benchmarks has to be done carefully to avoid individual benchmarks being over-represented due to different scales of performances; we now did this using average normalized regret. Fig. 1 and Fig. 2 in the _rebuttal PDF_ include per-dataset regret plots for good priors and average normalized regret for different prior strengths, respectively. Due to page restrictions, in the main paper, we can only include either the aggregated average rank or the average regret. Your suggestion on how to effectively communicate and persuade the reader with Appendix support is welcome, but we will make sure to add regret plots in a camera-ready version, including a pair (good-bad priors) of per-dataset normalized regret plots.
>
> ---
> > In Alg. 2, L5, can you explain why $p_\pi$ is used instead of $p_{\hat{\lambda}}$?
> ---
>
> 2. We acknowledge any potential confusion that may have arisen from our use of $p_\pi$ in Alg. 2. We shall rename $p_\pi$ in L1 and RHS of L5-6 in Alg. 2 to $p_\pi^{\mathrm{old}}$ in the camera-ready.
>
>      L3 in Alg. 1 computes the probability of sampling uniformly random and the remaining probability is assigned to $p_\pi$. The role of Alg. 2 now (taking the current $p_\pi$ as input) is to split this probability between the prior sampler and the incumbent-based sampler. L5 in Alg. 2 thus assigns the determined proportion of this probability to $p_{\hat{\lambda}}$ and in L6 $p_\pi$ is updated as well.
>
> Please let us know if this clarifies the confusion or whether you have any further questions.
>
>
> ---
> > In Appendix D.3, the authors describe the way a "good" prior is generated, using the best of 25 configurations. In practice, this would add computational overhead to the overall process of training a model with good hyper-parameters. In that sense, perhaps it should be emphasized that the Bad prior results are more useful to a practitioner, who, following their intuition/expertise/random choice, might possibly use a non-optimal prior (i.e., without running anything). Can you comment on this?
> ---
>
> 3. This is a crucial misunderstanding; please let us explain.
>
>     **a)** The best-of-25-samples methodology is strictly an experimental setup protocol to obtain a sense of what a good configuration is. That is, we are not suggesting running 25 random configurations before running PriorBand (with that budget we want to long be done with the optimization), but rather we assume that a practitioner may have existing knowledge based on earlier tuning experience or the literature. (If you’re, e.g., optimizing GPT-4, you use your knowledge gathered from GPT-2 and GPT-3.) For our experimental evaluation, we require a simple, general and reproducible protocol and therefore followed prior work on Bayesian optimization with expert priors [1]. However, we acknowledge that this type of prior may not perfectly match a practitioner’s prior. In _Appendix D.3, Fig 13_, we show the performance of random samples produced from these priors to help judge their relative quality.
>
>     **b**) We agree that the issue of prior quality is a relevant one. _Tab. 10_ displays PriorBand’s performance across prior strengths, compared to other algorithms. PriorBand’s performance degrades gracefully even in the face of a highly adversarial prior (the worst of 50000 random samples), as it maintains the performance of HyperBand on a 12-evaluation budget and only lags behind slightly on a 5-evaluation budget. This adversarial prior is highly unrealistic; if a user doesn’t know which hyperparameter values would work well they can always specify a uniform prior, in which case prior samples are additional random samples.
>
>   Given the prevalence of manual tuning, we ultimately believe that practitioners are substantially more well-informed than a vanilla (random) approach. As such, we believe that they should generally have a positive influence on PriorBand’s performance.
>
> ---
> > In Fig. 14, it seems the lines have confidence intervals. How are these computed, and how many repeats of the experiment were used?
> ---
> 4. We would like to refer you to _Section 6.3_, _Appendices D.4_ and _D.5_ where we detail our experimental setup. We hope this addresses your question about Fig. 14. If not, we are happy to clarify further.
>
> ### ___
>
> We appreciate your feedback and believe to have addressed all the points you raised. Given our additional results for your main concern of only showing rank-based results and our clarification of the crucial misunderstanding of the source of priors (25 random points being far beyond our total budget), we would greatly appreciate it if you were to increase your score correspondingly. If you have additional questions we would be more than glad to discuss them.
>
> We shall certainly fix the typos you pointed out, in the final draft.
>
> ### References:
>
> [1] Hvarfner et al. (2022). $\pi$BO: Augmenting acquisition functions with user beliefs for Bayesian Optimization. arXiv preprint arXiv:2204.11051.

---

> ### Comment · Reviewer_zHth · 2023-08-10
>
> The authors have sufficiently addressed my major concern with the paper. I have accordingly revised my score from 5/4 to 7/4.

---

> > ### Author Response · Authors · 2023-08-14
> >
> > We appreciate and thank you for the revision!
> >
> > Any general comment for improving a camera-ready draft is most welcome!

---

### Official Review · Reviewer_QGiN · 2023-07-05

**Soundness:** 3 good
**Presentation:** 4 excellent
**Contribution:** 2 fair
**Rating:** 6
**Confidence:** 4

**Summary:**

In the paper "PriorBand: Practical Hyperparameter Optimization in the Age of Deep Learning" the authors propose an extension to the well-known HPO methods Hyperband by adapting the way how candidate hyperparameter settings are sampled. To this end, the authors propose to use a weighting mechanism to balance between three sampling distributions: random, locally random close to the incumbent, and a prespecified prior. The latter allows experts to inject beliefs about the optimum. In the empirical study PriorBand is found to perform best on average among the considered methods.

**Strengths:**

+ PriorBand compares favorably to its competitors and allows for hyperparameter optimization with comparably low budgets.
+ The paper is very well written and the presentation in general is excellent.
+ The authors put very much effort into making their work reproducible, openly accessible and easy to understand.
+ For the self-imposed desiderata, PriorBand offers the most desired properties.

**Weaknesses:**

- The work is pretty incremental. The only original contribution of this work is to weight the different probability distributions from which candidates are sampled.
- It is not clear what effect the incumbent-sampling has on the overall performance. At least I could not find any ablation studies in this regard. Is it really necessary to include incumbent-sampling? What would happen if only random sampling is balanced against prior-sampling? Is it then harder to retain decent performance in the light of bad priors?
- Nothing is stated about the impact on theoretical guarantees that are known to hold for Hyperband. Does the theoretical framework of Hyperband still apply for the changed distributions?
- When comparing different hyperparameter optimizers only relative ranks are

**Questions:**

- What is the individual effect of incumbent-based and prior-based sampling?
- What is the impact on the theoretical guarantees for Hyperband? Are all the assumptions still fulfilled?
- What is the definition of "relative rank"? Is it only a rank or does it also include performance differences?
- Where do these desiderata stem from? Although they all seem intuitive it is not clear whether they are exhaustive and what the coverage is like.

**Limitations:**

Limitations of PriorBand are well discussed. However, it is not entirely clear to what extent the desiderata for HPO in the age of deep learning is limited or maybe even exhaustive.

---

> ### Author Rebuttal · Authors · 2023-08-08
>
> We thank the reviewer for their comments and for appreciating the strong performance of PriorBand, our paper presentation and our efforts to make research open-source and reproducible.
>
> ---
> > What is the individual effect of incumbent-based and prior-based sampling?
> ---
> 1.
>   **a)** We apologize that this was not sufficiently clear from the main paper. We actually do report multiple ablation studies on the individual components of PriorBand in the appendix. The incumbent-based ablation can be found in _Appendix E.2, Fig. 17_, showing that, for any prior quality and fidelity correlation, the incumbent-based sampling improves performance. _Appendix E.3_, _Fig. 19_ also highlights the role of how the weights of each sampler adapt to lend robustness to PriorBand.
>
>   **b)**  The effect of prior-based sampling can be found in _Fig. 1_ and _Fig. 2_, (as a substitute to RS) and in _Fig. 5_ (the effect of good/bad priors on the outcome of optimization) with complementary results in _Fig. 21_ and _Fig. 22_. While a bad prior naturally degrades performance initially, its long-run impact is negligible compared to RS.
> If the reviewer has additional ablations in mind, we would be happy to add them to the camera-ready.
>
> ---
> > The work is pretty incremental. The only original contribution of this work is to weight the different probability distributions from which candidates are sampled.
> ---
> 2. While your observation holds merit, we believe our contribution facilitates new directions in hyperparameter optimization (HPO), adding novelty to the existing literature
>
>     Our work is the first, to our knowledge, to empower experts in enhancing multi-fidelity optimization through their knowledge and intuition. It's principled, well-motivated, with thorough ablations, enabling effective model-based extensions. Based on the gap it identifies and fills, we believe our work is novel.
>
>     We would also like to mention that the established HPO algorithms we compare to may be considered to be incremental: ASHA is a simple if-condition different from SuccessiveHalving, BOHB is Bayesian Optimization (BO) as a sampler in HyperBand (HB), etc. However, they all provide unique benefits and are extensively used algorithms.
>
>     PriorBand adds to this list by uniquely supporting expert priors, additionally empowering _all_ such algorithms to interface with expert priors through our contribution of the Ensemble Sample Policy (ESP), making our approach novel in scope, design, and application. Moreover, its robustness and simplicity make it extremely practical - a largely overlooked criterion in the BO/HPO community.
>
> ---
> > Nothing is stated about the impact on theoretical guarantees that are known to hold for HB. Does the theoretical framework of HB still apply to the changed distributions?
>
> > What is the impact on the theoretical guarantees for HB? Are all the assumptions still fulfilled?
> ---
> 3.
>   **a)** Thank you, we would like to clarify that PriorBand ensures a constant proportion of random sampling at the highest fidelity (_Eq. 2_) and thus can trivially be proven to be no more than a constant factor worse than HB in the worst case. We did not add this theorem due to its simplicity and since the important practical reductions of anytime regret over HB come from the local search component and the user prior, NOT through random sampling, any theoretical results for these speedups remain elusive.
> We are hopeful that we can apply a multi-armed bandit formulation to the selection of sampling ratios in the future to obtain a regret bound compared to the best of the candidate sampling strategies.
>
>   **b)** While it's feasible to apply HB's multi-armed bandit formulation to PriorBand by considering the local search as a form of exploitation, we believe this topic deserves its own dedicated paper. As such, we consider it as potential future work.
>
> ---
> > When comparing different hyperparameter optimizers only relative ranks are _(used?)_
>
> > What is the definition of "relative rank"? Is it only a rank or does it also include performance differences?
> ---
> 4.
>   **a)** Our ranking procedure is aggregated to show robustness across benchmarks and is described in detail in _Appendix D.5_.
>
>   We agree that ranking plots only show part of the picture and therefore have now also uploaded plots showing average normalized regret in our _rebuttal PDF_, whose conclusions remain the same.
>
>   **b)** It ranks algorithms based on their best solutions at a given time. We compare these rankings across benchmarks for one seed, average across benchmarks for robustness, and repeat across seeds to calculate mean and standard errors for overall ranking.
> ---
> > Where do these desiderata stem from? Although they all seem intuitive it is not clear whether they are exhaustive and what the coverage is like.
>
> > ...it is not entirely clear to what extent the desiderata for HPO in the age of deep learning is limited or maybe even exhaustive.
> ---
> 5. Our desiderata are extending the desiderata stated in the influential paper on BOHB, Falkner et al. [1] which combines BO and HB, in combination with recent insights [2, 3] that manual search is preferred to (more sophisticated) HPO. When HPO is adopted, simple, yet effective, algorithms like HB are preferred. As such, “Expert Beliefs” and “Simplicity” were well-grounded additions to that list. We believe our list is thorough, but we invite the reviewer to suggest anything that might be missing to make it truly exhaustive.
>
> ### ___
> We'd appreciate knowing if our responses have satisfied you and if our clarified perspective might improve your evaluation. Any further questions, comments or suggestions for enhancing the paper are most welcome.
>
> ### References:
>
> [1] Falkner et al., BOHB: Robust and Efficient Hyperparameter Optimization at Scale, 2018
>
> [2] Bouthillier and Varoquaux, Survey of machine-learning experimental methods at NeurIPS2019 and ICLR2020, 2020
>
> [3] Schneider et al., HITY workshop poll, NeurIPS, 2022

---

> > ### Comment · Reviewer_QGiN · 2023-08-18
> > **Re: Rebuttal by Authors**
> >
> > Thank you very much for the thorough rebuttal and the clarifications. Most of my points are reasonably addressed in your comments and revisions.
> >
> > However, the aspect of being incremental remains and I do not find the argument that there exist other incremental works very convincing. Furthermore, I do not consider myself a deep learning practitioner, therefore I unfortunately need to decline the invitation of providing more desiderata. Yet, I would wish for a survey among practitioners, on what they would actually desire from HPO methods to incorporate them into their daily routine when working with such expensive training processes. I would agree with the authors that reducing the cost for HPO might indeed be a key point but maybe not the only one. In general, I would not expect such studies but in this paper on a presentation level, this was very prominently exposed to the reader.
> >
> > Appreciating the work of the authors in their rebuttal and the practical use of PriorBand, I will increase my score to weak accept but only to that level due to the remaining points.

---

> > > ### Author Response · Authors · 2023-08-18
> > > **Re: Re: Rebuttal by Authors**
> > >
> > > We thank you for your revision, it is much appreciated.
> > >
> > > Regarding your point about a survey from practitioners discussing their needs, there are related works. In [1], a survey was conducted to show that tuning algorithms are either partially adopted or not at all in over 60% of cases. This survey includes demographic breakdowns and can provide insight into what can make HPO easy to integrate into existing pipelines, depending on the end-user or application.
> > >
> > > A more recent and relevant survey was conducted among various practitioners from novices to experts in different ML research areas [2]. The key findings are well summarized in Figures 1, 2, and 3 in the paper. Their survey identifies that _increased model performance_, _decreased compute requirements_  are the top-2 requirements for a practitioner to adopt HPO.
> > > Our list of desiderata includes these requirements while our empirical evaluation of PriorBand supports them.
> > >
> > > The third goal in Hasebrook et al. [2], focused on _reducing practitioner effort_, aligns closely with our aim for _simplicity_, which led us to adopt HyperBand as the foundation for PriorBand. Our approach employs straightforward early stopping and ranking, mirroring manual tuning practices. The survey [2] also mentions how practitioners often intend to not just tune but understand step by step _“what is working and what is not”_ (a quote from the survey [2]). This perspective underscores the significance of our Expert Prior Interface in an HPO algorithm. Additionally, in _Appendix E.3, Fig. 19_ illustrates how practitioners can perform post-hoc analysis to gauge whether the prior input remains pertinent to the problem at hand.
> > >
> > > Our specific list of desiderata aims to thus capture such requirements that make HPO more amenable in practice. Our contributed algorithm PriorBand ticks all these boxes while allowing model-based extensions, that is, PriorBand can be leveraged with Bayesian Optimization, thereby covering a large set of requirements as reported in Hasebrook et al. [2].
> > >
> > > We agree that it will be useful to back up our desiderata with references to these surveys and will do so in the camera-ready version; thanks a lot for the suggestion! Ours is the first work satisfying these desiderata; in particular, it is the first work to show how expert priors can be effectively applied to a multi-fidelity algorithm that not only benefits an HPO run but also is generally robust to any kind of expert input, making the decision of using HPO with expert priors a potential default in practice. Also, our contribution of dynamic weighting of samplers can be applied to _any_ algorithm which means that adding an expert prior interface to an existing implementation of an HPO algorithm is rather trivial while satisfying the desiderata we identify.
> > >
> > > ### ___
> > >
> > > We highly appreciate your time and comments and are thankful for your feedback. If there are any other points we can clarify to increase your confidence further or improve our final draft, we are happy to hear them.
> > >
> > > ### References:
> > >
> > > [1] van der Blom et al., AutoML Adoption in ML Software, 2021
> > >
> > > [2] Hasebrook et al., Practitioner Motives to Select Hyperparameter Optimization Methods 2023

---

### Official Review · Reviewer_ejik · 2023-07-06

**Soundness:** 4 excellent
**Presentation:** 4 excellent
**Contribution:** 3 good
**Rating:** 6
**Confidence:** 4

**Summary:**

This paper presents PriorBand, a hyperparameter optimization (HPO) algorithm designed specifically for deep learning models. PriorBand fulfills six key requirements and tackles the shortcomings of existing HPO methods that are unsuitable for DL. It leverages cheap proxy tasks while considering expert input. The algorithm eliminates the need for a naive solution for integrating expert domain knowledge into HPO. Experiments across a wide array of DL tasks are conducted to demonstrate the efficiency and robustness of PriorBand.

**Strengths:**

+ The proposed algorithm is simple yet effective. The augmented prior based on the existing HyperBand algorithm is technically sound and exhibits clear improvement over baselines.

+ The paper is well-written and well-structured. Motivation is clearly stated before the introduction of detailed algorithms (e.g., Sect. 4).

**Weaknesses:**

- Though the experiments conducted in the paper are quite comprehensive, I found most of them are moved to the appendix where the figures in the main paper are of limited information (most are about robustness to bad priors). I would suggest authors rearrange the paper by moving figures from Appendix F to the main text.

- In Table 1, given the fact that all the baseline methods satisfy "Mixed search spaces" + "Speedup under parallelism", I would suggest the authors remove these two rows as they are also not the major technical contributions of the proposed method. Maybe instead, replace with one sentence in the caption highlighting both the baselines and the proposed method can satisfy these two criteria.

**Questions:**

See Weakness.

**Limitations:**

See Weakness.

---

> ### Author Rebuttal · Authors · 2023-08-08
>
> We’d like to thank the reviewer for their kind remarks of appreciation for the method itself and we are delighted to read that the effort put into the presentation, structure, and motivation made the paper clear.
>
> ---
> > Though the experiments conducted in the paper are quite comprehensive, I found most of them are moved to the appendix where the figures in the main paper are of limited information (most are about robustness to bad priors). I would suggest authors rearrange the paper by moving figures from Appendix F to the main text.
> ---
>
> 1.
>   **a)** We appreciate that the reviewer took the time to read our Appendix and we agree with the reviewer's remarks at large. We decided on a set of plots aggregated across tasks in the main paper, that are focused on the primary message of how the Ensemble Sampling Policy (ESP) improves performance. For the camera-ready, we plan to move _Fig. 21_ and _Fig. 22_ to the main paper.
>
>   **b)** As an alternative, we now also created average normalized regret plots which we show on the _additional page_ we can upload during the rebuttal. They could substitute average relative ranks as an alternative way of communicating aggregated results over benchmarks.
>
>   We kindly request the reviewer to check these plots to see if they are a more suitable set of visualizations for the final main draft.
>
> ---
> > In Table 1, given the fact that all the baseline methods satisfy "Mixed search spaces" + "Speedup under parallelism", I would suggest the authors remove these two rows as they are also not the major technical contributions of the proposed method. Maybe instead, replace with one sentence in the caption highlighting both the baselines and the proposed method can satisfy these two criteria.
> ---
>
> 2. Thank you for pointing this out, it is indeed correct. However, we would like to clarify that we do not intend the desiderata to be technical contributions, but an overview of the HPO approach landscape. In fact, in terms of technical contributions, the Ensemble Sampling Policy, which is our major contribution, improves both anytime performance and final performance and can therefore not be isolated into any one of the 7 criteria in Table 1. In the end, we believe users of an HPO system are agnostic to *how* a desideratum such as efficiency is fulfilled, as long as it is indeed fulfilled.
>
>
> ### ___
>
> We value the reviewer's input and recommendations and invite the reviewer to explore the additional PDF we've provided. We are happy to address additional questions to increase your confidence in the review.

---

> > ### Comment · Reviewer_ejik · 2023-08-17
> > **Re: Rebuttal**
> >
> > Thanks for the authors' reply. The rebuttal has resolved my concerns. I am not mainly working on hyperparameter optimization so I cannot faithfully judge the novelty + the technical contribution of the paper compared with the existing SOTA ones, as concerned by Reviewer QGiN and meUK. However, the comprehensive comparison in the paper has convinced me of the method's effectiveness, so I will maintain my positive score.

---

### Author Rebuttal · Authors · 2023-08-08

To all reviewers and chairs,

We upload the permitted extra PDF with plots addressing a few points raised across the reviews.

**Figure 1)** shows the influence of a "good" prior on PriorBand, per-dataset under the reproducible protocol for prior generation in our experiments (_Appendix D.3_). This is an alternative to the mean relative ranks in the draft.

In this plot, we show the mean normalized regret over iterations for PriorBand per benchmark. The regret is calculated by normalizing all values between the minimum and maximum values seen across all included algorithms and seeds for each benchmark.

**Figure 2)** shows that the absolute performance of PriorBand is significant over the baselines, something not visible in relative ranking plots. It shows the mean normalized regret averaged across all benchmarks under different strengths of priors. This demonstrates both strong anytime and final performance of PriorBand across different prior qualities, under tractable budgets.

The second from right plot in this figure is the aggregated view of Figure 1.

**Figure 3)** highlights that our chosen hyperparameters for PriorBand offer a good balance between exploration and exploitation. This figure shows two ablation studies over the PriorBand local search hyperparameters: (i) the standard deviation of the local perturbation, and (ii) the mutation rate, which determines the chance of selection of each hyperparameter for perturbation.

More exploitative local search hyperparameters naturally give gains under strong priors. However, given the goal of being robust across all possible scenarios, a conservative local search (our default for PriorBand) offers steady anytime gains under different prior qualities. This allows PriorBand to be truly practical for HPO.

We have provided further captions to explain the figures but we are happy to clarify further if required.

We are looking forward to an engaged discussion period!

---

### Decision · Program_Chairs · 2023-09-21

**Decision:**

Accept (poster)

**Comment:**

The paper provides an approach for hyper paramater optimization (HPO), suited for deep learning. The method provided by the authors doesn’t contain an entirely new technique, rather carefully combines several existing methods. A major concern raised by some of the reviewers points this out as a lack of novelty. This being said, the reviewers agree that the provided combination of techniques is sensible and well described. The paper provides a good survey of existing works and a thorough set of experiments analyzing the technique and proving its efficiency compared to SoTA techniques. Given that multi-fidelty HPO already has many published paper, being able to innovate there is not an easy task. With this in mind, along with the paper’s mentioned strengths, I think the paper would be of intererst to the NeurIPS community.